# Follow-the-Perturbed-Leader for Adversarial Markov Decision Processes with Bandit Feedback

**Yan Dai**
Tsinghua University
yan-dai20@mails.tsinghua.edu.cn

**Haipeng Luo**
University of Southern California
haipengl@usc.edu

**Liyu Chen**
University of Southern California
liyuc@usc.edu

## Abstract

We consider regret minimization for Adversarial Markov Decision Processes (AMDPs), where the loss functions are changing over time and adversarially chosen, and the learner only observes the losses for the visited state-action pairs (i.e., bandit feedback). While there has been a surge of studies on this problem using Online-Mirror-Descent (OMD) methods, very little is known about the Follow-the-Perturbed-Leader (FTPL) methods, which are usually computationally more efficient and also easier to implement since it only requires solving an offline planning problem. Motivated by this, we take a closer look at FTPL for learning AMDPs, starting from the standard episodic finite-horizon setting. We find some unique and intriguing difficulties in the analysis and propose a workaround to eventually show that FTPL is also able to achieve near-optimal regret bounds in this case. More importantly, we then find two significant applications: First, the analysis of FTPL turns out to be readily generalizable to delayed bandit feedback with order-optimal regret, while OMD methods exhibit extra difficulties (Jin et al., 2022). Second, using FTPL, we also develop the first no-regret algorithm for learning communicating AMDPs in the infinite-horizon setting with bandit feedback and stochastic transitions. Our algorithm is efficient assuming access to an offline planning oracle, while even for the easier full-information setting, the only existing algorithm (Chandrasekaran and Tewari, 2021) is computationally inefficient.

## 1 Introduction

Markov Decision Processes (MDPs) have long been used to model problems in reinforcement learning, where the agent takes sequential actions in an environment, leading to transitions among different states and observations on loss (or reward equivalently) signals. While the classical MDP model assumes a fixed loss function, there has been increasing interest in studying regret minimization under non-stationary or even adversarial loss functions via the Adversarial MDP (AMDP) model, starting from the work of Even-Dar et al. (2009).

Similar to other regret minimization problems, there are typically two categories of algorithms for AMDPs: those based on the Follow-the-Perturbed-Leader (FTPL) framework (Even-Dar et al., 2009; Neu et al., 2010, 2012; Chandrasekaran and Tewari, 2021) and those based on the Online-Mirror-Descent (OMD) or the closely related Follow-the-Regularized-Leader (FTRL) framework (Zimin and Neu, 2013; Rosenberg and Mansour, 2019a,b; Jin et al., 2020, 2021, 2022). FTPL methods are usually computationally more efficient and easier to implement as it only requires solving an offline

36th Conference on Neural Information Processing Systems (NeurIPS 2022).

Table 1: An overview of the proposed algorithms/results and comparisons with related works.

| Setting | Transition | Feedback | Algorithm | Regret[a] | Method | Effi.[b] |
|---------|-----------|----------|-----------|-----------|--------|----------|
| Episodic $H$-horizon AMDPs | Known | Bandit | Zimin and Neu (2013) | $\widetilde{\mathcal{O}}(H\sqrt{SAK})$ | OMD | ✓ |
| | | | **This work** (Theorem 4) | $\widetilde{\mathcal{O}}(H^{3/2}\sqrt{SAK})$ | FTPL | ✓ |
| | Unknown | Bandit | Jin et al. (2020) | $\widetilde{\mathcal{O}}(H^2S\sqrt{AK})$ | OMD | ✓ |
| | | | **This work** (Theorem 5) | $\widetilde{\mathcal{O}}(H^2S\sqrt{AK})$ | FTPL | ✓ |
| | | Bandit & Delayed | Delayed HEDGE (Jin et al., 2022) | $\widetilde{\mathcal{O}}(H^2S\sqrt{AK}+H^{3/2}\sqrt{S\mathfrak{D}})$ | OMD | ✗ |
| | | | Delayed UOB-FTRL (Jin et al., 2022) | $\widetilde{\mathcal{O}}(H^2S\sqrt{AK}+H^{3/2}SA\sqrt{\mathfrak{D}})$ | OMD | ✓ |
| | | | Delayed UOB-REPS (Jin et al., 2022) | $\widetilde{\mathcal{O}}(H^2S\sqrt{AK}+H^{5/4}(SA)^{1/4}\sqrt{\mathfrak{D}})$ | OMD | ✓ |
| | | | **This work** (Theorem 6) | $\widetilde{\mathcal{O}}(H^2S\sqrt{AK}+H^{3/2}\sqrt{SA\mathfrak{D}})$ | FTPL | ✓ |
| Infinite-horizon AMDPs | Known | Full-info | Even-Dar et al. (2009) | $\widetilde{\mathcal{O}}(\tau^2\sqrt{T})$ (Ergodic) | OMD | ✓ |
| | | | Chandrasekaran and Tewari (2021) | $\widetilde{\mathcal{O}}(S^4\sqrt{T})$ (Deterministic) | FTPL | ✓ |
| | | | Chandrasekaran and Tewari (2021) | $\widetilde{\mathcal{O}}(D^2\sqrt{ST})$ (Commu) | FTPL | ✗ |
| | | Bandit | Neu et al. (2014) | $\widetilde{\mathcal{O}}(\sqrt{\tau^3AT})$ (Ergodic) | OMD | ✓ |
| | | | Dekel and Hazan (2013) | $\widetilde{\mathcal{O}}(S^3AT^{2/3})$ (Deterministic) | OMD | ✓ |
| | | | **This work** (Theorem 7) | $\widetilde{\mathcal{O}}(A^{1/2}(SD)^{2/3}T^{5/6})$ (Commu) | FTPL | ✓! |
| | | | **This work** (Theorem 8) | $\widetilde{\mathcal{O}}(A^{1/3}(SDT)^{2/3})$ (Commu) | OMD | ✗ |
| | | | Dekel et al. (2014) | $\Omega(S^{1/3}T^{2/3})$ (if only Commu) | — | — |

[a]Here, $S$ and $A$ are the number of states and actions respectively, $K$ is the number of episodes, $T$ is the total number of steps, $\mathfrak{D}$ is the total amount of delay, $\tau$ is the mixing time of an ergodic MDP, and $D$ is the diameter of a communicating MDP. Several related works use different notations from ours, and their regret bounds have been converted based on our notations. For infinite-horizon AMDPs, the extra assumptions are listed after the regret bounds, with "Ergodic" standing for ergodic MDPs, "Deterministic" standing for MDPs with deterministic transitions, and "Commu" standing for communicating MDPs (the weakest assumption).

[b]This column indicates the algorithm's efficiency: ✓ means polynomial (in all parameters) time complexity, ✗ means $\Omega(A^S)$ time complexity, and ✓! means efficient assuming access to a planning oracle (that returns the best policy given all the MDP's parameters). Note that FTPL-based algorithms are usually easier to implement compared to OMD/FTRL-based ones (both treated as OMD-based in this table as they are quite similar).

optimization problem (a.k.a. a *planning* problem in the MDP literature). In contrast, OMD/FTRL methods require solving convex optimization problems over a complicated occupancy measure space.

Despite its computational advantages and ease in implementation, FTPL methods are much less studied (especially for learning AMDPs) since they are harder to analyze, less versatile, and are believed to suffer worse regret compared to OMD/FTRL methods. A recent work by Wang and Dong (2020) disputes the last common belief and shows that, for episodic AMDPs with full-information feedback, FTPL also enjoys near-optimal regret, similarly to OMD/FTRL. Nevertheless, little is known about FTPL for learning AMDPs with the more challenging bandit feedback — to our knowledge, the only FTPL algorithm for this case is by Neu et al. (2010). However, that algorithm is analyzed under a strong assumption that every state is reachable by any policy with at least a constant probability $\alpha > 0$. Such an exploratory assumption is too strong to be used in realistic applications.

Motivated by this fact, we take a closer look at FTPL for learning AMDPs under bandit feedback, aiming at showing strong regret guarantees while enjoying its computational advantages. We start with the standard episodic finite-horizon setting and indeed find some intriguing difficulties compared to OMD/FTRL. After addressing these difficulties, we then show critical applications of FTPL methods to two more challenging setups: episodic AMDPs with *delayed* bandit feedback and *infinite-horizon* AMDPs with only communicating assumptions, with the latter result advancing the state-of-the-art. More specifically, our contributions are (see also Table 1 for a summary):

1. We start with the heavily studied episodic setting with $K$ episodes, $H$ steps in each episode, $S$ states, and $A$ actions. Our first intriguing observation is that: since the loss of each policy is linear in a non-binary vector (i.e., the occupancy measure), existing analysis for the stability term of FTPL fails, even though it works for the binary case (e.g., Neu and Bartók (2016)). Our next important observation is that there exists a simple fix to this issue that only leads to an extra $H$ factor. This eventually leads to $\widetilde{\mathcal{O}}(H^{3/2}\sqrt{SAK})$ regret when the transition is known (Algorithm 1, Theorem 4), which is only $\sqrt{H}$ factor larger than the near-optimal regret achieved

by OMD (Zimin and Neu, 2013), and $\widetilde{\mathcal{O}}(H^2 S\sqrt{AK})$ regret when the transition is unknown (Algorithm 3, Theorem 5), matching the state-of-the-art again achieved by OMD (Jin et al., 2020). See Section 3 for details.

2. We next find that compared to OMD, the analysis of FTPL is much easier to be generalized to the delayed feedback setting where losses for episode $k$ are observed only at the end of episode $k + d_k$ for some $d_k \geq 0$ (Lancewicki et al., 2022; Jin et al., 2022). Indeed, these two prior works demonstrate the difficulty of analyzing OMD with delay feedback, with Lancewicki et al. (2022) only achieving $\widetilde{\mathcal{O}}((K+\mathfrak{D})^{2/3})$ regret (where $\mathfrak{D} = \sum_k d_k$ is the total amount of delay; dependence on other parameters is omitted) and Jin et al. (2022) improving it to $\widetilde{\mathcal{O}}(\sqrt{K+\mathfrak{D}})$ via either an inefficient algorithm or an efficient OMD-based algorithm with more involved analysis and/or new delayed-adapted loss estimators. FTPL, on the other hand, achieves $\widetilde{\mathcal{O}}(\sqrt{K+\mathfrak{D}})$ regret by a simple extension of the analysis (Theorem 6). The dependence on $S$ and $A$ is also better than the OMD method of (Jin et al., 2022) with the same kind of standard loss estimators (though worse than their best result with the delayed-adapted estimators; see Table 1 and Section 4 for details).

3. While our results above do not improve the best existing ones, our final application of FTPL provides the *first* result for learning infinite-horizon communicating AMDPs with bandit feedback and known stochastic transitions. Specifically, our algorithm achieves $\widetilde{\mathcal{O}}(A^{1/2}(SD)^{2/3}T^{5/6})$ regret (Algorithm 6, Theorem 7), where $D$ is the diameter of the MDP and $T$ is the total number of steps. It is efficient assuming access to an offline planning oracle (that returns the best stationary policy given a fixed transition function and a sequence of loss functions for each step). Previous results either only handle deterministic transitions (Dekel and Hazan, 2013) or full-information loss feedback (Chandrasekaran and Tewari, 2021). Moreover, the FTPL algorithm of Chandrasekaran and Tewari (2021) for stochastic transitions is inefficient even given the same planning oracle (since it explicitly adds independent noise to *every* policy). For completeness, we also provide an inefficient algorithm (Algorithm 7) that achieves $\widetilde{\mathcal{O}}(A^{1/2}(SDT)^{2/3})$ regret in our bandit setting, matching the $\Omega(T^{2/3})$ lower bound of Dekel et al. (2014) in terms of $T$. See Section 5 for details.

## 1.1 Related Work

**Follow-the-Perturbed-Leader:** FTPL is first proposed by Hannan (1957) and later popularized by Kalai and Vempala (2005). It has proven to be extremely powerful for structured online learning problems (such as online shortest path) since its implementation is as easy as solving the corresponding offline optimization problem (such as finding the shortest path of a given graph). Over the years, FTPL has been extended to problems with semi-bandit feedback (Neu, 2015; Neu and Bartók, 2016), contextual information (Syrgkanis et al., 2016), non-linear losses (Dudík et al., 2020), smoothed adversaries (Block et al., 2022; Haghtalab et al., 2022), and others. However, FTPL for learning AMDPs under bandit feedback is poorly understood, which motivates this work. As we successfully show, improving our understanding of FTPL is indeed beneficial since it at least leads to new results for the infinite-horizon setting (in addition to its computational advantages for other settings). Below, we briefly review the literature of AMDPs for the three settings we consider.

**Episodic Finite-Horizon AMDPs:** Earlier works on this topic focus on the easier known transition case. In particular, the OMD-based O-REPS algorithm by Zimin and Neu (2013) achieves $\widetilde{\mathcal{O}}(H\sqrt{K})$ regret with full-information feedback and $\widetilde{\mathcal{O}}(H\sqrt{SAK})$ regret with bandit feedback, both optimal up to logarithmic factors. On the other hand, FTPL is recently shown to achieve $\widetilde{\mathcal{O}}(H^2\sqrt{K})$ regret with full-information feedback (Wang and Dong, 2020). As mentioned, the only FTPL algorithm for bandit feedback is by Neu et al. (2010), which guarantees $\widetilde{\mathcal{O}}(H^2\sqrt{AK}/\alpha)$ regret assuming that all states are reachable by any policy with a probability of at least $\alpha$. In contrast, our FTPL algorithm removes this requirement and achieves $\widetilde{\mathcal{O}}(H^{3/2}\sqrt{SAK})$ regret, which is only $\sqrt{H}$ away from optimal.

When the transition is unknown, with full-information feedback, the OMD-based algorithm UC-O-REPS (Rosenberg and Mansour, 2019a) achieves $\widetilde{\mathcal{O}}(H^2 S\sqrt{AK})$ regret, while the FTPL-based FPOP (Neu et al., 2012) is shown to achieve $\widetilde{\mathcal{O}}(H^2 S\sqrt{AK})$ regret as well (Wang and Dong, 2020). With bandit feedback, the OMD-based algorithm UOB-REPS (Jin et al., 2020) also achieves the same $\widetilde{\mathcal{O}}(H^2 S\sqrt{AK})$ regret. At the same time, our algorithm enjoys the same guarantee and is the first FTPL algorithm for bandit feedback and unknown transition. However, the current best lower bound for this problem is $\Omega(H^{3/2}\sqrt{SAK})$ (Jin et al., 2018), so there is still an $\mathcal{O}(\sqrt{HS})$ gap.

Besides OMD and FTPL, there is, in fact, another category of algorithms for learning AMDPs: policy optimization (Shani et al., 2020; Luo et al., 2021), which performs OMD in each *state* and is also efficient. However, the regret bounds are worse by at least an $H$ factor (Luo et al., 2021).

**Delayed Feedback:** The most related works are Lancewicki et al. (2022) and Jin et al. (2022), and we refer the reader to the references therein for the literature on delayed feedback for different problems. Importantly, Jin et al. (2022) point out the unique difficulty when analyzing OMD/FTRL for AMDPs with delayed feedback. Circumventing this difficulty one way or another, they develop three algorithms: the first one, Delayed HEDGE, is inefficient; the second one, Delayed UOB-FTRL, achieves worse regret ($\sqrt{SA}$ larger for the delay-related term) compared to ours; and the third one makes use of a delay-adapted estimator and achieves the best bound (see Table 1). We emphasize again that our FTPL analysis is much simpler and a direct extension of the non-delayed case. The current best lower bound for this problem is $\Omega(H^{3/2}\sqrt{SAK} + H\sqrt{\mathfrak{D}})$ (Lancewicki et al., 2022).

**Infinite-Horizon AMDPs:** Learning AMDPs becomes significantly more difficult in the infinite horizon setting. As far as we know, all works in this line (including ours) assume a known transition function. Earlier works focus on the simpler case with a strong *ergodic* assumption (Even-Dar et al., 2009; Neu et al., 2014). For the more general *communicating* assumptions, a recent work (Chandrasekaran and Tewari, 2021) considers full-information feedback and develops an efficient FTPL algorithm for deterministic transitions with $\widetilde{\mathcal{O}}(S^4\sqrt{T})$ regret and another inefficient FTPL algorithm for stochastic transitions with $\widetilde{\mathcal{O}}(D^2\sqrt{ST})$ regret. Under bandit feedback, prior works only study deterministic transitions (Arora et al., 2012; Dekel and Hazan, 2013), with Dekel and Hazan (2013) achieving $\widetilde{\mathcal{O}}(S^3AT^{2/3})$ regret, matching the lower bound (Dekel et al., 2014) for the $T$-dependency. Our results are the first for bandit feedback and stochastic transitions. Note that since bandit feedback is only more general, our oracle-efficient algorithm can also be applied to the full-information setting, while the only existing algorithm (Chandrasekaran and Tewari, 2021) is computationally inefficient.

## 2   Preliminaries

**General Notations:** We use $[N]$ to denote the set $\{1, 2, \ldots, N\}$. For a (finite) set $X$, we use $\triangle(X) \triangleq \{x \in \mathbb{R}_{\geq 0}^{|X|} \mid \sum_{i=1}^{|X|} x_i = 1\}$ to denote the probability simplex over the set $X$. We use $\widetilde{\mathcal{O}}(\cdot)$ to hide all terms logarithmic in $H, S, A, K$ and $T$. Laplace$(\eta)$ denotes the Laplace (also known as double-exponential) distribution with center $0$ and parameter $\eta$, whose probability density is $f(x) = \frac{\eta}{2}\exp(-\eta|x|), \forall x \in \mathbb{R}$. For an event $\mathcal{E}$, let $\mathbb{1}[\mathcal{E}]$ be its indicator. In episodic settings, let $\{\mathcal{F}_k\}_{k=0}^K$ be the natural filtration such that $\mathcal{F}_k$ contains the history of episodes $1, \ldots, k$. With a slight abuse of notation, in the infinite-horizon setting, we also use $\{\mathcal{F}_t\}_{t=0}^T$ to denote the natural filtration.

**Episodic Adversarial Markov Decision Process:** An episodic Adversarial Markov Decision Process (AMDP) is defined by a tuple $\mathcal{M} = (\mathcal{S}, \mathcal{A}, \mathbb{P}, \ell, K, H, s^1)$, where $\mathcal{S}$ is the state space, $\mathcal{A}$ is the action space, $\mathbb{P}\colon [H] \times \mathcal{S} \times \mathcal{A} \to \triangle(\mathcal{S})$ is the transition function, $\ell\colon [K] \times [H] \times \mathcal{S} \times \mathcal{A} \to [0, 1]$ is the loss function unknown to the agent but fixed before the game (i.e., we are assuming an *oblivious* adversary),[1] $K$ is the number of episodes, $H$ is the horizon length, and $s^1 \in \mathcal{S}$ is the initial state. Denote by $S = |\mathcal{S}| < \infty$ and $A = |\mathcal{A}| < \infty$, the number of states and actions, respectively.

The agent interacts with the environment for $K$ episodes. For the $k$-th one ($k \leq K$), she starts from the initial state $s^1$ and sequentially interacts with the environment for $H$ steps. At the $h$-th step (where $h \in [H]$), the agent observes state $s_k^h \in \mathcal{S}$, chooses an action $a_k^h \in \mathcal{A}$, observes and suffers the loss $\ell_k^h(s_k^h, a_k^h)$ (bandit feedback),[2] and then transits to state $s_k^{h+1}$ according to the probability distribution $\mathbb{P}^h(\cdot \mid s_k^h, a_k^h)$. After $H$ steps, the episode ends and the agent proceeds to episode $k + 1$.

A (deterministic) *policy* of the agent is defined by $\pi = \{\pi^h\colon \mathcal{S} \to \mathcal{A}\}_{h \in [H]}$. Denote the set of all deterministic policies by $\Pi$. The expected loss incurred by policy $\pi \in \Pi$ for an episode with loss function $\widehat{\ell}$ is denoted by $V(\pi; \widehat{\ell}) \triangleq \mathbb{E}\left[\sum_{h=1}^H \widehat{\ell}^h(s^h, \pi^h(s^h)) \Big| s^{h+1} \sim \mathbb{P}^h(\cdot \mid s^h, \pi^h(s^h)), \forall h < H\right]$. Suppose the agent uses policies $\pi_1, \pi_2, \ldots, \pi_K$ for episodes $1, 2, \ldots, K$, respectively. The total expected loss of the agent is then $\mathbb{E}\left[\sum_{k=1}^K V(\pi_k; \ell_k)\right]$, where the expectation is taken with respect to

---

[1]Note that the loss function can vary arbitrarily for different $(k, h)$-pairs, instead of being stochastic.

[2]On the other hand, in the easier full-information setting, the entire $\ell_k^h$ is revealed.

the agent's private randomness. The baseline is the best deterministic policy in hindsight, defined by $\pi^* \in \operatorname{argmin}_{\pi \in \Pi} \sum_{k=1}^{K} V(\pi; \ell_k)$. The goal of the agent is to minimize her *regret* over $K$ episodes, which is the difference between her total loss and that of $\pi^*$, formally defined as

$$\mathcal{R}_K \triangleq \mathbb{E}\left[\sum_{k=1}^{K} V(\pi_k; \ell_k)\right] - \sum_{k=1}^{K} V(\pi^*; \ell_k).$$

**Episodic AMDPs with Delayed Feedback:** This setup is exactly the same as the episodic AMDPs, except that the feedback $\{\ell_k^h(s_k^h, a_k^h)\}_{h=1}^{H}$ for episode $k$ is only available after $d_k$ episodes, i.e., at the end of the $(k + d_k)$-th episode. Define $\mathfrak{D} = \sum_{k=1}^{K} d_k$ to be the total feedback delay, assumed to be known to the agent as this assumption can be easily relaxed via a doubling trick (Thune et al., 2019).[3]

**Infinite-Horizon AMDPs:** Similar to episodic AMDPs, infinite-horizon AMDPs is defined by a tuple $\mathcal{M} = (\mathcal{S}, \mathcal{A}, \mathbb{P}, \ell, T, s^1)$. Here, starting from the initial state $s^1 \in \mathcal{S}$, the agent interacts with the environment for $T$ total steps without any reset, under the transition model $\mathbb{P}: \mathcal{S} \times \mathcal{A} \to \triangle(\mathcal{S})$ (which does not vary over time) and loss functions $\ell: [T] \times \mathcal{S} \times \mathcal{A} \to [0, 1]$. More specifically, at time $t \in [T]$, the agent observes state $s^t \in \mathcal{S}$, chooses an action $a^t \in \mathcal{A}$, observes and suffers loss $\ell^t(s^t, a^t)$, and then transits to $s^{t+1} \sim \mathbb{P}(\cdot \mid s^t, a^t)$. Her goal is also to minimize the regret, defined as

$$\mathcal{R}_T \triangleq \mathbb{E}\left[\sum_{t=1}^{T} \ell^t(s^t, a^t)\middle| s^{t+1} \sim \mathbb{P}(\cdot \mid s^t, a^t)\right] - \min_{\pi \in \Pi} \mathbb{E}\left[\sum_{t=1}^{T} \ell^t(s^t, \pi(s^t))\middle| s^{t+1} \sim \mathbb{P}(\cdot \mid s^t, \pi(s^t))\right],$$
(1)

where $\Pi$ is now the set of all deterministic policies mapping from $\mathcal{S}$ to $\mathcal{A}$. As pointed out by Bartlett and Tewari (2009), without any extra assumptions, sublinear regret is impossible for this problem due to the lack of resets. Earlier works make a strong *ergodic* assumption such that, intuitively, any mistake will be forgiven after logarithmic steps (Even-Dar et al., 2009). Here, we instead focus on the much weaker *communicating* assumption as in Chandrasekaran and Tewari (2021):

**Definition 1** (Communicating MDP). *We call an MDP $\mathcal{M}$ communicating if it has a finite diameter $D \triangleq \max_{s \neq s'} \min_{\pi \in \Pi} \mathbb{E}[T(s' \mid \mathcal{M}, \pi, s)]$ where $T(s' \mid \mathcal{M}, \pi, s)$ is the (random) time step when state $s'$ is first reached by policy $\pi$ starting from state $s$.*

Just like Chandrasekaran and Tewari (2021), for technical reasons, we also need the following mild assumption saying that there exists a special state for the agent to "park" there without moving.

**Assumption 2.** *There exist state $s^* \in \mathcal{S}$ and action $a^* \in \mathcal{A}$ such that $\mathbb{P}(s^* \mid s^*, a^*) = 1$.*

## 3 FTPL for Episodic AMDPs

In this section, we consider the basic (non-delayed) episodic setting. To best illustrate the unique difficulty we meet when analyzing FTPL and the way we address it, we first discuss the known-transition case (i.e., $\{\mathbb{P}^h\}_{h=1}^{H}$ is known to the agent), and then move on to unknown transitions.

### 3.1 Known Transition

Our algorithm follows the standard FTPL framework (see Algorithm 1). Ahead of time (as the adversary is oblivious), we sample a perturbation vector $z: [H] \times \mathcal{S} \times \mathcal{A} \to \mathbb{R}$ so that $z^h(s, a)$ is an independent sample from $\operatorname{Laplace}(\eta)$ for some parameter $\eta$. At the beginning of episode $k$, given the loss estimators $\widehat{\ell}_1, \ldots, \widehat{\ell}_{k-1}$ from previous episodes (whose construction will be specified later), we simply play the policy that minimizes the cumulative perturbed estimated loss (break tie arbitrarily):

$$\pi_k = \operatorname{argmin}_{\pi \in \Pi}\left(V(\pi; z) + \sum_{k'=1}^{k-1} V(\pi; \widehat{\ell}_{k'})\right) = \operatorname{argmin}_{\pi \in \Pi} V\left(\pi; \widehat{\ell}_{0:k-1}\right),$$

where we use $\widehat{\ell}_{l:r}$ (where $0 \leq l \leq r \leq K$) as a shorthand notation for $\sum_{k'=l}^{r} \widehat{\ell}_{k'}$ and $\widehat{\ell}_0$ as an alias for $z$ for notational convenience. This optimization over $\pi \in \Pi$ is a simple planning problem and can be solved by dynamic programming efficiently.

---

[3]As in Jin et al. (2022), we only consider delayed loss feedback, but not delayed trajectory feedback, since the latter only affects the transition estimation and can be handled similarly to Lancewicki et al. (2022).

---

**Algorithm 1** FTPL for Episodic AMDPs with Bandit Feedback and Known Transition

---

**Require:** Laplace distribution parameter $\eta$. Geometric Re-sampling parameter $L$.

1: Sample perturbation $\widehat{\ell}_0 = z$ such that $z^h(s, a)$ is an independent sample of Laplace$(\eta)$.
2: **for** $k = 1, 2, \ldots, K$ **do**
3:      Calculate $\pi_k = \arg\min_{\pi \in \Pi} V(\pi; \widehat{\ell}_{0:k-1})$ (via dynamic programming).
4:      **for** $h = 1, 2, \ldots, H$ **do**
5:          Observe $s_k^h$, play $a_k^h = \pi_k(s_k^h)$, suffer and observe loss $\ell_k^h(s_k^h, a_k^h)$.
6:          Calculate loss estimator $\widehat{\ell}_k^h$ via Geometric Re-sampling (Neu and Bartók, 2016):
7:          **for** $M_k^h = 1, 2, \ldots, L$ **do**
8:              Sample a fresh perturbation $\widetilde{z}$ in the same way as $z$.
9:              Calculate $\pi_k' = \arg\min_{\pi \in \Pi} V(\pi; \widehat{\ell}_{1:k-1} + \widetilde{z})$.
10:            Simulate $\pi_k'$ for $h$ steps starting from $s^1$ and following transitions $\mathbb{P}^1, \ldots, \mathbb{P}^h$.
11:            **if** $(s_k^h, a_k^h)$ is visited at step $h$ or $M_k^h = L$ **then**
12:               Set $\widehat{\ell}_k^h(s, a) = M_k^h \cdot \ell_k^h(s_k^h, a_k^h) \cdot \mathbb{1}[(s_k^h, a_k^h) = (s, a)]$ and break.

---

Upon seeing $s_k^h$, $a_k^h$, and $\ell_k^h(s_k^h, a_k^h)$, we construct the loss estimator $\widehat{\ell}_k^h$ using the Geometric Re-sampling technique (Neu and Bartók, 2016). The idea is to repeat the sampling procedure (Line 8 to 10) until the same pair $(s_k^h, a_k^h)$ is visited again at step $h$ or this has been repeated $L$ times for some parameter $L$. Let the total number of trials be $M_k^h$, then the estimator is defined as $\widehat{\ell}_k^h(s, a) = M_k^h \cdot \ell_k^h(s_k^h, a_k^h) \cdot \mathbb{1}[(s_k^h, a_k^h) = (s, a)]$ (Line 12). Note that the sampling procedure can be done freely without interacting with the environment as the transition is known. The rational behind this estimator is that as long as $L$ is reasonably large, $M_k^h$ is a good approximation of the inverse probability of visiting $(s_k^h, a_k^h)$ (which is hard to calculate directly for FTPL), making $\widehat{\ell}_k^h$ a good (and efficient) approximation of the standard importance weighted estimator (Zimin and Neu, 2013).

**Analysis Sketch:** While our algorithm follows the standard FTPL framework, we find some intriguing difficulty in the analysis that is unique to MDPs and undiscovered before. To illustrate this difficulty, let us first describe an overview of the analysis. First, since the loss estimators are almost unbiased (as shown by Neu and Bartók (2016)), we only need to focus on the regret with respect to the estimated losses, that is, $\mathbb{E}\left[\sum_{k=1}^{K} V(\pi_k; \widehat{\ell}_k) - \sum_{k=1}^{K} V(\pi^*; \widehat{\ell}_k)\right]$. Adding and subtracting $\mathbb{E}\left[\sum_{k=1}^{K} V(\pi_{k+1}; \widehat{\ell}_k)\right]$ (the loss of an imaginary "leader" that looks one episode ahead), our next goal is to bound the so-called *stability term* $\mathbb{E}\left[\sum_{k=1}^{K} V(\pi_k; \widehat{\ell}_k) - \sum_{k=1}^{K} V(\pi_{k+1}; \widehat{\ell}_k)\right]$ (the rest, usually referred as the *error term*, can be bounded by the standard "be-the-leader" lemma).

For the stability term, fix an episode $k$ and define $p_k(\pi)$ as the probability of selecting $\pi$ as $\pi_k$ w.r.t. the randomness of the perturbation $z$. Further introduce the notion of *occupancy measures* (Altman, 1999; Neu et al., 2012): each policy $\pi \in \Pi$ induces $H$ occupancy measures $\mu_\pi^h \in \triangle(\mathcal{S} \times \mathcal{A})$, $\forall h \in [H]$, where $\mu_\pi^h(s, a)$ denotes the probability of visiting $(s, a)$ at step $h$ if one executes policy $\pi$ starting from the initial state $s^1$. With these notations, each summand for the stability term becomes:

$$\mathbb{E}\left[V(\pi_k; \widehat{\ell}_k) - V(\pi_{k+1}; \widehat{\ell}_k)\right] = \mathbb{E}\left[\sum_{\pi \in \Pi} (p_k(\pi) - p_{k+1}(\pi)) \left\langle \mu_\pi, \widehat{\ell}_k \right\rangle\right],$$

where $\left\langle \mu_\pi, \widehat{\ell}_k \right\rangle \triangleq \sum_{h=1}^{H} \left\langle \mu_\pi^h, \widehat{\ell}_k^h \right\rangle$. This stability term is exactly in the same form as that in Lemma 8 of Neu and Bartók (2016) or Lemma 10 of Syrgkanis et al. (2016) for (contextual) semi-bandit problems, except that in their contexts, $\mu_\pi$ is a *binary* vector. This seemingly slight difference turns out to be important! Specifically, in these two prior works, they both show (using our notations):

$$p_{k+1}(\pi) \geq p_k(\pi) \exp\left(-\eta \left\langle \mu_\pi, \widehat{\ell}_k \right\rangle\right), \tag{2}$$

which, together with the fact $\exp(-x) \geq 1 - x$, implies

$$\mathbb{E}\left[V(\pi_k; \widehat{\ell}_k) - V(\pi_{k+1}; \widehat{\ell}_k)\right] \leq \eta \, \mathbb{E}\left[\sum_{\pi \in \Pi} p_k(\pi) \left\langle \mu_\pi, \widehat{\ell}_k \right\rangle^2\right]. \tag{3}$$

Readers familiar with the online learning literature would have recognized the last expression, since it is also the standard stability term achieved by (inefficiently) running the classical HEDGE

algorithm (Freund and Schapire, 1997) over all policies (see e.g. Theorem 7.3 of Bubeck (2011)). Indeed, this term is small enough and can be shown to be of order $\mathcal{O}(\eta HSA)$ in our context after plugging in the definition of the loss estimators, which would then basically complete the proof.

However, not only do we realize that the proof of Eq. (2) heavily rely on the binary nature of $\mu_\pi$, we in fact also find a counterexample where Eq. (3) is simply *incorrect* when $\mu_\pi$ is non-binary (see Appendix B.1.5 for the counterexample). We find this fact intriguing, because Eq. (3) holds for the aforementioned inefficient HEDGE algorithm regardless whether $\mu_\pi$ is binary or not.

Further examining the proof of Neu and Bartók (2016) and Syrgkanis et al. (2016), however, one can prove the following weaker version of Eq. (2) and Eq. (3) (namely Eq. (4) and Eq. (5) respectively).

**Lemma 3** (Single-Step Stability). *For all $k \in [K]$ and $\pi \in \Pi$, we have*

$$p_{k+1}(\pi) \geq p_k(\pi) \exp\left(-\eta \sum_{h=1}^{H} \|\widehat{\ell}_k^h\|_1\right), \tag{4}$$

*and thus*

$$\mathbb{E}\left[V(\pi_k; \widehat{\ell}_k) - V(\pi_{k+1}; \widehat{\ell}_k)\right] \leq \eta \, \mathbb{E}\left[\left(\sum_{h=1}^{H} \|\widehat{\ell}_k^h\|_1\right) \sum_{\pi \in \Pi} p_k(\pi) \left\langle \mu_\pi, \widehat{\ell}_k \right\rangle\right]. \tag{5}$$

Fortunately, while Eq. (5) looks seemingly much larger than the classic bound Eq. (3), it is in fact at most larger by an $H$ factor, that is, the right-hand side of Eq. (5) can be shown be of order $\mathcal{O}(\eta H^2 SA)$ (see Lemma 12 in the appendix). Putting everything together, this allows us to prove the following regret guarantee for Algorithm 1, which is $\sqrt{H}$ larger than the optimal bound (Zimin and Neu, 2013) due to the weakened stability bound. One may refer to Appendix B.1 for the formal proof.

**Theorem 4.** *For episodic AMDPs with bandit feedback and known transitions, Algorithm 1 with $\eta = 1/\sqrt{HSAK}$ and $L = \sqrt{SAK/H}$ ensures $\mathcal{R}_T = \widetilde{\mathcal{O}}\left(H^{3/2}\sqrt{SAK}\right)$.*

## 3.2 Unknown Transition

To handle unknown transitions, we mostly follow existing ideas. First, for each episode $k$ we maintain a confidence set $\mathcal{P}_k$ of the transition function as Jin et al. (2022), whose construction is given in Appendix B.2.1. These confidence sets ensure that i) $\mathbb{P} \in \mathcal{P}_k$ with high probability and ii) $\mathcal{P}_{k+1} \subseteq \mathcal{P}_k$. Generalizing the notation $V(\pi; \widehat{\ell})$, we use $V(\pi; \widehat{\ell}, P)$ to denote the expected loss of policy $\pi$ for an episode with loss function $\widehat{\ell}$ and transition $P$ (so $V(\pi; \widehat{\ell}) = V(\pi; \widehat{\ell}, \mathbb{P})$). Then deploying the idea of optimism, we replace Line 3 of Algorithm 1 with $\pi_k = \operatorname{argmin}_{\pi \in \Pi} \min_{P \in \mathcal{P}_k} V(\pi; \widehat{\ell}_{0:k-1}, P)$, which can be efficiently found using Extended Value Iteration (Jaksch et al., 2010). As Wang and Dong (2020) argues, this is far more efficient than performing OMD over occupancy measure spaces.

We also need to modify the Geometric Re-sampling procedure accordingly since Line 10 requires using the true transition. To do so, we combine the procedure with the idea of *upper occupancy measures* from Jin et al. (2020). Specifically, in each trial we sample $\pi_k'$ in the same way as $\pi_k$ but with a fresh perturbation, then find the optimistic transition within $\mathcal{P}_k$ that maximizes the probability of $\pi_k'$ visiting $(s_k^h, a_k^h)$ (which can be done efficiently using dynamic programming as shown by Jin et al. (2020)), and finally simulate $\pi_k'$ for $h$ steps following this optimistic transition.

Due to space limit, the full algorithm, Algorithm 3, is deferred to Appendix B.2. The analysis of the extra regret caused by the transition estimation error can be handled similarly to Jin et al. (2022) (more specifically, their Delayed HEDGE algorithm). As in previous works, this happens to be of order $\widetilde{\mathcal{O}}(H^2 S\sqrt{AK})$ and becomes the dominating term of the regret. This makes our final regret the same as the state-of-the-art (Jin et al., 2020), despite the weaker single-step stability lemma discussed in Section 3.1 (since this part is dominated now). Formally, we have the following regret guarantee.

**Theorem 5.** *For episodic AMDPs with bandit feedback and unknown transitions, Algorithm 3 with $\eta = 1/\sqrt{HSAK}$ and $L = \sqrt{SAK/H}$ ensures $\mathcal{R}_T = \widetilde{\mathcal{O}}\left(H^2 S\sqrt{AK}\right)$.*

# 4 FTPL for Episodic AMDPs with Delayed Feedback

In this section, we show how our FTPL algorithm and analysis can be easily extended to the delayed feedback setting where the losses for episode $k$ are only observed at the end of episode $k + d_k$.

The only change to the algorithm is to naturally delay the loss estimator construction until the loss feedback is received, and at each episode $k$ only use the estimators constructed so far, i.e., $\Omega_k \triangleq \{k' \mid k' + d_{k'} < k\}$, to compute the current policy $\pi_k$. See Algorithm 4 in Appendix C.

To show how the analysis works, we focus on the known transition case at this moment for simplicity. Similar to the non-delayed case, the key is to bound the stability term, which was $\mathbb{E}\left[\sum_{k=1}^{K} V(\pi_k; \widehat{\ell}_k) - \sum_{k=1}^{K} V(\pi_{k+1}; \widehat{\ell}_k)\right]$ in Section 3.1, but now becomes $\mathbb{E}\left[\sum_{k=1}^{K} V(\pi_k; \widehat{\ell}_k) - \sum_{k=1}^{K} V(\widetilde{\pi}_{k+1}; \widehat{\ell}_k)\right]$ where $\widetilde{\pi}_{k+1} = \arg\min_{\pi \in \Pi} V(\pi; \widehat{\ell}_{0:k})$ is a "cheating policy' (Gyorgy and Joulani, 2021; Jin et al., 2022) that uses all loss estimators from the first $k$ episodes (which matches $\pi_{k+1}$ for the non-delayed case). By the exact same analysis as Eq. (4) and Eq. (5), one can show

$$\mathbb{E}\left[V(\pi_k; \widehat{\ell}_k) - V(\widetilde{\pi}_{k+1}; \widehat{\ell}_k)\right] \leq \eta\, \mathbb{E}\left[\Big(\underbrace{\sum_{k' \in [k] \backslash \Omega_k} \sum_{h=1}^{H} \|\widehat{\ell}_{k'}^h\|_1}_{\text{DIFF}}\Big) \sum_{\pi \in \Pi} p_k(\pi) \left\langle \mu_\pi, \widehat{\ell}_k \right\rangle\right],$$

where the DIFF term is the cumulative $\ell_1$ norms of all the estimators used in computing $\widetilde{\pi}_{k+1}$ but not $\pi_k$ (again, a direct generalization of Eq. (5) where only $k$ satisfies such conditions for $k'$). It is then not hard to imagine that when summed over $k$, the DIFF term is eventually related to the total amount of delay $\mathfrak{D} = \sum_k d_k$. Indeed, the sum of all stability terms over $K$ episodes can be shown to be of order $\mathcal{O}(\eta H^2 SA(K + \mathfrak{D}))$. This is basically all the extra elements we need in the proof. More generally for unknown transitions, we prove the following guarantee (see Appendix C for the proof).

**Theorem 6.** *For episodic AMDPs with delayed bandit feedback and unknown transitions, Algorithm 4 with $\eta = 1/\sqrt{HSA(K+\mathfrak{D})}$ and $L = \sqrt{HSA/H}$ ensures $\mathcal{R}_T = \widetilde{\mathcal{O}}\left(H^2 S\sqrt{AK} + H^{3/2}\sqrt{SA\mathfrak{D}}\right)$.*

The simplicity of our analysis is similar to the Delayed HEDGE algorithm (Jin et al., 2022), but the latter is inefficient with time complexity $\Omega(A^S)$. The efficient Delayed UOB-FTRL algorithm (Jin et al., 2022) requires a more complicated analysis and only achieves $\widetilde{\mathcal{O}}\left(H^2 S\sqrt{AK} + H^{3/2} SA\sqrt{\mathfrak{D}}\right)$ regret (which is worse than ours), while its improved variant Delayed UOB-REPS with a new *delay-adapted* estimator achieves the current best bound $\widetilde{\mathcal{O}}(H^2 S\sqrt{AK} + H^{5/4}(SA)^{1/4}\sqrt{\mathfrak{D}})$. However, it is unclear to us whether such delay-adapted estimators can help improve FTPL. Finally, we again remark that the current best lower bound is $\Omega(H^{3/2}\sqrt{SAK} + H\sqrt{\mathfrak{D}})$ (Lancewicki et al., 2022).

## 5 FTPL for Infinite-Horizon AMDPs

At last, we discuss how FTPL can be used to derive the first no-regret algorithm for infinite-horizon communicating AMDPs with bandit feedback and (known) stochastic transition. Note that learning infinite-horizon AMDPs is much more difficult due to the lack of resets (in a sense, this is like a finite-horizon problem but with only one long episode with $T$ steps). Another way to see the difficulty is that the benchmark in the regret definition Eq. (1) is evaluated on states generated by following $\pi^*$ repeatedly for $T$ rounds, without any resets. From a technical viewpoint, this requires the algorithm to also make sure that, when following a policy $\pi$, its suffered loss is indeed close to the total loss if $\pi$ has been followed since the very beginning, which is unnatural without ergodic assumptions.

Chandrasekaran and Tewari (2021) resolve this issue by the combination of two ideas. First, under the mild Assumption 2, they show that whenever the agent wants to switch the current policy to another policy $\pi$, there exists a procedure to make sure that after $\mathcal{O}(D^2)$ steps of a transition phase, the agent's state distribution is exactly the same as that induced by following $\pi$ from the very beginning. That is, after this switching procedure, the agent can "pretend" that she has followed $\pi$ all the time. Second, since this procedure requires a cost of $\mathcal{O}(D^2)$ steps (where the loss of the agent can be arbitrarily bad and only trivially bounded by $\mathcal{O}(D^2)$), the algorithm needs to switch its policy infrequently.

Our algorithm follows the same ideas. However, while low-switching is relatively easy to ensure in the full-information case without paying extra regret, it is known that with bandit feedback there is an unavoidable trade-off between the number of switches and the regret, which can be optimally balanced via a simple epoching scheme (Dekel et al., 2014). To this end, we divide the total $T$ steps into $J = o(T)$ epochs, each with length $H = T/J = \omega(D^2)$. At the beginning of the $j$-th epoch, we compute a new policy $\pi_j$, apply the switching procedure of Chandrasekaran and Tewari (2021) to adjust the state distribution (see Algorithm 5), and finally follow the same policy $\pi_j$ for the rest of the epoch. This clearly only introduces $J$ switches, which contributes to at most $\mathcal{O}(JD^2)$ extra regret.

It remains to specify how to find $\pi_j$ in epoch $j$ using FTPL. The key difference compared to the episodic case is that, due to the lack of resets, we need to add perturbation to *every* time step instead of just to each of the $H$ steps of an episode. We then still play the policy that minimizes the cumulative estimated losses plus all the perturbed losses. Formally, $\pi_j$ is defined as:

$$\pi_j = \operatorname*{argmin}_{\pi \in \Pi} \mathbb{E}\left[ \sum_{t=1}^{(j-1)T/J} \widehat{\ell}^t(s^t, \pi(s^t)) + \sum_{t=1}^{T} z^t(s^t, \pi(s^t)) \middle| s^{t+1} \sim \mathbb{P}(\cdot \mid s^t, \pi(s^t)), \ \forall t \right], \quad (6)$$

where $\{z^t : \mathcal{S} \times \mathcal{A} \to \mathbb{R}\}_{t \in [T]}$ is such that each $z^t(s, a)$ is an independent sample of Laplace$(\eta)$, and each $\widehat{\ell}^t$ is the estimator of $\ell^t$ constructed from the Geometric Re-sampling procedure.

Unfortunately, as far as we know, there is in fact no existing polynomial time algorithm for solving Eq. (6) (the difficulty comes from the restriction on *stationary* policies whose behavior does not vary over time). Even if the losses are stochastic, the problem is only known to be P-hard (Papadimitriou and Tsitsiklis, 1987; Mundhenk et al., 2000) and no polynomial algorithm has been developed.

However, note that this optimization is exactly in the same form as the benchmark in the regret definition Eq. (1). Following many prior works such as Dudík et al. (2020); Block et al. (2022); Haghtalab et al. (2022), we thus assume access to a planning oracle that solves this offline problem, making our algorithm only oracle-efficient instead of truly polynomial-time-efficient. Note that even given this oracle, the algorithm of Chandrasekaran and Tewari (2021) is inefficient since it creates independent perturbation for *each* of the $A^S$ policies, while our perturbation is much more compact.

In terms of the analysis, the key extra challenge is caused by having $T$ perturbed losses. Indeed, the same analysis from the episodic case (Lemma 44) would lead to a term of order $\widetilde{\mathcal{O}}(T/\eta)$, which is prohibitively large. Instead, inspired by Syrgkanis et al. (2016), we provide a different analysis showing that this can be improved to $\widetilde{\mathcal{O}}(S\sqrt{AT}/\eta)$, which has worse dependencies on $S$ and $A$ but better dependency on $T$, the key to ensure sub-linear regret eventually. To conclude, our FTPL algorithm achieves the following guarantee (see Appendix D.1 for the full algorithm and analysis).

**Theorem 7.** *For infinite-horizon AMDPs with bandit feedback and known transitions, Algorithm 6 with $\eta = \frac{S^{1/3}}{D^{2/3}T^{1/3}}$, $J = \frac{S^{2/3}A^{1/2}T^{5/6}}{D^{4/3}}$ and $L = \frac{S^{1/3}A^{1/2}T^{1/6}}{D^{2/3}}$ ensures $\mathcal{R}_T = \widetilde{\mathcal{O}}\left(A^{1/2}(SD)^{2/3}T^{5/6}\right)$.*

We emphasize again that this is the first (oracle-efficient) algorithm for this setting. Even in the easier full-information setting (where $\ell^t$ is fully revealed at the end of time $t$), our algorithm also has its computational advantages compared to that of Chandrasekaran and Tewari (2021), since, as mentioned, their algorithm requires $\Omega(A^S)$ complexity (albeit with a better regret bound $\widetilde{\mathcal{O}}(D^2\sqrt{ST})$).

The best lower bound for this setting is $\Omega(S^{1/3}T^{2/3})$ (Dekel et al., 2014). Dekel and Hazan (2013) achieve $\widetilde{\mathcal{O}}(S^3AT^{2/3})$ but only when the transition is deterministic. For completeness, we provide a HEDGE-based *inefficient* algorithm (Appendix D.2) for general stochastic transitions, which achieves the optimal regret in terms of the dependence on $T$, improving our oracle-efficient FTPL algorithm.

**Theorem 8.** *For infinite-horizon AMDPs with bandit feedback and known transitions, Algorithm 7 with $\eta = \frac{S^{1/3}}{A^{1/3}(DT)^{2/3}}$ and $J = \frac{(ST)^{2/3}A^{1/3}}{D^{4/3}}$ ensures $\mathcal{R}_T = \widetilde{\mathcal{O}}\left(A^{1/3}(SDT)^{2/3}\right)$.*

## 6 Conclusion

In this paper, we designed FTPL-based algorithms for adversarial MDPs with bandit feedback in various settings, including episodic settings, delayed feedback settings and infinite-horizon settings. Our algorithms are easy to implement as they only require solving the offline planing problem, and in some cases they match the state-of-the-art performance or are even the first ever no-regret algorithms.

One interesting open question is whether, despite our counterexample, Eq. (3) can still hold with a larger constant for the right-hand side, either with our current algorithm or via some modified versions (for example with a different kind of perturbation). Achieving this would lead to an improved version of Lemma 3 and thus give the near-optimal delay-related regret term $\widetilde{\mathcal{O}}(H^{3/2}\sqrt{\mathfrak{D}})$ for the delayed feedback setting, which is not currently achieved by any existing algorithms.

An alternative direction is to try to equip our Algorithm 4 (for episodic AMDPs with delayed feedback) with the "delay-adapted" loss estimators proposed by Jin et al. (2022). As their analysis

heavily relies on the exponential weight scheme (see their Lemma D.7, which bounds KL divergences between consecutive policies), it is unclear to us whether FTPL enjoys a similar property.

Another important future direction is to improve our results in the infinite-horizon setting, such as improving the $\widetilde{\mathcal{O}}(T^{5/6})$ oracle-efficient regret upper bound, removing the usage of oracles, or dealing with the unknown transition case (which has not yet been studied at all).

There are also several possible generalizations of our setting. For example, we only assume the losses to be adversarial. Further incorporating evolving transition is an important next step. There is already an FTPL-based algorithm (Yu and Mannor, 2009) for evolving dynamics (though they are assuming ergodic infinite-horizon MDPs), which builds upon the FTPL analysis by Even-Dar et al. (2009) (see their Lemma III.3). Although our work directly improves the performance guarantee of Even-Dar et al. (2009), it is highly unclear whether we can adopt the algorithm of Yu and Mannor (2009) for unknown-transition episodic MDPs (they assumed the transitions to be revealed after each episode) or infinite-horizon weakly communicating MDPs. Solving either case will be interesting. Moreover, considering dynamic regret instead of static regret can also be challenging.

## Acknowledgments and Disclosure of Funding

We greatly acknowledge Vasilis Syrgkanis for the helpful discussion about whether their single-step stability lemma (Syrgkanis et al., 2016, Lemma 10) holds for non-binary action spaces. We also thank the anonymous reviewers for their insightful comments, which we greatly benefit from. HL is supported by NSF Award IIS-1943607 and a Google Faculty Research Award.

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
