# Supplementary Materials

## A  Notations

We summarize our notations used in the appendix below:

- For a policy $\pi \in \Pi$ and a transition $P\colon [H] \times \mathcal{S} \times \mathcal{A} \to \triangle(\mathcal{S})$, the occupancy measure of $\pi$ at the $h$-th step ($h \in [H]$) is defined as

$$\mu_\pi^h(s,a; P) = \Pr\{(s^h, a^h) = (s,a) \mid a^h = \pi^h(s^h), s^{h+1} \sim P^h(\cdot \mid s^h, a^h), s^1\}.$$

  We will use $\mu_\pi^h(P)$ to denote the vector $\{\mu_\pi^h(s,a;P)\}_{(s,a)\in\mathcal{S}\times\mathcal{A}}$. Specifically, if $P$ is the true transition $\mathbb{P}$, we will abbreviate it as $\mu_\pi^h$ for simplicity.

- With a slight abuse of notation, for infinite-horizon AMDPs, we will also use the same notation $\mu_\pi^t \in \triangle(\mathcal{S} \times \mathcal{A})$ ($t \in [T]$) to refer to the occupancy measure of $\pi$ at time slot $t$, starting from the first state $s^1$ and following the transition $\mathbb{P}$ (as we do not consider unknown transition cases for infinite-horizon AMDPs, we will always abbreviate the transitions).

- For a policy $\pi \in \Pi$, a transition $P\colon [H] \times \mathcal{S} \times \mathcal{A} \to \triangle(\mathcal{S})$ and a loss function $\widehat{\ell}\colon [H] \times \mathcal{S} \times \mathcal{A} \to \mathbb{R}_{\geq 0}$, the value function is defined as

$$V(\pi; \widehat{\ell}, P) = \mathbb{E}\left[\widehat{\ell}^h(s^h, a^h) \Big| a^h = \pi(s^h), s^{h+1} \sim P^h(s^h, a^h), s^1\right] = \sum_{h=1}^{H} \langle \mu_\pi^h(P), \widehat{\ell}^h \rangle.$$

- A perturbation $z\colon \mathbb{R}^{[H]\times\mathcal{S}\times\mathcal{A}}$ is a fresh sample such that

$$z^h(s,a) \sim \mathrm{Laplace}(\eta) \text{ and each entry is independently sampled.}$$

  For simplicity in notations, we use $\widehat{\ell}_0$ as an alias of $z$.

- For a sequence of loss functions $\widehat{\ell}_1, \widehat{\ell}_2, \ldots, \widehat{\ell}_k$, we use $\widehat{\ell}_{1:k}$ to denote $\sum_{k'=1}^{k} \widehat{\ell}_{k'}$.

**Algorithm 2** FTPL for Episodic AMDPs with Bandit Feedback and Known Transition

---

**Require:** Laplace distribution parameter $\eta$. Geometric Re-sampling parameter $L$.
1: Sample perturbation $\widehat{\ell}_0 = z$ such that $z^h(s,a)$ is an independent sample of Laplace($\eta$).
2: **for** $k = 1, 2, \ldots, K$ **do**
3:     Calculate $\pi_k = \operatorname{argmin}_{\pi \in \Pi} V(\pi; \widehat{\ell}_{0:k-1})$ (via dynamic programming).
4:     **for** $h = 1, 2, \ldots, H$ **do**
5:         Observe $s_k^h$, play $a_k^h = \pi_k(s_k^h)$, suffer and observe loss $\ell_k^h(s_k^h, a_k^h)$.
6:         Calculate loss estimator $\widehat{\ell}_k^h$ via Geometric Re-sampling (Neu and Bartók, 2016):
7:         **for** $M_k^h = 1, 2, \ldots, L$ **do**
8:             Sample a fresh perturbation $\widetilde{z}$ in the same way as $z$.
9:             Calculate $\pi_k' = \operatorname{argmin}_{\pi \in \Pi} V(\pi; \widehat{\ell}_{1:k-1} + \widetilde{z})$.
10:            Simulate $\pi_k'$ for $h$ steps starting from $s^1$ and following transitions $\mathbb{P}^1, \ldots, \mathbb{P}^h$.
11:            **if** $(s_k^h, a_k^h)$ is visited at step $h$ or $M_k^h = L$ **then**
12:               Set $\widehat{\ell}_k^h(s,a) = M_k^h \cdot \ell_k^h(s_k^h, a_k^h) \cdot \mathbb{1}[(s_k^h, a_k^h) = (s,a)]$ and break.

---

## B    Analysis of Episodic AMDP Algorithms

### B.1    Known Transition Case (Theorem 4)

For convenience, we restate the algorithm for episodic AMDPs with bandit feedback and known transitions in Algorithm 2. As shown by Syrgkanis et al. (2016, Appendix A.2), for an oblivious adversary (which is our case), it suffices to draw the perturbations once at the beginning of the interaction (i.e., the perturbation $z$ is fixed throughout the game).

Then we give the proof of Theorem 4. As sketched in the main text, we define the following probability, as-if we are resampling a purturbation $z$ for each round:

$$p_k(\pi) = \Pr_z\{\pi_k = \pi \mid \widehat{\ell}_1, \widehat{\ell}_2, \ldots, \widehat{\ell}_{k-1}\}.$$

Note that, as mentioned in (Syrgkanis et al., 2016, Appendix A.2), $p_k$ is just the probability of picking $\pi$ at episode $k$ given all history from episodes $1, 2, \ldots, k-1$. Now, we decompose our regret $\mathcal{R}_K$ into the following three terms:

$$\mathcal{R}_K = \mathbb{E}\left[\sum_{k=1}^{K}\sum_{h=1}^{H}\sum_{\pi \in \Pi} p_k(\pi)\langle \mu_\pi^h, \ell_k^h \rangle - \sum_{k=1}^{K}\sum_{h=1}^{H}\langle \mu_{\pi^*}^h, \ell_k^h \rangle\right]$$

$$= \mathbb{E}\Bigg[\underbrace{\sum_{k=1}^{K}\sum_{h=1}^{H}\langle \mu_{\pi_k}^h, \ell_k^h - \widehat{\ell}_k^h \rangle + \sum_{k=1}^{K}\sum_{h=1}^{H}\langle \mu_{\pi^*}^h, \widehat{\ell}_k^h - \ell_k^h \rangle}_{\text{GR error term}}\Bigg]$$

$$\underbrace{\mathbb{E}\left[\sum_{k=1}^{K}\sum_{h=1}^{H}\sum_{\pi \in \Pi} p_{k+1}(\pi)\langle \mu_\pi^h, \widehat{\ell}_k^h \rangle - \sum_{k=1}^{K}\sum_{h=1}^{H}\langle \mu_{\pi^*}^h, \widehat{\ell}_k^h \rangle\right]}_{\text{Error term}} +$$

$$\underbrace{\mathbb{E}\left[\sum_{k=1}^{K}\sum_{h=1}^{H}\sum_{\pi \in \Pi} (p_k(\pi) - p_{k+1}(\pi))\langle \mu_\pi^h, \widehat{\ell}_k^h \rangle\right]}_{\text{Stability term}}.$$

#### B.1.1    Bouding the GR Error Term

**Lemma 9** (Bounding GR Error Term). *The GR error term is bounded by*

$$\mathbb{E}\left[\sum_{k=1}^{K}\sum_{h=1}^{H}\langle \mu_{\pi_k}^h, \ell_k^h - \widehat{\ell}_k^h \rangle + \sum_{k=1}^{K}\sum_{h=1}^{H}\langle \mu_{\pi^*}^h, \widehat{\ell}_k^h - \ell_k^h \rangle\right] \leq \frac{SAHK}{eL}.$$

*Proof.* First notice that, from Lemma 38, $\mathbb{E}[\widehat{\ell}_k^h(s,a) \mid \mathcal{F}_{k-1}] \leq \ell_k^h(s,a)$. Moreover, as $\pi^*$ is deterministic (i.e., it does not depend on the randomness from the algorithm), the second term

$$\mathbb{E}\left[\sum_{k=1}^K \sum_{h=1}^H \langle \mu_{\pi^*}^h, \widehat{\ell}_k^h - \ell_k^h \rangle \right] = \mathbb{E}\left[\sum_{k=1}^K \sum_{h=1}^H \langle \mu_{\pi^*}^h, \mathbb{E}[\widehat{\ell}_k^h \mid \mathcal{F}_{k-1}] - \ell_k^h \rangle \right] \leq 0.$$

For the first term, again by Lemma 38, we have

$$\mathbb{E}\left[\sum_{k=1}^K \sum_{h=1}^H \langle \mu_{\pi_k}^h, \ell_k^h - \widehat{\ell}_k^h \rangle \right] = \sum_{k=1}^K \sum_{h=1}^H \sum_{(s,a)\in\mathcal{S}\times\mathcal{A}} \mathbb{E}\left[ \mu_{\pi_k}^h(s,a) \cdot (1 - q_k^h(s,a))^L \ell_k^h(s,a) \right],$$

where $q_k^h(s,a)$ is the probability of visiting $(s,a)$ in a single trial of the Geometric Re-sampling process, which is just (note that $q_k^h$ itself is also a random variable as $p_k$ is non-deterministic)

$$q_k^h(s,a) = \mathbb{E}[\mu_{\pi_k}^h(s,a)] = \sum_{\pi\in\Pi} p_k(\pi) \mu_\pi^h(s,a)$$

in our case. By noticing that $q(1-q)^L \leq q e^{-Lq} \leq \frac{1}{eL}$ for all $q \geq 0$ (Neu and Bartók, 2016), we have

$$\mathbb{E}\left[\sum_{k=1}^K \sum_{h=1}^H \langle \mu_{\pi_k}^h, \ell_k^h - \widehat{\ell}_k^h \rangle \right] \leq HKSA \frac{1}{eL} = \frac{SAHK}{eL},$$

as claimed. □

### B.1.2 Bounding the Error Term

**Lemma 10** (Bounding Error Term). *The error term is bounded by*

$$\mathbb{E}\left[\sum_{k=1}^K \sum_{h=1}^H \sum_{\pi\in\Pi} p_{k+1}(\pi) \langle \mu_\pi^h, \widehat{\ell}_k^h \rangle \right] - \sum_{k=1}^K \sum_{h=1}^H \langle \mu_{\pi^*}^h, \widehat{\ell}_k^h \rangle \leq \frac{2H}{\eta}(1 + \ln(SA)).$$

*Proof.* The proof uses the standard "be-the-leader" technique. For simplicity, we rewrite the error term as

$$\mathbb{E}\left[\sum_{k=1}^K V(\pi_{k+1}; \widehat{\ell}_k, \mathbb{P}) - \sum_{k=1}^K V(\pi^*; \widehat{\ell}_k, \mathbb{P}) \right].$$

Now consider the summation inside the expectation. If we add an extra term $V(\pi_1; \widehat{\ell}_0, \mathbb{P}) - V(\pi^*; \widehat{\ell}_0, \mathbb{P})$ where $\widehat{\ell}_0 = z$ is the perturbation, we will have

$$\sum_{k=0}^K V(\pi_{k+1}; \widehat{\ell}_k, \mathbb{P}) - V(\pi^*; \widehat{\ell}_{0:K}, \mathbb{P})$$

$$\overset{(a)}{\leq} \sum_{k=0}^K V(\pi_{k+1}; \widehat{\ell}_k, \mathbb{P}) - V(\pi_{K+1}; \widehat{\ell}_{0:K}, \mathbb{P}) = \sum_{k=0}^{K-1} V(\pi_{k+1}; \widehat{\ell}_k, \mathbb{P}) - V(\pi_{K+1}; \widehat{\ell}_{0:K-1}, \mathbb{P})$$

$$\overset{(b)}{\leq} \sum_{k=0}^{K-1} V(\pi_{k+1}; \widehat{\ell}_k, \mathbb{P}) - V(\pi_K; \widehat{\ell}_{0:K-1}, \mathbb{P}) = \sum_{k=0}^{K-2} V(\pi_{k+1}; \widehat{\ell}_k, \mathbb{P}) - V(\pi_K; \widehat{\ell}_{0:K-2}, \mathbb{P})$$

$$\leq \cdots \leq V(\pi_1; \widehat{\ell}_0, \mathbb{P}) - V(\pi_2; \widehat{\ell}_0, \mathbb{P}) \overset{(c)}{\leq} 0,$$

where (a) used the optimality of $\pi_{K+1}$ w.r.t. $\widehat{\ell}_{0:K}$, (b) used the optimality of $\pi_K$ w.r.t. $\widehat{\ell}_{0:K-1}$ and so on, until the last step (c) where the optimality of $\pi_1$ w.r.t. $\widehat{\ell}_0$ is used. So we have

$$\mathbb{E}\left[\sum_{k=1}^K V(\pi_{k+1}; \widehat{\ell}_k, \mathbb{P}) - \sum_{k=1}^K V(\pi^*; \widehat{\ell}_k, \mathbb{P}) \right] \leq \mathbb{E}[V(\pi^*; \widehat{\ell}_0, \mathbb{P}) - V(\pi_1; \widehat{\ell}_0, \mathbb{P})].$$

By the notation of occupancy measures, we can rewrite it as

$$\mathbb{E}\left[\sum_{h=1}^{H}\langle\mu_{\pi^*}^h,\widehat{\ell}_0^h\rangle-\sum_{h=1}^{H}\langle\mu_{\pi_1}^h,\widehat{\ell}_0^h\rangle\right]\leq 2\sum_{h=1}^{H}\mathbb{E}[\|\widehat{\ell}_0^h\|_\infty].$$

Recall that $\ell_0^h(s,a)\sim\text{Laplace}(\eta)$, so we have

$$\mathbb{E}[\|\widehat{\ell}_0^h\|_\infty]=\mathbb{E}\left[\max_{s,a}|\ell_0^h(s,a)|\right]\leq\frac{1+\ln(SA)}{\eta},$$

where the last step is due to the fact that $|\ell_0^h(s,a)|$ is an exponential distribution and Lemma 44. $\quad\square$

### B.1.3 Bounding the Stability Term

For the stability term, we first prove the following "single-step stability" lemma that we stated without proof in the main body.

**Lemma 11** (Single-Step Stability). *For all $k\in[K]$ and $(s,a)\in\mathcal{S}\times\mathcal{A}$,*

$$p_{k+1}(\pi)\geq p_k(\pi)\exp\left(-\eta\sum_{h=1}^{H}\|\widehat{\ell}_k^h\|_1\right),\quad\forall\pi\in\Pi.$$

*Proof.* For simplicity, we use $\pi=\text{best}(\ell)$ to denote $\pi=\text{argmin}_{\pi\in\Pi}V(\pi;\ell,\mathbb{P})$. Then we have

$$\begin{aligned}p_k(\pi)&=\int_z\mathbb{1}\left[\pi=\text{best}\left(\widehat{\ell}_{1:k-1}+z\right)\right]f(z)\ \mathrm{d}z\\&=\int_z\mathbb{1}\left[\pi=\text{best}\left(\widehat{\ell}_{1:k-1}+\left(z+\widehat{\ell}_k\right)\right)\right]f\left(z+\widehat{\ell}_k\right)\ \mathrm{d}z\\&=\int_z\mathbb{1}\left[\pi=\text{best}\left(\widehat{\ell}_{1:k}+z\right)\right]f\left(z+\widehat{\ell}_k\right)\ \mathrm{d}z,\end{aligned}$$

where $f(z)$ is the probability density function of $z$ and the second step made use of the fact that $z+\widehat{\ell}_k$ is still linear in $z$. Moreover,

$$p_{k+1}(\pi)=\int_z\mathbb{1}\left[\pi=\text{best}\left(\widehat{\ell}_{1:k}+z\right)\right]f(z)\ \mathrm{d}z.$$

Recall that the definition of $f(z)$ is just $f(z)=\prod_{h=1}^{H}\sum_{s,a}\exp(-\eta|z^h(s,a)|)=\prod_{h=1}^{H}\exp(-\eta\|z^h\|_1)$ as each entry of $z$ is i.i.d. We thus have

$$f\left(z+\widehat{\ell}_k\right)=\prod_{h=1}^{H}\exp\left(-\eta\left(\|z^h+\widehat{\ell}_k\|_1-\|z^h\|_1\right)\right)f(z),$$

which gives

$$\frac{f\left(z+\widehat{\ell}_k\right)}{f(z)}\in\left[\exp\left(-\eta\sum_{h=1}^{H}\|\widehat{\ell}_k^h\|_1\right),\exp\left(\eta\sum_{h=1}^{H}\|\widehat{\ell}_k^h\|_1\right)\right]$$

by triangle inequality. Therefore, $p_{k+1}(\pi)/p_k(\pi)$ lies in this interval as well, which is just our claim. $\quad\square$

**Lemma 12** (Bounding Stability Term). *The stability term is bounded by*

$$\mathbb{E}\left[\sum_{k=1}^{K}\sum_{h=1}^{H}\sum_{\pi\in\Pi}(p_k(\pi)-p_{k+1}(\pi))\langle\mu_\pi^h,\widehat{\ell}_k^h\rangle\right]\leq 3\eta H^2 SAK.$$

*Proof.* By summing up Lemma 11 for all $\pi \in \Pi$ and using the fact that $1 - \exp(-x) \le x$, we have

$$\sum_{\pi \in \Pi} (p_k(\pi) - p_{k+1}(\pi)) \sum_{h=1}^{H} \langle \mu_\pi^h, \widehat{\ell}_k^h \rangle \le \eta \sum_{h'=1}^{H} \|\widehat{\ell}_k^h\|_1 \cdot \sum_{\pi \in \Pi} p_k(\pi) \sum_{h=1}^{H} \langle \mu_\pi^h, \widehat{\ell}_k^h \rangle, \quad \forall k \in [K]. \quad (7)$$

To proceed, we need to investigate the Geometric Re-sampling process. Consider the random variable $M_k^h$ whose value is determined in the last line of Algorithm 1. One may view it as a "truncated" geometric random variable, where $\mathrm{Geo}(q)$ is a geometric random variable with parameter $q$, i.e., $\Pr\{\mathrm{Geo}(q) = n\} = (1-q)^{n-1}q$. Formally, we have:

$$M_k^h = \min\{\mathrm{Geo}(q_k^h(s_k^h, a_k^h)), L\}, \quad \text{where } q_k^h(s,a) = \mathop{\mathbb{E}}_{\pi \sim p_k}[\mu_\pi^h(s,a)] = \sum_{\pi \in \Pi} p_k(\pi)\mu_\pi^h(s,a). \quad (8)$$

So if we calculate the expectation of $\widehat{\ell}_k^h(s,a)$ only with respect to $M_k^h$, we will have

$$\mathbb{E}\left[\widehat{\ell}_k^h(s,a)\Big|(s_k^h, a_k^h) = (s,a)\right] \le \frac{\ell_k^h(s,a)}{q_k^h(s,a)}.$$

Let $\mathbb{1}_k^h(s,a)$ be the shorthand notation of $\mathbb{1}[(s_k^h, a_k^h) = (s,a)]$. Then for those $h' \ne h$ in the RHS of Eq. (7), we have

$$\eta\,\mathbb{E}\left[\sum_{h=1}^{H} \sum_{s,a} \sum_{\pi \in \Pi} p_k(\pi)\mu_\pi^h(s,a)\widehat{\ell}_k^h(s,a) \sum_{h' \ne h} \|\widehat{\ell}_k^{h'}\|_1 \,\middle|\, \mathcal{F}_{k-1}\right]$$

$$\overset{(a)}{\le} \eta\,\mathbb{E}\left[\sum_{h=1}^{H}\sum_{s,a} \mathbb{1}_k^h(s,a)\ell_k^h(s,a)\frac{\sum_{\pi \in \Pi} p_k(\pi)\mu_\pi^h(s,a)}{q_k^h(s,a)} \sum_{h' \ne h}\|\widehat{\ell}_k^{h'}\|_1 \,\middle|\, \mathcal{F}_{k-1}\right]$$

$$\overset{(b)}{\le} \eta H\,\mathbb{E}\left[\sum_{h' \ne h}\|\widehat{\ell}_k^{h'}\|_1 \,\middle|\, \mathcal{F}_{k-1}\right] \overset{(c)}{\le} \eta H^2 SA.$$

where (a) is taking expectation w.r.t. $M_k^h$, (b) used the definition of $q_k^h$ together with the fact that $\sum_{(s,a)} \mathbb{1}_k^h(s,a) = 1$, and (c) used the fact that $\mathbb{E}[\widehat{\ell}_k^{h'}(s',a') \mid \mathcal{F}_{k-1}] \le \ell_k^{h'}(s',a') \le 1$ (Lemma 38).

For those terms with $h = h'$ in Eq. (7), by direct calculation and the fact that $\widehat{\ell}_k^h$ is a one-hot vector, we can write them as

$$\eta\,\mathbb{E}\left[\sum_{h=1}^{H}\sum_{s,a}\sum_{\pi \in \Pi} p_k(\pi)\mu_\pi^h(s,a)\left(\widehat{\ell}_k^h(s,a)\right)^2 \,\middle|\, \mathcal{F}_{k-1}\right] \le 2\eta\,\mathbb{E}\left[\sum_{h,s,a}\frac{q_k^h(s,a)}{q_k^h(s,a)} \,\middle|\, \mathcal{F}_{k-1}\right] \le 2\eta HSA$$

where we use $\mathbb{E}[(\widehat{\ell}_k^h(s,a))^2 \mid \mathcal{F}_{k-1}] \le 2(q_k^h(s,a))^{-1}$ (Lemma 39). Combining the terms with $h' \ne h$ and the ones with $h' = h$ gives our conclusion. $\quad\square$

### B.1.4 Proof of Theorem 4

*Proof of Theorem 4.* From Lemmas 9, 10 and 12, we have

$$\mathcal{R}_K \le \frac{SAHK}{eL} + \frac{2H}{\eta}(1 + \ln(SA)) + 3\eta H^2 SAK.$$

Therefore, if we pick $\eta^{-1} = \sqrt{HSAK}$ and $L = \sqrt{SAK/H}$,

$$\mathcal{R}_K \le \frac{H^{3/2}\sqrt{SAK}}{e} + 2H^{3/2}\sqrt{SAK}(1 + \ln(SA)) + 3H^{3/2}\sqrt{SAK} = \widetilde{\mathcal{O}}\left(H^{3/2}\sqrt{SAK}\right),$$

as desired. $\quad\square$

---

**Algorithm 3** FTPL for Episodic AMDPs with Bandit Feedback and Unknown Transition

---
**Require:** Laplace distribution parameter $\eta$. Geometric Re-sampling parameter $L$.
1: Initialize $\mathcal{P}_1 \leftarrow (\triangle(\mathcal{S}))^{[H] \times \mathcal{S} \times \mathcal{A}}$ (the set of all possible transition functions).
2: Sample perturbation $\widehat{\ell}_0 = z$ such that $z^h(s, a)$ is an independent sample of Laplace($\eta$).
3: **for** $k = 1, 2, \ldots, K$ **do**
4:     Let $(\pi_k, P_k) = \operatorname{argmin}_{(\pi, P) \in \Pi \times \mathcal{P}_k} V(\pi; \widehat{\ell}_{1:k-1} + z, P)$ by Extended Value Iteration (Jaksch et al., 2010). (See also Remark 16 for more details.)
5:     **for** $h = 1, 2, \ldots, H$ **do**
6:         Observe $s_k^h$, play $a_k^h = \pi_k(s_k^h)$, suffer and observe loss $\ell_k^h(s_k^h, a_k^h)$.
7:         **for** $M_k^h = 1, 2, \ldots, L$ **do**
8:             Sample a fresh perturbation $\widetilde{z}$ in the same way as $z$.
9:             Calculate $(\pi_k', P_k') = \operatorname{argmin}_{(\pi, P) \in \Pi \times \mathcal{P}_k} V(\pi; \widehat{\ell}_{1:k-1} + \widetilde{z}, P)$.
10:             Pick the transition $\widehat{P}_k' \in \mathcal{P}_k$ such that $\mu_\pi^h(s_k^h, a_k^h; \widehat{P}_k')$ is maximized via the COMP-UOB procedure proposed by Jin et al. (2020).
11:             Simulate $\pi_k'$ for $h$ steps starting from $s^1$ and following transitions $(\widehat{P}_k')^1, \ldots, (\widehat{P}_k')^h$.
12:             **if** $(s_k^h, a_k^h)$ is visited at step $h$ or $M_k^h = L$ **then**
13:                 Set $\widehat{\ell}_k^h(s, a) = M_k^h \cdot \ell_k^h(s_k^h, a_k^h) \cdot \mathbb{1}[(s_k^h, a_k^h) = (s, a)]$ and break.
14:     Calculate $\mathcal{P}_{k+1}$ according to Eq. (10).

---

### B.1.5 Comparism with the CONTEXT-FTPL algorithm

One may think that our algorithm together with its analysis looks quite similar to the CONTEXT-FTPL algorithm (Syrgkanis et al., 2016, Algorithm 2) for adversarial contextual bandits. In fact, we can even convert the episodic AMDP problem with known transition as an instance of their contextual semi-bandit problem: for time slot $(k, h)$, the "context" is $h$ and the loss vector is $\widehat{\ell}_k^h$. A policy $\pi$ under context $x = h$ will then give an "action" $\pi(h) = \mu_\pi^h$ (the occupancy measure), which means it will suffer loss $\langle \mu_\pi^h, \widehat{\ell}_k^h \rangle$. Both algorithms add perturbations to each of the contexts, $1, 2, \ldots, H$, denoted by $z^1, z^2, \ldots, z^H \in \mathbb{R}^{SA}$ respectively.

However, there is a main differences between our setting and theirs: in their setting, the action space (where $\pi(x)$ belongs) is binary. However, in our case, $\mu_\pi^h \in [0, 1]^{SA}$ is continuous. Though this difference may look tiny, it actually induces extra difficulties: this subtle difference will make their Lemma 10, stated as follows, no longer hold.

**Lemma 13** (Syrgkanis et al. (2016, Lemma 10)). *For any contexts $x^1, x^2, \ldots, x^T$ and non-negative linear loss functions $\ell^1, \ell^2, \ldots, \ell^T$, suppose that $z^h(s, a) \sim$ Laplace($\eta$), CONTEXT-FTPL satisfies*

$$\mathbb{E}_z \left[ \langle \pi^t(x^t), \ell^t \rangle - \langle \pi^{t+1}(x^t), \ell^t \rangle \right] \leq \eta \cdot \mathbb{E}[\langle \pi^t(x^t), \ell^t \rangle^2], \quad \forall 1 \leq t < T. \tag{9}$$

To see this, consider the simple case that there is only one possible value of the context together with two policies, each associated with action vectors $(0.1, 0.1, 0.2)$ and $(0.2, 0.1, 0.1)$, denoted by $\pi_1$ and $\pi_2$, respectively. Set the cumulative (perturbed) loss vector $\ell_{0:t-1}$ as $(0.75, 0.2, 0.6)$ and $\ell_t = (0.1, 0, 0)$ (this is set to be one-hot, so it can be yielded from our Geometric Re-sampling process). Set the Laplace distribution parameter $\eta = 3$. Then, by direct calculation via integration, $p_t(\pi_1) = 0.609453$ and $p_{t+1}(\pi_1) = 0.675248$. As $\langle \pi_1, \ell^t \rangle = 0.01$ and $\langle \pi_2, \ell^t \rangle = 0.02$, the LHS of the Eq. (9) will be $0.00065795$ while the RHS will be $0.000651492$. Therefore, Eq. (9) simply *does not* hold, even if there are only 2 policies, 3 dimensions and 1 context.

Fortunately, as explained in the main text, though this strong version of "single-step stability lemma" does not hold, we are still able to prove a weaker version, Lemma 3 (which is restated as Lemma 11 in the appendix), to bound the stability term, which is worse only by a factor $H$, instead of $\|\widehat{\ell}_t\|_\infty \leq L$.

### B.2 Unknown Transition Case (Theorem 5)

We first present our algorithm for the unknown transion case in Algorithm 3.

### B.2.1 Transitions' Confidence Set Construction

We first discuss our construction of transitions' confidence sets. As in Jin et al. (2022), we maintain a confidence set of transitions $\mathcal{P}_k$ for each episode $k \in [K]$ as Eq. (10), where $\mathcal{P}_1 = (\triangle(\mathcal{S}))^{[H] \times \mathcal{S} \times \mathcal{A}}$.

As mentioned in the main text, we also want to ensure that $\mathcal{P}_{k+1} \subseteq \mathcal{P}_k$. Instead of taking $\mathcal{P}_1 \cap \mathcal{P}_2 \cap \cdots \mathcal{P}_k$ when doing the optimization, we directly ensure $\mathcal{P}_{k+1} \subseteq \mathcal{P}_k$ when constructing the confidence sets, such that they are always *shrinking*. This is to ensure a well-bounded error term, as we will illustrate in Lemma 18.

$$\mathcal{P}_{k+1} = \mathcal{P}_k \cap \left\{ \widehat{P} \colon [H] \times \mathcal{S} \times \mathcal{A} \to \triangle(\mathcal{S}) \middle| \left| \widehat{P}^h(s' \mid s, a) - \overline{P}_k^h(s' \mid s, a) \right| \le \varepsilon_i^h(s' \mid s, a), \forall s, s' \in \mathcal{S}, a \in \mathcal{A} \right\},$$

$$(10)$$

where $\varepsilon_k^h(s' \mid s, a) = 4 \sqrt{\dfrac{\overline{P}_k^h(s' \mid s, a) \ln(10 HSAK/\delta)}{\max\{1, N_k^h(s, a)\}}} + 10 \dfrac{\ln(10 HSAK/\delta)}{\max\{1, N_k^h(s, a)\}},$ $\quad (11)$

and $\overline{P}_k^h(s' \mid s, a) = \dfrac{N_k^h(s' \mid s, a)}{N_k^h(s, a)},$

$$N_k^h(s, a) = \sum_{k'=1}^k \mathbb{1}[(s_{k'}^h, a_{k'}^h) = (s, a)], N_k^h(s' \mid s, a) = \sum_{k'=1}^k \mathbb{1}[s_{k'}^{h+1} = s', (s_{k'}^h, a_{k'}^h) = (s, a)].$$

By the following lemma from Jin et al. (2020), we define $K$ good events, $\mathcal{E}_1, \mathcal{E}_2, \ldots, \mathcal{E}_K$, where $\mathcal{E}_k$ means $\mathbb{P} \in \mathcal{P}_k$. From the following lemma, we can conclude that $\Pr\{\mathcal{E}_1, \mathcal{E}_2, \ldots, \mathcal{E}_K\} \ge 1 - 4\delta$. For simplicity, we also denote $\mathcal{E} = \mathcal{E}_1 \wedge \mathcal{E}_2 \wedge \cdots \wedge \mathcal{E}_K$. Hence, $\Pr\{\mathcal{E}\} \ge 1 - 4\delta$ (in fact, we have $\mathcal{E} = \mathcal{E}_K$ as $\mathcal{P}_k \subseteq \mathcal{P}_{k-1}$).

**Lemma 14** ((Jin et al., 2020, Lemma 2)). *With probability $1 - 4\delta$, we have $\mathbb{P} \in \mathcal{P}_k$ for all $k \in [K]$.*

**Remark 15.** *Note that the original definition is slightly different from ours, where there is no intersection operations taken with previous confidence sets. However, as long as $\mathbb{P}$ belongs to all the confidence sets, it clearly belongs to the intersection of them.*

**Remark 16.** *Note that the Extended Value Iteration (Jaksch et al., 2010) approach works as long as $\mathcal{P}_k$ has the form $\{P \mid P^h(s' \mid s, a) \in [L^h(s' \mid s, a), R^h(s' \mid s, a)]\}$, but does not require $[L^h(s' \mid s, a), R^h(s' \mid s, a)]$ to be centered exactly at $\overline{P}_k^h(s' \mid s, a)$ (which is indeed the case for our algorithm due to the intersection operations).*

### B.2.2 Regret Decomposition

For the unknown-transition cases, we first do the following regret decomposition as Jin et al. (2020):

$$\mathcal{R}_K = \underbrace{\mathbb{E}\left[\sum_{k=1}^K (V(\pi_k; \ell_k, \mathbb{P}) - V(\pi_k; \ell_k, P_k))\right]}_{\text{ERROR}} + \underbrace{\mathbb{E}\left[\sum_{k=1}^K \left(V(\pi_k; \ell_k, P_k) - V(\pi_k; \widehat{\ell}_k, P_k)\right)\right]}_{\text{BIAS1}} +$$

$$\underbrace{\mathbb{E}\left[\sum_{k=1}^K \left(V(\pi_k; \widehat{\ell}_k, P_k) - V(\pi^*; \widehat{\ell}_k, \mathbb{P})\right)\right]}_{\text{ESTREG}} + \underbrace{\mathbb{E}\left[\sum_{k=1}^K \left(V(\pi^*; \widehat{\ell}_k, \mathbb{P}) - V(\pi^*; \ell_k, \mathbb{P})\right)\right]}_{\text{BIAS2}}.$$

Intuitively, the ERROR term is due to the transition estimation, BIAS1 and BIAS2 terms are due to loss estimation for $\pi_k$ and $\pi^*$, respectively, and ESTREG is the regret of our FTPL algorithm on the estimated transitions $P_k$ and the estimated losses $\widehat{\ell}_k$.

### B.2.3 Bounding the ESTREG Term

**Theorem 17** (Bounding ESTREG Term). *The ESTREG term is bounded by*

$$\text{ESTREG} = \mathbb{E}\left[\sum_{k=1}^K \left(V(\pi_k; \widehat{\ell}_k, P_k) - V(\pi^*; \widehat{\ell}_k, \mathbb{P})\right)\right] \le \frac{2H}{\eta}(1 + \ln(SA)) + 3\eta H^2 SAK + 8\delta KHL.$$

*Proof.* For the ESTREG term, we will also decompose it into an error term (not to be confused with the ERROR term which occurs in the decomposition of $\mathcal{R}_K$; this error term appears in the decomposition of ESTREG and is related to the 'be-the-leader' lemma) and a stability term (as it is defined for the estimated losses, there is no GR error term anymore). However, here we should define our "leader" as

$$(\widetilde{\pi}_{k+1}, \widetilde{P}_{k+1}) = \underset{(\pi, P) \in \Pi \times \mathcal{P}_k}{\operatorname{argmin}} V\left(\pi; \widehat{\ell}_{0:k}, P\right).$$

Instead of directly using $(\pi_{k+1}, P_{k+1})$ as the leader (as we did in the known transition case), we allow the transition $\widetilde{P}_{k+1}$ selected from $\mathcal{P}_k \supseteq \mathcal{P}_{k+1}$. This is critical to ensure a low stability term, as we can only derive the "single-step stability lemma" (Lemma 19 in this case) for two probability distributions sharing a same support ($\Pi \times \mathcal{P}_k$ here).

As an analog to the known transition case, we define $p_k(\pi, P)$ as the probability density function (with respect of the perturbation $z$) of $(\pi_k, P_k)$ conditioning on $\widehat{\ell}_1, \widehat{\ell}_2, \ldots, \widehat{\ell}_{k-1}$. Note that as there are infinitely many transitions, we cannot directly write $\operatorname{Pr}_z$ as in the known-transition setting.

Moreover, as explained before, we allow $\widetilde{P}_{k+1}$ to be picked from $\mathcal{P}_k$ instead of $\mathcal{P}_{k+1}$, so the probability of picking $(\widetilde{\pi}_{k+1}, \widetilde{P}_{k+1})$ as $(\pi, P)$ is not simply $p_{k+1}(\pi, P)$. Therefore, we have to define another notation representing the probability density of picking each $(\pi, P)$ as $(\widetilde{\pi}_{k+1}, \widetilde{P}_{k+1})$, namely $\widetilde{p}_{k+1}(\pi, P)$, which is the probability density of $(\widetilde{\pi}_{k+1}, \widetilde{P}_{k+1})$ with respect to $z$, conditioning on $\widehat{\ell}_1, \widehat{\ell}_2, \ldots, \widehat{\ell}_k$.

Hence, we can write

$$\text{ESTREG} = \mathbb{E}\left[\underbrace{\sum_{k=1}^{K} \sum_{h=1}^{H} \langle \mu_{\widetilde{\pi}_{k+1}}^h(\widetilde{P}_{k+1}) - \mu_{\pi^*}^h(\mathbb{P}), \widehat{\ell}_k^h \rangle}_{\text{Error term}}\right] +$$

$$\mathbb{E}\left[\underbrace{\sum_{k=1}^{K} \sum_{h=1}^{H} \sum_{\pi \in \Pi} \int_{\mathcal{P}_k} (p_k(\pi, P) - \widetilde{p}_{k+1}(\pi, P)) \langle \mu_\pi^h(P), \widehat{\ell}_k^h \rangle \, \mathrm{d}P}_{\text{Stability Term}}\right].$$

For the error term, we only need to verify that the "be-the-leader argument" that we used in Lemma 10 still holds. Fortunately, it turns out as long as $\mathcal{P}_1 \supseteq \mathcal{P}_2 \supseteq \cdots \supseteq \mathcal{P}_K \supseteq \{\mathbb{P}\}$, we can always conclude the following lemma, whose proof is presented later.

**Lemma 18** (Bounding Error Term). *The error term in this case is bounded by*

$$\mathbb{E}\left[\sum_{k=1}^{K} \sum_{h=1}^{H} \langle \mu_{\widetilde{\pi}_{k+1}}^h(\widetilde{P}_{k+1}) - \mu_{\pi^*}^h(\mathbb{P}), \widehat{\ell}_k^h \rangle\right] \leq \frac{2H}{\eta}(1 + \ln(SA)) + 4\delta KHL.$$

For the stability term, we need a similar but different single-step stability bound, as

**Lemma 19** (Single Step Stability). *For all $k \in [K]$, $(s, a) \in \mathcal{S} \times \mathcal{A}$ and $(\pi, P) \in \Pi \times \mathcal{P}_k$,*

$$\widetilde{p}_{k+1}(\pi, P) \geq p_k(\pi, P) \exp\left(-\eta \sum_{h=1}^{H} \|\widehat{\ell}_k^h\|_1\right).$$

With this lemma, our derivation for the stability term in known-transition cases (Lemma 12) also holds, except that we are using the upper occupancy measures in the Geometric Re-sampling process, instead of the actual occupancy measures. Technically, this means that the event $(s_k^h, a_k^h) = (s, a)$ will happen with a probability

$$\widehat{q}_k^h(s, a) = \operatorname{Pr}\{(s_k^h, a_k^h) = (s, a) \mid \mathcal{F}_{k-1}\} = \sum_{\pi \in \Pi} p_k(\pi) \mu_\pi^h(s, a; \mathbb{P}), \tag{12}$$

where $p_k(\pi) = \int_{\mathcal{P}_k} p_k(\pi, P)\,\mathrm{d}P$ is the marginal probability of picking $\pi$ for episode $k$ (with a slight abuse of notation). However, in each execution of the Geometric Re-sampling process, the probability of visiting $(s, a)$ is another probability

$$q_k^h(s,a) = \sum_{\pi \in \Pi} p_k(\pi) \max_{P' \in \mathcal{P}_k} \mu_\pi^h(s,a;P') \neq \widehat{q}_k^h(s,a), \tag{13}$$

which means we cannot use Lemmas 38 and 39 anymore.

Fortunately, we are able to derive Corollaries 40 and 41 in such a case, which actually implies the previous two lemmas, given that the actual occupancy measure $\widehat{q}_k^h(s,a)$ is bounded by the upper occupancy measure $q_k^h(s,a)$ (which is indeed this case as long as $\mathbb{P} \in \mathcal{P}_k$, i.e., event $\mathcal{E}_k$ holds). However, for the BIAS1 term (Theorem 22), as we will see later, this inconsistency will indeed induce extra difficulties, leading to a $\widetilde{\mathcal{O}}(H^2 S\sqrt{AK})$ dominating term as in Jin et al. (2020).

The detailed proof of Lemma 20 will be presented after the proof of this theorem.

**Lemma 20** (Bounding Stability Term). *The stability term in this case is bounded by*

$$\mathbb{E}\left[\sum_{k=1}^K \sum_{h=1}^H \sum_{\pi \in \Pi} \int_{\mathcal{P}_k} (p_k(\pi, P) - \widetilde{p}_{k+1}(\pi, P))\langle \mu_\pi^h(P), \widehat{\ell}_k^h \rangle\,\mathrm{d}P\right] \leq 3\eta H^2 SAK + 4\delta KHL.$$

Combining them together gives

$$\mathrm{ESTREG} \leq \frac{2H}{\eta}(1 + \ln(SA)) + 3\eta H^2 SAK + 8\delta KHL,$$

as claimed. $\qquad\square$

*Proof of Lemma 18.* The proof still follows the idea of Lemma 10. We rewrite the error term as

$$\mathbb{E}\left[\left(\sum_{k=1}^K V(\widetilde{\pi}_{k+1}; \widehat{\ell}_k, \widetilde{P}_{k+1}) - \sum_{k=1}^K V(\pi^*; \widehat{\ell}_k, \mathbb{P})\right)\mathbb{1}[\mathcal{E}]\right] +$$

$$\mathbb{E}\left[\left(\sum_{k=1}^K \sum_{h=1}^H \langle \mu_{\widetilde{\pi}_{k+1}}^h(\widetilde{P}_{k+1}) - \mu_{\pi^*}^h(\mathbb{P}), \widehat{\ell}_k^h \rangle\right)\mathbb{1}[\neg\mathcal{E}]\right].$$

For the second term, since by definition $0 \leq \widehat{\ell}_k^h(s,a) \leq L$ for all $k, h, s, a$ and both $\mu_{\widetilde{\pi}_{k+1}}^h(\widetilde{P}_{k+1})$ and $\mu_{\pi^*}^h(\mathbb{P})$ are probability distributions, we can bound it as

$$KHL \cdot \Pr\{\neg\mathcal{E}\} \leq 4\delta KHL.$$

Now consider the summation inside the first expectation. If we add an extra term $V(\widetilde{\pi}_1; \widehat{\ell}_0, \widetilde{P}_1) - V(\pi^*; \widehat{\ell}_0, \mathbb{P})$ where $\widehat{\ell}_0 = z$ is the perturbation. The following deduction holds under the event $\mathcal{E}$:

$$\sum_{k=0}^K V(\widetilde{\pi}_{k+1}; \widehat{\ell}_k, \widetilde{P}_{k+1}) - V(\pi^*; \widehat{\ell}_{0:K}, \mathbb{P})$$

$$\overset{(a)}{\leq} \sum_{k=0}^K V(\widetilde{\pi}_{k+1}; \widehat{\ell}_k, \widetilde{P}_{k+1}) - V(\widetilde{\pi}_{K+1}; \widehat{\ell}_{0:K}, \widetilde{P}_{K+1}) = \sum_{k=0}^{K-1} V(\widetilde{\pi}_{k+1}; \widehat{\ell}_k, \widetilde{P}_{k+1}) - V(\widetilde{\pi}_{K+1}; \widehat{\ell}_{0:K-1}, \widetilde{P}_{K+1})$$

$$\overset{(b)}{\leq} \sum_{k=0}^{K-1} V(\widetilde{\pi}_{k+1}; \widehat{\ell}_k, \widetilde{P}_{k+1}) - V(\widetilde{\pi}_K; \widehat{\ell}_{0:K-1}, \widetilde{P}_K) = \sum_{k=0}^{K-2} V(\widetilde{\pi}_{k+1}; \widehat{\ell}_k, \widetilde{P}_{k+1}) - V(\widetilde{\pi}_K; \widehat{\ell}_{0:K-2}, \widetilde{P}_K)$$

$$\leq \cdots \leq V(\widetilde{\pi}_1; \widehat{\ell}_0, \widetilde{P}_1) - V(\widetilde{\pi}_2; \widehat{\ell}_0, \widetilde{P}_2) \overset{(c)}{\leq} 0.$$

Here, (a) used the optimality of $(\widetilde{\pi}_{K+1}, \widetilde{P}_{K+1})$ over the set $\Pi \times \mathcal{P}_K$ w.r.t. losses $\widehat{\ell}_{0:K}$, which is valid due to $\mathcal{E}$; (b) used the optimality of $(\widetilde{\pi}_K, \widetilde{P}_K)$ over the set $\Pi \times \mathcal{P}_{K-1}$ w.r.t. losses $\widehat{\ell}_{0:K-1}$, which is

again valid since $\widetilde{P}_{K+1} \in \mathcal{P}_K \subseteq \mathcal{P}_{K-1}$; similarly (c) used the optimality of $(\widetilde{\pi}_1, \widetilde{P}_1)$ over $\Pi \times \mathcal{P}_0$, which again holds as $\widetilde{P}_2 \in \mathcal{P}_0$ (which is the set of all transitions). So we still have the following inequality as Lemma 10:

$$\mathbb{E}\left[\left(\sum_{k=1}^{K} V(\pi_{k+1}; \widehat{\ell}_k, \widetilde{P}_{k+1}) - \sum_{k=1}^{K} V(\pi^*; \widehat{\ell}_k, \mathbb{P})\right) \mathbb{1}[\mathcal{E}]\right] \leq \mathbb{E}[V(\pi^*; \widehat{\ell}_0, \mathbb{P}) - V(\widetilde{\pi}_1; \widehat{\ell}_0, \widetilde{P}_1)].$$

By the notation of occupancy measures, we can rewrite the last term as

$$\mathbb{E}\left[\sum_{h=1}^{H} \langle \mu_{\pi^*}^h(\mathbb{P}), \widehat{\ell}_0^h \rangle - \sum_{h=1}^{H} \langle \mu_{\widetilde{\pi}_1}^h(\widetilde{P}_1), \widehat{\ell}_0^h \rangle\right] \leq 2 \sum_{h=1}^{H} \mathbb{E}[\|\widehat{\ell}_0^h\|_\infty],$$

which is again bounded by $\frac{2H}{\eta}(1 + \ln(SA))$ due to Lemma 44. Combining these two parts (with or without $\mathcal{E}$) together gives our conclusion. $\qquad \square$

*Proof of Lemma 19.* We follow the proof of Lemma 11. For a fixed episode $k \in [K]$, we consider any $(\pi, P) \in \Pi \times \mathcal{P}_k$. We use the notation $(\pi, P) = \text{best}(\ell; \mathcal{P})$ to denote $(\pi, P) = \arg\min_{(\pi, P) \in \Pi \times \mathcal{P}} V(\pi; \ell, P)$. Then we have

$$\begin{aligned}
p_k(\pi, P) &= \int_z \mathbb{1}\left[(\pi, P) = \text{best}\left(\widehat{\ell}_{1:k-1} + z; \mathcal{P}_k\right)\right] f(z) \, \mathrm{d}z \\
&= \int_z \mathbb{1}\left[(\pi, P) = \text{best}\left(\widehat{\ell}_{1:k-1} + \left(z + \widehat{\ell}_k\right); \mathcal{P}_k\right)\right] f\left(z + \widehat{\ell}_k\right) \, \mathrm{d}z \\
&= \int_z \mathbb{1}\left[(\pi, P) = \text{best}\left(\widehat{\ell}_{1:k} + z; \mathcal{P}_k\right)\right] f\left(z + \widehat{\ell}_k\right) \, \mathrm{d}z,
\end{aligned}$$

where $f(z)$ is the probability density function of $z$ and the second step made use of the fact that $z + \widehat{\ell}_k$ is still linear in $z$. Moreover,

$$\widetilde{p}_{k+1}(\pi, P) = \int_z \mathbb{1}\left[(\pi, P) = \text{best}\left(\widehat{\ell}_{1:k} + z; \mathcal{P}_k\right)\right] f(z) \, \mathrm{d}z.$$

Again by the fact that $f(z) = \prod_{h=1}^{H} \exp(-\eta \|z^h\|_1)$, which we used in the proof of Lemma 11, we have

$$f\left(z + \widehat{\ell}_k\right) = \prod_{h=1}^{H} \exp\left(-\eta\left(\|z^h + \widehat{\ell}_k\|_1 - \|z^h\|\right)\right) f(z),$$

which gives

$$\frac{f\left(z + \widehat{\ell}_k\right)}{f(z)} \in \left[\exp\left(-\eta \sum_{h=1}^{H} \|\widehat{\ell}_k^h\|_1\right), \exp\left(\eta \sum_{h=1}^{H} \|\widehat{\ell}_k^h\|_1\right)\right]$$

by triangle inequality. Therefore, $\widetilde{p}_{k+1}(\pi, P)/p_k(\pi, P)$ lies in this interval as well, which is just our claim. $\qquad \square$

*Proof of Lemma 20.* Let us focus on a single episode, say $k \in [K]$. We should first make sure that $\widehat{q}_k^h(s, a) \leq q_k^h(s, a)$ (defined in Equations (12) and (13)), which happens when $\mathbb{P} \in \mathcal{P}_k$, i.e., $\mathcal{E}_k$ holds. Therefore, we rewrite the $k$-th summand of the stability term as

$$\mathbb{E}\left[\sum_{h=1}^{H} \sum_{\pi \in \Pi} \int_{\mathcal{P}_k} (p_k(\pi, P) - \widetilde{p}_{k+1}(\pi, P)) \langle \mu_\pi^h(P), \widehat{\ell}_k^h \rangle \, \mathrm{d}P \mathbb{1}[\mathcal{E}_k]\right] +$$

$$\mathbb{E}\left[\sum_{h=1}^{H} \sum_{\pi \in \Pi} \int_{\mathcal{P}_k} (p_k(\pi, P) - \widetilde{p}_{k+1}(\pi, P)) \langle \mu_\pi^h(P), \widehat{\ell}_k^h \rangle \, \mathrm{d}P \mathbb{1}[\neg \mathcal{E}_k]\right]. \tag{14}$$

For the second term, we will bound it trivially as $HL\Pr\{\neg\mathcal{E}_k\} \leq 4\delta HL$ as $p_k, \widetilde{p}_{k+1} \in \triangle(\Pi \times \mathcal{P}_k)$ and $\mu_\pi^h \in \triangle(\mathcal{S} \times \mathcal{A})$. For the first term, we will do something similar to Lemma 12, as follows:

Summing up Lemma 19 for all $(\pi, P) \in \Pi \times \mathcal{P}_k$ and using the fact that $\exp(-x) \geq (1-x)$ gives

$$
\sum_{\pi \in \Pi} \int_{\mathcal{P}_k} (p_k(\pi, P) - \widetilde{p}_{k+1}(\pi, P)) \sum_{h=1}^{H} \langle \mu_\pi^h(P), \widehat{\ell}_k^h \rangle \, \mathrm{d}P
$$
$$
\leq \eta \sum_{h'=1}^{H} \|\widehat{\ell}_k^{h'}\|_1 \cdot \sum_{\pi \in \Pi} \int_{\mathcal{P}_k} p_k(\pi, P) \sum_{h=1}^{H} \langle \mu_\pi^h(P), \widehat{\ell}_k^h \rangle \, \mathrm{d}P. \tag{15}
$$

By considering the randomness of $M_k^h$, we will still have the following property, except for a different definition of $q_k^h$:

$$
\mathbb{E}\left[\widehat{\ell}_k^h(s,a) \Big| (s_k^h, a_k^h) = (s,a)\right] \leq \frac{\ell_k^h(s,a)}{q_k^h(s,a)}, \quad \text{where } q_k^h(s,a) = \sum_{\pi \in \Pi} p_k(\pi) \max_{P' \in \mathcal{P}_k} \mu_\pi^h(s,a; P'), \tag{16}
$$

as when doing the Geometric Re-sampling process, we are picking the transition in $\mathcal{P}_k$ that maximizes the probability of reaching $(s,a)$. Still use $\mathbb{1}_k^h(s,a)$ as the shorthand notation of $\mathbb{1}[(s_k^h, a_k^h) = (s,a)]$. Then for any history $\mathcal{F}_{k-1}$ and those $h' \neq h$ in Eq. (15),

$$
\eta \, \mathbb{E}\left[\sum_{h=1}^{H} \sum_{s,a} \sum_{\pi \in \Pi} \int_{\mathcal{P}_k} p_k(\pi, P)\mu_\pi^h(s,a; P)\widehat{\ell}_k^h(s,a) \, \mathrm{d}P \cdot \sum_{h' \neq h} \|\widehat{\ell}_k^{h'}\|_1 \mathbb{1}[\mathcal{E}_k] \Big| \mathcal{F}_{k-1}\right]
$$
$$
\stackrel{(a)}{\leq} \eta \, \mathbb{E}\left[\sum_{h=1}^{H} \sum_{s,a} \mathbb{1}_k^h(s,a)\ell_k^h(s,a) \frac{\sum_{\pi \in \Pi} \int_{\mathcal{P}_k} p_k(\pi, P)\mu_\pi^h(s,a; P) \, \mathrm{d}P}{q_k^h(s,a)} \sum_{h' \neq h} \|\widehat{\ell}_k^{h'}\|_1 \mathbb{1}[\mathcal{E}_k] \Big| \mathcal{F}_{k-1}\right]
$$
$$
\stackrel{(b)}{\leq} \eta H \, \mathbb{E}\left[\sum_{h' \neq h} \|\widehat{\ell}_k^{h'}\|_1 \mathbb{1}[\mathcal{E}_k] \Big| \mathcal{F}_{k-1}\right] \stackrel{(c)}{\leq} \eta H \sum_{s,a} \mathbb{E}\left[\sum_{h=1}^{H} \frac{\widehat{q}_k^h(s,a)}{q_k^h(s,a)} \mathbb{1}[\mathcal{E}_k] \Big| \mathcal{F}_{k-1}\right] \stackrel{(d)}{\leq} \eta H^2 SA,
$$

where (a) is taking expectation w.r.t. $M_k^h$, (b) used the (new) definition of $q_k^h$ together with the fact that $\sum_{(s,a)} \mathbb{1}_k^h(s,a) = 1$, (c) used Corollary 40 and (d) used $\widehat{q}_k^h(s,a) \leq q_k^h(s,a)$ (which is due to $\mathbb{1}[\mathcal{E}_k]$).

For those terms with $h' = h$ in Eq. (15), by direct calculation and the fact that $\widehat{\ell}_k^h$ is a one-hot vector, we can write them as

$$
\eta \, \mathbb{E}\left[\sum_{h=1}^{H} \sum_{s,a} \sum_{\pi \in \Pi} \int_{\mathcal{P}_k} p_k(\pi, P)\mu_\pi^h(s,a; P) \left(\widehat{\ell}_k^h(s,a)\right)^2 \, \mathrm{d}P \mathbb{1}[\mathcal{E}_k] \Big| \mathcal{F}_{k-1}\right]
$$
$$
\leq 2\eta \, \mathbb{E}\left[\sum_{h,s,a} \frac{\sum_{\pi \in \Pi} \int_{\mathcal{P}_k} p_k(\pi, P)\mu_\pi^h(s,a; P) \, \mathrm{d}P}{q_k^h(s,a)} \frac{\widehat{q}_k^h(s,a)}{q_k^h(s,a)} \mathbb{1}[\mathcal{E}_k] \Big| \mathcal{F}_{k-1}\right]
$$
$$
\leq 2\eta \sum_{s,a} \mathbb{E}\left[\sum_{h=1}^{H} \frac{\widehat{q}_k^h(s,a)}{q_k^h(s,a)} \mathbb{1}[\mathcal{E}_k] \Big| \mathcal{F}_{k-1}\right] \leq 2\eta HSA,
$$

by applying Corollary 41 together with the fact that $\widehat{q}_k^h(s,a) \leq q_k^h(s,a)$ when $\mathcal{E}_k$ happens. Combining the terms with $h' \neq h$ and the ones with $h' = h$ gives

$$
\text{Eq. (14)} \leq 3\eta H^2 SA + 4\delta HL.
$$

Therefore, the stability term is bounded by $3\eta H^2 SAK + 4\delta KHL$, as claimed. $\qquad \square$

### B.2.4 Bounding Other Terms

The terms other than ESTREG can be bounded similarly to Jin et al. (2020), as follows:

**Theorem 21** (Bounding ERROR Term). *The* ERROR *term is bounded by*

$$\text{ERROR} = \mathbb{E}\left[\sum_{k=1}^{K}\left(V(\pi_k; \ell_k, \mathbb{P}) - V(\pi_k; \ell_k, P_k)\right)\right] \leq \widetilde{\mathcal{O}}\left(H^2 S\sqrt{AK} + \delta KH\right).$$

**Theorem 22** (Bounding BIAS1 Term). *The* BIAS *term is bounded by*

$$\text{BIAS1} = \mathbb{E}\left[\sum_{k=1}^{K}\left(V(\pi_k; \ell_k, P_k) - V(\pi_k; \widehat{\ell}_k, P_k)\right)\right]$$

$$\leq \widetilde{\mathcal{O}}\left(\frac{HKSA}{L} + H^2 S\sqrt{AK} + H^3 S^3 A + \delta KH\right).$$

**Remark 23.** *This term looks quite similar to the GR error term (Lemma 9). However, they are in fact different as we will have some extra terms due to the UOB technique. In other words, we are having different probabilities when reaching $(s_k^h, a_k^h)$ and when doing Geometric Re-sampling (c.f. Lemma 38 and corollary 40). Therefore, this term will be further decomposed into two parts, where the first one is due to bias of the GR estimator and the second one is due to the UOB technique and can be bounded similar to Jin et al. (2022, Lemma A.3). Check the proof below for more details.*

**Theorem 24** (Bounding BIAS2 Term). *The* BIAS2 *term is bounded by*

$$\text{BIAS2} = \mathbb{E}\left[\sum_{k=1}^{K}\left(V(\pi^*; \widehat{\ell}_k, \mathbb{P}) - V(\pi^*; \ell_k, \mathbb{P})\right)\right] = \widetilde{\mathcal{O}}(\delta KHL).$$

*Proof of Theorem 5.* By combining Theorems 17, 21, 22 and 24 together, we will have

$$\mathcal{R}_T \leq \widetilde{\mathcal{O}}\left(H^2 S\sqrt{AK} + \frac{H}{\eta} + \eta H^2 SAK + \frac{HSAK}{L} + H^2 S\sqrt{AK} + H^3 S^3 A + \delta KHL\right).$$

Picking $\eta = \left(\sqrt{HSAK}\right)^{-1}$, $L = \sqrt{SAK/H}$ and $\delta = 1/K$ gives

$$\mathcal{R}_T \leq \widetilde{\mathcal{O}}\left(H^2 S\sqrt{AK} + H^{3/2}\sqrt{SAK} + H\sqrt{K} + H^3 S^3 A\right) = \widetilde{\mathcal{O}}\left(H^2 S\sqrt{AK} + H^3 S^3 A\right),$$

which finishes the proof. □

*Proof of Theorem 21.* We need the following key lemma from Jin et al. (2020):[4]

**Lemma 25** (Jin et al. (2020, Lemma 4)). *Conditioning on $\mathcal{E}$, for any set of policies $\{\pi_k \in \Pi\}_{k \in [K]}$ and any collection of transitions $\{P_k^{s,h}\}_{s \in \mathcal{S}, h \in [H]}$ such that $P_k^{s,h} \in \mathcal{P}_k$, with probability $1 - 2\delta$,*

$$\sum_{k=1}^{K}\sum_{h=1}^{H}\sum_{(s,a) \in \mathcal{S} \times \mathcal{A}} |\mu_{\pi_k}^h(s, a; P_k^{s,h}) - \mu_{\pi_k}^h(s, a; \mathbb{P})| \leq \widetilde{\mathcal{O}}\left(H^2 S\sqrt{AK}\right).$$

As all losses are in $[0, 1]$ (note that in the ERROR term we are considering true losses), we have

$$\sum_{k=1}^{K}\left(V(\pi_k; \ell_k, \mathbb{P}) - V(\pi_k; \ell_k, P_k)\right) \leq \sum_{k=1}^{K}\sum_{h=1}^{H}\sum_{(s,a) \in \mathcal{S} \times \mathcal{A}} |\mu_{\pi_k}^h(s, a; P_k) - \mu_{\pi_k}^h(s, a; \mathbb{P})|,$$

which is bounded by $\widetilde{\mathcal{O}}(H^2 S\sqrt{AK})$ with probability $1 - 2\delta$ by the previous lemma. Let the event (i.e., it is bounded by $\widetilde{\mathcal{O}}(H^2 S\sqrt{AK})$) be $\mathcal{E}'$. Then

$$\Pr\{\mathcal{E}' \wedge \mathcal{E}\} = \Pr\{\mathcal{E}' \mid \mathcal{E}\}\Pr\{\mathcal{E}\} \geq 1 - 6\delta.$$

---

[4]The original paper has a slightly different notation as they assumed the states to be 'layered', i.e., $\mathcal{S} = \mathcal{S}_1 \cup \mathcal{S}_2 \cup \cdots \cup \mathcal{S}_H$ such that the states in $\mathcal{S}_h$ can only transit to $\mathcal{S}_{h+1}$, $\forall 1 \leq h < H$. Therefore, their $S$ should be $H$ times larger than ours. They also used $T$ for our $K$, $L$ for our $H$ and $X$ for our $\mathcal{S}$.

Therefore, we write

$$\mathbb{E}\left[\sum_{k=1}^{K}\left(V(\pi_k;\ell_k,\mathbb{P})-V(\pi_k;\ell_k,P_k)\right)\right]=\mathbb{E}\left[\sum_{k=1}^{K}\left(V(\pi_k;\ell_k,\mathbb{P})-V(\pi_k;\ell_k,P_k)\right)\mathbb{1}[\mathcal{E}\wedge\mathcal{E}']\right]+$$

$$\mathbb{E}\left[\sum_{k=1}^{K}\sum_{h=1}^{H}\langle\mu_{\pi_k}^h(\mathbb{P})-\mu_{\pi_k}^h(P_k),\ell_k^h\rangle\mathbb{1}[\neg\mathcal{E}\vee\neg\mathcal{E}']\right]$$

$$=\widetilde{\mathcal{O}}\left(H^2S\sqrt{AK}+\delta KH\right),$$

where the last step used the fact that $\mu_{\pi_k}^h(\mathbb{P})$ and $\mu_{\pi_k}^h(P_k)$ are both probability distributions and $0\le\ell_k^h(s,a)\le1$. $\qquad\square$

*Proof of Theorem 22.* Write our BIAS1 term in terms of occupancy measures:

$$\text{BIAS1}=\mathbb{E}\left[\sum_{k=1}^{K}\sum_{\pi\in\Pi}\int_{\mathcal{P}_k}p_k(\pi,P)\sum_{h=1}^{H}\langle\mu_\pi^h(P),\ell_k^h-\widehat{\ell}_k^h\rangle\,\mathrm{d}P\right].$$

Consider the $k$-th summand of it, denoted as $\text{BIAS1}_k$. We decompose it into two parts, depending on whether $\mathcal{E}_k$ holds:

$$\text{BIAS1}_k\le\mathbb{E}\left[\sum_{\pi\in\Pi}\int_{\mathcal{P}_k}p_k(\pi,P)\sum_{h=1}^{H}\mathbb{E}\left[\langle\mu_\pi^h(P),\ell_k^h-\widehat{\ell}_k^h\rangle\mid\mathcal{F}_{k-1}\right]\,\mathrm{d}P\mathbb{1}[\mathcal{E}_k]\right]+$$

$$\mathbb{E}\left[\sum_{\pi\in\Pi}\int_{\mathcal{P}_k}p_k(\pi,P)\sum_{h=1}^{H}\langle\mu_\pi^h(P),\ell_k^h\rangle\,\mathrm{d}P\mathbb{1}[\neg\mathcal{E}_k]\right]$$

$$\triangleq\text{BIAS1}_k^{\mathcal{E}}+\text{BIAS1}_k^{\neg\mathcal{E}}.$$

For $\text{BIAS1}_k^{\neg\mathcal{E}}$, we bound it trivially as $H\Pr\{\neg\mathcal{E}_k\}\le4\delta H$ as $p_k\in\triangle(\pi\times\mathcal{P}_k)$, $\mu_\pi^h(P)\in\triangle(\mathcal{S}\times\mathcal{A})$ and $\ell_k^h(s,a)\in[0,1]$. For $\text{BIAS1}_k^{\mathcal{E}}$, we still adopt the notations of $\widehat{q}_k^h(s,a)$ and $q_k^h(s,a)$, which are defined as

$$\widehat{q}_k^h(s,a)=\sum_{\pi\in\Pi}\int_{\mathcal{P}_k}p_k(\pi,P)\mu_\pi^h(s,a;\mathbb{P})\,\mathrm{d}P,$$

$$q_k^h(s,a)=\sum_{\pi\in\Pi}\int_{\mathcal{P}_k}p_k(\pi,P)\max_{P'\in\mathcal{P}_k}\mu_\pi^h(s,a;P')\,\mathrm{d}P.$$

Applying Corollary 40 to $\mathbb{E}[\widehat{\ell}_k^h(s,a)\mid\mathcal{F}_{k-1}]$, $\forall(s,a)\in\mathcal{S}\times\mathcal{A}$ then gives

$$\text{BIAS1}_k^{\mathcal{E}}=\mathbb{E}\left[\sum_{\pi\in\Pi}\int_{\mathcal{P}_k}p_k(\pi,P)\sum_{h=1}^{H}\left\langle\mu_\pi^h(P),\left(1-\frac{\widehat{q}_k^h}{q_k^h}+\frac{\widehat{q}_k^h}{q_k^h}(1-q_k^h)^L\right)\ell_k^h\right\rangle\,\mathrm{d}P\mathbb{1}[\mathcal{E}_k]\right]$$

(every operation for the second term of the inner product is element-wise). As $\mathbb{1}[\mathcal{E}_k]$ implies $\widehat{q}_k^h(s,a)\le q_k^h(s,a)$, we can further bound $\text{BIAS1}_k^{\mathcal{E}}$ as

$$\text{BIAS1}_k^{\mathcal{E}}\le\mathbb{E}\left[\sum_{h=1}^{H}\sum_{(s,a)\in\mathcal{S}\times\mathcal{A}}\sum_{\pi\in\Pi}\int_{\mathcal{P}_k}p_k(\pi,P)\mu_\pi^h(s,a;P)\frac{q_k^h(s,a)-\widehat{q}_k^h(s,a)}{q_k^h(s,a)}\ell_k^h(s,a)\,\mathrm{d}P\mathbb{1}[\mathcal{E}_k]\right]+$$

$$\mathbb{E}\left[\sum_{h=1}^{H}\sum_{(s,a)\in\mathcal{S}\times\mathcal{A}}\sum_{\pi\in\Pi}\int_{\mathcal{P}_k}p_k(\pi,P)\mu_\pi^h(s,a;P)(1-q_k^h(s,a))^L\,\mathrm{d}P\mathbb{1}[\mathcal{E}_k]\right].$$

For the second term, we can simply make use of the fact that

$$\mathbb{1}[\mathcal{E}_k]=1\implies\sum_{\pi\in\Pi}\int_{\mathcal{P}_k}p_k(\pi,P)\mu_\pi^h(s,a;P)\,\mathrm{d}P\le q_k^h(s,a)\tag{17}$$

together with the condition that $\ell_k^h(s,a) \in [0,1]$ and consequently bound it by

$$\mathbb{E}\left[\sum_{h=1}^{H}\sum_{(s,a)\in\mathcal{S}\times\mathcal{A}} q_k^h(s,a)(1-q_k^h(s,a))^L \mathbb{1}[\mathcal{E}_k]\right] \overset{(c)}{\leq} \frac{HSA}{eL},$$

where (c) used the fact that $q(1-q)^L \leq qe^{-Lq} \leq \frac{1}{eL}$, just as what we did in Lemma 9. For the first term, with a slight abuse of notations, we still use $p_k(\pi)$ to denote the probability of playing $\pi$ at episode $k$, i.e., $p_k(\pi) = \int_{P\in\mathcal{P}_k} p_k(\pi,P)\,\mathrm{d}P$. Then again by Eq. (17), we are actually facing

$$\mathbb{E}\left[\sum_{h=1}^{H}\sum_{(s,a)\in\mathcal{S}\times\mathcal{A}} \left(q_k^h(s,a) - \widehat{q}_k^h(s,a)\right)\mathbb{1}[\mathcal{E}_k]\right] \tag{18}$$

$$= \mathbb{E}\left[\sum_{\pi\in\Pi} p_k(\pi)\sum_{h=1}^{H}\sum_{(s,a)\in\mathcal{S}\times\mathcal{A}} \left(\max_{P'\in P_k}\mu_\pi^h(s,a;P') - \mu_\pi^h(s,a;\mathbb{P})\right)\mathbb{1}[\mathcal{E}_k]\right]. \tag{19}$$

Then we follow the idea of Jin et al. (2022, Lemma A.3). We fix the step $h \in [H]$ and the state-action pair $(s,a) \in \mathcal{S}\times\mathcal{A}$. Therefore, for each policy $\pi \in \Pi$, we can define $\widehat{P}_\pi \in \mathcal{P}_k$ to be transition corresponding to the upper-occupancy bound, i.e., it maximizes $\mu_\pi^h(s,a;P)$ over all transitions $P \in \mathcal{P}_k$. Therefore, with the help of the so-called "occupancy difference lemma" (Jin et al., 2021, Lemma D.3.1), we can write the summand in Eq. (19) corresponding to $k,h,s,a$ as

$$q_k^h(s,a) - \widehat{q}_k^h(s,a) = \sum_{\pi\in\Pi} p_k(\pi)\left(\mu_\pi^h(s,a;\widehat{P}_{\pi,h,s,a}) - \mu_\pi^h(s,a;\mathbb{P})\right)$$

$$= \sum_{\pi\in\Pi} p_k(\pi)\sum_{h'=0}^{h-1}\sum_{x\in\mathcal{S},y\in\mathcal{A},z\in\mathcal{S}} \mu_\pi^{h'}(x,y;\mathbb{P})\left(\mathbb{P}^{h'}(z\mid x,y) - \widehat{P}_\pi^{h'}(z\mid x,y)\right)\mu_\pi^{h\mid h'+1}(s,a\mid z;\widehat{P}_\pi),$$

where $\mu_\pi^{h\mid h'+1}(s,a\mid z;P)$ is the so-called "conditional occupancy measure", which is defined as the conditional probability of reaching the state-action pair $(s,a)$ at step $h$ from state $z$ at step $h'+1$ with policy $\pi$ and transition $P$. By $\mathcal{E}_k$, we have $\mathbb{P} \in \mathcal{P}_k$. Therefore, by the definition of confidence radii, we can further bound

$$|q_k^h(s,a) - \widehat{q}_k^h(s,a)| \leq \sum_{\pi\in\Pi} p_k(\pi)\sum_{h'=0}^{h-1}\sum_{x,y,z} \mu_\pi^{h'}(x,y;\mathbb{P})\epsilon_k^{h'}(z\mid x,y)\mu_\pi^{h\mid h'+1}(s,a\mid z;\widehat{P}_\pi),$$

where $\epsilon_k^h$ is defined as in Eq. (11).

Then, we consider the conditional occupancy measure $\mu_\pi^{h\mid h'+1}$ w.r.t. $\widehat{P}_\pi$. We can still use occupancy difference lemmas (but now we only consider steps between $h'+1$ and $h$) to write its difference with the conditional occupancy measure w.r.t. $\mathbb{P}$ as

$$\mu_\pi^{h\mid h'+1}(s,a\mid z;\widehat{P}_\pi) - \mu_\pi^{h\mid h'+1}(s,a\mid z;\mathbb{P})$$

$$\leq \sum_{h''=h'+1}^{h-1}\sum_{u\in\mathcal{S},v\in\mathcal{A},w\in\mathcal{S}} \mu_\pi^{h''\mid h'+1}(u,v\mid z;\mathbb{P})\epsilon_k^{h''}(w\mid u,v)\mu_\pi^{h\mid h''+1}(s,a\mid w;\widehat{P}_\pi)$$

$$\leq \sum_{h''=h'+1}^{h-1}\sum_{u\in\mathcal{S},v\in\mathcal{A},w\in\mathcal{S}} \mu_\pi^{h''\mid h'+1}(u,v\mid z;\mathbb{P})\epsilon_k^{h''}(w\mid u,v)\pi^h(a\mid s),$$

where the first step follows from the same reasoning as the unconditioned ones and the second step,

$$\mu_\pi^{h\mid h''+1}(s,a\mid w;\widehat{P}_\pi) = \pi^h(a\mid s)\Pr\{s^h = s\mid s^{h''+1} = w,\pi,\widehat{P}_\pi\} \leq \pi^h(a\mid s).$$

Hence, plugging back into Eq. (19) gives its bound as

$$\mathbb{E}\left[\sum_{h,s,a}\sum_{\pi\in\Pi} p_k(\pi)\sum_{h'=0}^{h-1}\sum_{x,y,z} \mu_\pi^{h'}(x,y;\mathbb{P})\epsilon_k^{h'}(z\mid x,y)\mu_\pi^{h\mid h'+1}(s,a\mid z;\widehat{P}_\pi)\right]$$

$$\leq \sum_{h,s,a} \mathbb{E}\left[\sum_{\pi\in\Pi} p_k(\pi)\sum_{h'=0}^{h-1}\sum_{x,y,z}\mu_\pi^{h'}(x,y;\mathbb{P})\epsilon_k^{h'}(z\mid x,y)\mu_\pi^{h\mid h'+1}(s,a\mid z;\mathbb{P})\right] +$$

$$\sum_{h,s,a} \mathbb{E}\left[\sum_{\pi\in\Pi} p_k(\pi)\sum_{h'=0}^{h-1}\sum_{x,y,z}\mu_\pi^{h'}(x,y;\mathbb{P})\epsilon_k^{h'}(z\mid x,y)\sum_{h''=h'+1}^{h-1}\sum_{u,v,w}\mu_\pi^{h''\mid h'+1}(u,v\mid z;\mathbb{P})\epsilon_k^{h''}(w\mid u,v)\pi^h(a\mid s)\right].$$

The remaining part of the proof is exactly the same as that for Lemma A.3 of Jin et al. (2022), which eventually shows,

$$\sum_{k=1}^K \text{BIAS}1_k^{\mathcal{E}} = \widetilde{\mathcal{O}}\left(\frac{HSAK}{L} + H^2 S\sqrt{AK} + H^3 S^3 A + \delta H K\right). \tag{20}$$

Combining the two parts together (with or without $\mathcal{E}_k$) gives,

$$\text{BIAS}1 = \widetilde{\mathcal{O}}\left(\frac{HSAK}{L} + H^2 S\sqrt{AK} + H^3 S^3 A + \delta H K\right),$$

as claimed. $\qquad\square$

*Proof of Theorem 24.* This proof is quite simple. We still decompose $\text{BIAS}2$ into two parts:

$$\text{BIAS}2 = \mathbb{E}\left[\sum_{k=1}^K\left(V(\pi^*;\widehat{\ell}_k,\mathbb{P}) - V(\pi^*;\ell_k,\mathbb{P})\right)\right]$$

$$= \sum_{k=1}^K\mathbb{E}\left[\left(V(\pi^*;\widehat{\ell}_k,\mathbb{P}) - V(\pi^*;\ell_k,\mathbb{P})\right)\mathbb{1}[\mathcal{E}_k]\right] + \sum_{k=1}^K\mathbb{E}\left[\left(V(\pi^*;\widehat{\ell}_k,\mathbb{P}) - V(\pi^*;\ell_k,\mathbb{P})\right)\mathbb{1}[\neg\mathcal{E}_k]\right].$$

For the first term, as $\mathbb{1}[\mathcal{E}_k]$ infers $\widehat{q}_k^h(s,a) \leq q_k^h(s,a)$, from Corollary 40, we have

$$\mathbb{E}[\widehat{\ell}_k(s,a)\mid\mathcal{F}_{k-1}] \leq \ell_k(s,a), \quad \forall k\in[K], (s,a)\in\mathcal{S}\times\mathcal{A}.$$

Therefore, as both $\pi^*$ and $\mathbb{P}$ are deterministic, this term is upper bounded by 0. For the second term, we trivially bound each of the summand by $HL\Pr\{\neg\mathcal{E}_k\} \leq 4\delta HL$ as $|\widehat{\ell}_k^h(s,a)| \leq L$. Therefore, combining two terms together completes the proof. $\qquad\square$

## C    Analysis of Episodic AMDP Algorithms with Delayed Feedback (Theorem 6)

In this section, we consider episodic AMDPs with delayed bandit feedback and unknown transitions. The algorithm is presented in Algorithm 4, which is very similar to Algorithm 3 except for the part on handling delayed feedback, highlighted in violet.

### C.1    Regret Decomposition

*Proof of Theorem 6.* For this case, we still use the regret decomposition as Theorem 5, as follows:

$$\mathcal{R}_K = \underbrace{\mathbb{E}\left[\sum_{k=1}^K (V(\pi_k;\ell_k,\mathbb{P}) - V(\pi_k;\ell_k,P_k))\right]}_{\text{ERROR}} + \underbrace{\mathbb{E}\left[\sum_{k=1}^K\left(V(\pi_k;\ell_k,P_k) - V(\pi_k;\widehat{\ell}_k,P_k)\right)\right]}_{\text{BIAS}1} +$$

$$\underbrace{\mathbb{E}\left[\sum_{k=1}^K\left(V(\pi_k;\widehat{\ell}_k,P_k) - V(\pi^*;\widehat{\ell}_k,\mathbb{P})\right)\right]}_{\text{ESTREG}} + \underbrace{\mathbb{E}\left[\sum_{k=1}^K\left(V(\pi^*;\widehat{\ell}_k,\mathbb{P}) - V(\pi^*;\ell_k,\mathbb{P})\right)\right]}_{\text{BIAS}2}.$$

**Algorithm 4** FTPL for Episodic AMDPs with Delayed Bandit Feedback and Unknown Transition

**Require:** Laplace distribution parameter $\eta$. Geometric Re-sampling parameter $L$.
1: Initialize $\mathcal{P}_1 \leftarrow (\triangle(\mathcal{S}))^{[H] \times \mathcal{S} \times \mathcal{A}}$.
2: Sample perturbation $\widehat{\ell}_0 = z$ such that $z^h(s, a)$ is an independent sample of Laplace($\eta$).
3: **for** $k = 1, 2, \ldots, K$ **do**
4:      Let $(\pi_k, P_k) = \operatorname{argmin}_{(\pi, P) \in \Pi \times \mathcal{P}_k} V(\pi; \sum_{k' \in \Omega_k} \widehat{\ell}_{k'} + z, P)$ by Extended Value Iteration (Jaksch et al., 2010), where $\Omega_k = \{k' \mid k' + d_{k'} < k\}$. (See also Remark 16 for more details.)
5:      **for** $h = 1, 2, \ldots, H$ **do**
6:          Observe $s_k^h$, play $a_k^h = \pi_k(s_k^h)$, suffer loss $\ell_k^h(s_k^h, a_k^h)$.
7:      **for** All $k' < k$ such that $k' + d_{k'} = k$ **do**
8:          **for** $h = 1, 2, \ldots, H$ **do**
9:              **for** $M_{k'}^h = 1, 2, \ldots, L$ **do**
10:                  Sample a fresh perturbation $\widetilde{z}$ in the same way as $z$.
11:                  Calculate $(\pi'_{k'}, P'_{k'}) = \operatorname{argmin}_{(\pi, P) \in \Pi \times \mathcal{P}_{k'}} V(\pi; \sum_{j \in \Omega_{k'}} \widehat{\ell}_j + \widetilde{z})$.
12:                  Pick the transition $\widehat{P}'_{k'} \in \mathcal{P}_{k'}$ such that $\mu_\pi^h(s_{k'}^h, a_{k'}^h; \widehat{P}'_{k'})$ is maximized via the Comp-UOB procedure proposed by Jin et al. (2020).
13:                  Simulate $\pi'_{k'}$ for $h$ steps starting from $s^1$ and following transitions $(\widehat{P}'_{k'})^1, \ldots, (\widehat{P}'_{k'})^h$.
14:                  **if** $(s_{k'}^h, a_{k'}^h)$ is visited at step $h$ or $M_{k'}^h = L$ **then**
15:                      Set $\widehat{\ell}_{k'}^h(s, a) = M_{k'}^h \cdot \ell_{k'}^h(s_{k'}^h, a_{k'}^h) \cdot \mathbb{1}[(s_{k'}^h, a_{k'}^h) = (s, a)]$ and break.
16:      Calculate $\mathcal{P}_{k+1}$ according to Eq. (10).

Note that as delays will not affect transitions as well as the loss estimators (viewed in hindsight, i.e., the sequence $\{\widehat{\ell}_k\}_{k \in [K]}$ will be the same as if there is no delays), so the ERROR, BIAS1 and BIAS2 can still be bounded by Theorems 21, 22 and 24, respectively. The only difference occurs when bounding ESTREG, which we show as follows.

**Lemma 26** (Bounding ESTREG Term with Delayed Feedback). *The* ESTREG *term is bounded by*

$$\mathbb{E}\left[\sum_{k=1}^K \left(V(\pi_k; \widehat{\ell}_k, P_k) - V(\pi^*; \widehat{\ell}_k, \mathbb{P})\right)\right] \leq \frac{2H}{\eta}\left(1 + \ln(SA)\right) + 5\eta H^2 SAK + \eta H^2 SA\mathfrak{D} + 12\delta KHL.$$

As mentioned in the main body, the key difference is that, we will compete a learner that is not only cheating but also stepping one episode further. However, as it is still using FTPL, we can still bound the stability term as in Lemma 20. Therefore, the proof is postponed to the end of this section.

Combining the bounds for the four terms together, we will have

$$\mathcal{R}_T \leq \widetilde{\mathcal{O}}\left(H^2 SA\sqrt{K} + \frac{H}{\eta} + \eta H^2 SA(K + \mathfrak{D}) + \frac{SAHK}{L} + \delta HKL\right).$$

Therefore, picking $\eta = \left(\sqrt{HSA(K + \mathfrak{D})}\right)$, $L = \sqrt{SAK/H}$ and $\delta = 1/K$ gives

$$\mathcal{R}_T \leq \widetilde{\mathcal{O}}\left(H^2 SA\sqrt{K} + H^{3/2}\sqrt{SA\mathfrak{D}}\right),$$

as claimed. $\qquad\qquad\qquad\qquad\qquad\qquad\qquad\qquad\qquad\qquad\qquad\qquad\qquad\qquad\qquad\qquad\square$

*Proof of Lemma 26.* Slightly different from the main text, we now consider the following *two* learners, where the first one is a "cheating learner" that does not suffer any delays, and the second one is a "cheating leader" that not only does not suffer any delays, but also looks one step further.

$$(\widehat{\pi}_k, \widehat{P}_k) \triangleq \operatorname*{argmin}_{(\pi, P) \in \Pi \times \mathcal{P}_k} V(\pi; \widehat{\ell}_{0:k-1}, P), \quad (\widetilde{\pi}_{k+1}, \widetilde{P}_{k+1}) \triangleq \operatorname*{argmin}_{(\pi, P) \in \Pi \times \mathcal{P}_k} V(\pi; \widehat{\ell}_{0:k}, P).$$

Note that both of them are defined w.r.t. transitions in $\mathcal{P}_k$ instead of the subset $\mathcal{P}_{k+1}$, which is the same as Appendix B.2. We also define the following three density functions with respect to

the perturbation $z$: $p_k(\pi, P)$ for $(\pi_k, P_k)$ conditioning on $\widehat{\ell}_1, \widehat{\ell}_2, \ldots, \widehat{\ell}_{k-1}$, $\widehat{p}_k(\pi, P)$ for $(\widehat{\pi}_k, \widehat{P}_k)$ conditioning on $\widehat{\ell}_1, \widehat{\ell}_2, \ldots, \widehat{\ell}_{k-1}$ and $\widetilde{p}_{k+1}(\pi, P)$ for $(\widetilde{\pi}_{k+1}, \widetilde{P}_{k+1})$ conditioning on $\widehat{\ell}_1, \widehat{\ell}_2, \ldots, \widehat{\ell}_k$.

The purpose of defining two learners is to decouple the effects from delays and the inherent FTPL regret. One can see that our $(\widehat{\pi}_k, \widehat{P}_k)$ is equivalent to $(\pi_k, P_k)$ in Appendix B.2 while $(\widetilde{\pi}_{k+1}, \widetilde{P}_{k+1})$ remains the same. Therefore, the difference between $(\widehat{\pi}_k, \widehat{P}_k)$ and $(\widetilde{\pi}_k, \widetilde{P}_k)$ can be bounded exactly the same as Appendix B.2 and we only need to care about delays, i.e., the difference between $(\pi_k, P_k)$ and $(\widehat{\pi}_k, \widehat{P}_k)$. Formally, we decompose the EstReg into three terms:

$$\mathbb{E}\left[\sum_{k=1}^{K}\sum_{h=1}^{H}\langle\mu_{\pi_k}^h(P_k) - \mu_{\pi^*}^h(\mathbb{P}), \widehat{\ell}_k^h\rangle\right] = \underbrace{\mathbb{E}\left[\sum_{k=1}^{K}\sum_{h=1}^{H}\langle\mu_{\pi_k}^h(P_k) - \mu_{\widehat{\pi}_k}^h(\widehat{P}_k), \widehat{\ell}_k^h\rangle\right]}_{\text{Cheating regret}} +$$

$$\underbrace{\mathbb{E}\left[\sum_{k=1}^{K}\sum_{h=1}^{H}\langle\mu_{\widehat{\pi}_k}^h(\widehat{P}_k) - \mu_{\widetilde{\pi}_{k+1}}^h(\widetilde{P}_{k+1}), \widehat{\ell}_k^h\rangle\right]}_{\text{Stability term}} + \underbrace{\mathbb{E}\left[\sum_{k=1}^{K}\sum_{h=1}^{H}\langle\mu_{\widetilde{\pi}_{k+1}}^h(\widetilde{P}_{k+1}) - \mu_{\pi^*}^h(\mathbb{P}), \widehat{\ell}_k^h\rangle\right]}_{\text{Error term}}.$$

Note that the error term and the stability term are exactly the same as Appendix B.2, so we can directly make use of Lemmas 18 and 20 and bound them by $\frac{2H}{\eta}(1 + \ln(SA)) + 4\delta KHL$ and $3\eta H^2 SAK + 4\delta KHL$, respectively. Now consider the cheating regret. Similar to the stability term, we will have the following single-step stability lemma:

**Lemma 27.** *For any $k \in [K]$, $(s,a) \in \mathcal{S} \times \mathcal{A}$ and $(\pi, P) \in \Pi \times \mathcal{P}_k$, we have*

$$\widehat{p}_k(\pi, P) \geq p_k(\pi, P)\exp\left(-\eta\sum_{k'\in\widetilde{\Omega}_k}\sum_{h=1}^{H}\|\widehat{\ell}_{k'}^h\|_1\right),$$

*where $\widetilde{\Omega}_k \triangleq [k-1] \setminus \Omega_k = \{k' < k \mid k' + d_{k'} \geq k\}$, i.e., the first $k-1$ rounds excluding those where the feedback is available before round $k$.*

*Proof.* Note that the proof of Lemma 19 does not rely on the concrete choice of $\pi_k$ and $\widetilde{\pi}_{k+1}$. Therefore, adopting the proof of Lemma 19 with $\widehat{\pi}_k$ as $\widetilde{\pi}_{k+1}$ will complete our proof. $\square$

With the help of Lemma 27, we can bound the cheating regret similar to the stability term. To see this, consider a fixed $k \in [K]$, we have

$$\mathbb{E}\left[\sum_{\pi\in\Pi}\int_{\mathcal{P}_k}(p_k(\pi, P) - \widehat{p}_k(\pi, P))\sum_{h=1}^{H}\langle\mu_\pi^h(P), \widehat{\ell}_k^h\rangle\,\mathrm{d}P\right]$$

$$= \mathbb{E}\left[\sum_{\pi\in\Pi}\int_{\mathcal{P}_k}(p_k(\pi, P) - \widehat{p}_k(\pi, P))\sum_{h=1}^{H}\langle\mu_\pi^h(P), \widehat{\ell}_k^h\rangle\,\mathrm{d}P\mathbb{1}[\mathcal{E}_k]\right] +$$

$$\mathbb{E}\left[\sum_{\pi\in\Pi}\int_{\mathcal{P}_k}(p_k(\pi, P) - \widehat{p}_k(\pi, P))\sum_{h=1}^{H}\langle\mu_\pi^h(P), \widehat{\ell}_k^h\rangle\,\mathrm{d}\mathbb{1}[\neg\mathcal{E}_k]\right]$$

$$\leq \mathbb{E}\left[\eta\sum_{k'\in\widetilde{\Omega}_k}\sum_{h'=1}^{H}\|\widehat{\ell}_{k'}^{h'}\|_1 \cdot \sum_{\pi\in\Pi}\int_{\mathcal{P}_k}p_k(\pi, P)\sum_{h=1}^{H}\langle\mu_\pi^h(P), \widehat{\ell}_k^h\rangle\,\mathrm{d}P\mathbb{1}[\mathcal{E}_k]\right] + 4\delta KHL$$

$$\leq \eta\mathbb{E}\left[\sum_{h=1}^{H}\sum_{(k',h')\in\widetilde{\Omega}_k\times[H]}\mathbb{1}[(k',h') \neq (k,h)]\|\widehat{\ell}_{k'}^{h'}\|_1\sum_{(s,a)\in\mathcal{S}\times\mathcal{A}}\widehat{q}_k^h(s,a)\widehat{\ell}_k^h(s,a)\mathbb{1}[\mathcal{E}_k]\right] +$$

$$\eta\mathbb{E}\left[\sum_{(s,a)\in\mathcal{S}\times\mathcal{A}}\sum_{h=1}^{H}\widehat{q}_k^h(s,a)\left(\widehat{\ell}_k^h(s,a)\right)^2\mathbb{1}[\mathcal{E}_k]\right] + 4\delta KHL,$$

where $\widehat{q}_k^h(s,a)$ is the actual probability of reaching $(s,a)$ and $q_k^h(s,a)$ is the probability of reaching $(s,a)$ in a single Geometric Re-sampling trial, as defined in Equations (12) and (13). Note that $\mathbb{1}[\mathcal{E}_k]$ implies $\widehat{q}_k^h(s,a) \leq q_k^h(s,a)$.

For the second term, using Lemma 39 and $\widehat{q}_k^h(s,a) \leq q_k^h(s,a)$ gives $2\eta SAH$. For the first one, taking expectation w.r.t. $M_k^h$ in $\widehat{\ell}_k^h(s,a) = \mathbb{1}[(s_k^h, a_k^h) = (s,a)]\ell_k^h(s,a)M_k^h(s,a)$ and then w.r.t. $\|\widehat{\ell}_{k'}^{h'}\|_1$ as in Lemmas 12 and 20 gives $\eta H^2 SA|\widetilde{\Omega}_k|$. Further noticing that

$$\sum_{k=1}^K |\widetilde{\Omega}_k| = \sum_{k=1}^K \sum_{k'=1}^{k-1} \mathbb{1}[k' + d_{k'} \geq k] = \sum_{k'=1}^{K-1} \sum_{k=k'+1}^K \mathbb{1}[k' + d_{k'} \geq k] = \sum_{k'=1}^{K-1} d_{k'} = \mathfrak{D},$$

we have the cheating regret is bounded by

$$\mathbb{E}\left[\sum_{k=1}^K \sum_{h=1}^H \langle \mu_{\pi_k}^h(P_k) - \mu_{\widehat{\pi}_k}^h(\widehat{P}_k), \widehat{\ell}_k^h \rangle\right] \leq \eta H^2 SA\mathfrak{D} + 2\eta HSAK + 4\delta KHL.$$

The EstReg term is then consequently bounded by

$$\textsc{EstReg} \leq \frac{2H}{\eta}(1 + \ln(SA)) + 3\eta H^2 SAK + \eta H^2 SA\mathfrak{D} + 2\eta HSAK + 12\delta KHL,$$

which is at most $\frac{2H}{\eta}(1 + \ln(SA)) + 5\eta H^2 SAK + \eta H^2 SA\mathfrak{D} + 12\delta KHL$, as claimed. $\square$

# D  Analysis of Infinite-horizon AMDP Algorithms

## D.1  FTPL-Based Efficient Algorithm (Theorem 7)

In this section present our Algorithm 6 together with its analysis. As described in the main body, we will divide the time horizon $[T]$ into $J$ epochs and fix a policy $\pi_j$ for the $j$-th epoch, namely $\mathcal{T}_j = \{(j-1)H+1, (j-1)H+2, \ldots, jH\}$ where $H = \frac{T}{J}$ is the length of each epoch (overloading the notation $H$ from the episodic setting since they have a similar meaning).

### D.1.1  Switching Procedure

The most significant difference between infinite-horizon AMDPs and episodic AMDPs is that the agent will not be reset to $s^1$ at the beginning of an "epoch". To formalize our problem as a online linear optimization problem (i.e., the total loss represented as $\sum_{t=1}^T \langle \mu_{\pi_t}^t, \ell_t \rangle$), we have to ensure the distribution over all states is exactly $\mu_{\pi_j}^t$ for most $t \in \mathcal{T}_j$. Before presenting the switching procesure from Chandrasekaran and Tewari (2021), we first restate the assumption together with several properties that they used. For the sake of completeness, we also include their proofs here.

**Assumption 28** (Existance of a Staying State, Restatement of Assumption 2 and Chandrasekaran and Tewari (2021, Assumption 5.1))**.** *The MDP $\mathcal{M}$ has a state $s^*$ and an action $a^*$ such that* $\mathbb{P}(s^* \mid s^*, a^*) = 1$.

**Lemma 29** (Chandrasekaran and Tewari (2021, Lemma 5.2))**.** *For any two distinct states $s, s' \in \mathcal{S}$, there exists a policy $\pi_{s,s'}$ and $l_{s,s'} \leq 2D$ such that*

$$\Pr\{T(s' \mid \pi_{s,s'}, s) = l_{s,s'}\} \geq \frac{1}{4D}.$$

*Proof.* By definition of diameter (as in Definition 1), there exists a policy $\pi_{s,s'}$ such that $\mathbb{E}[T(s' \mid \mathcal{M}, \pi_{s,s'}, s)] \leq D$. By Markov's inequality, this implies $\Pr\{T(s' \mid \mathcal{M}, \pi_{s,s'}, s)] \leq 2D\} \geq \frac{1}{2}$. By pigeonhole principle, there consequently exists $l_{s,s'} \leq 2D$ such that $\Pr\{T(s' \mid \mathcal{M}, \pi_{s,s'}, s) = l_{s,s'}\} \geq \frac{1}{2} \cdot \frac{1}{2D} = \frac{1}{4D}$. $\square$

**Lemma 30** (Chandrasekaran and Tewari (2021, Theorem 5.3))**.** *For an MDP that satisfies Assumption 28, there exists $l^* \leq 2D$ such that for all states $s' \neq s^*$, there exists policy $\pi_{s'}$ such that*

$$\Pr\{T(s' \mid \mathcal{M}, \pi_{s'}, s^*) = l^*\} \geq \frac{1}{4D}.$$

**Algorithm 5** Policy Switching in Infinite-Horizon AMDP (Chandrasekaran and Tewari, 2021)

**Require:** Current state $s \in \mathcal{S}$. Goal policy $\pi \in \Pi$. Current time $t$.
1: **while true do**
2:     Move to state $s^*$ using policy $\pi_{s,s^*}$ as defined by Lemma 29 and update $s, t$ concurrently.
3:     Sample the target state $g \sim \mu_\pi^t$.
4:     Use policy $\pi_g$ from Lemma 30 to move $l^*$ steps from $s^*$ and update $s, t$ concurrently.
5:     **if** $s = g$ **then**
6:         Sample a Bernoulli random variable $I \sim \text{Ber}(\frac{p^*}{p_g})$.
7:         **if** $I = 1$ **then**
8:             **return**
                                               ▷ The while loop will repeat if $s \neq g$ or $I \neq 1$.

---

*Furthermore, denote $p_{s'}$ as the probability above. Let $p^* = \min_{s \in \mathcal{S}} p_s$. Then $p^* \geq \frac{1}{4D}$.*

*Proof.* From the previous lemma, there exists an $l_{s'} \leq 4D$ for all $s' \neq s^*$ such that there is a policy $\pi_{s^*, s'}$ hitting $s'$ from $s^*$ in time exactly $l_{s'}$ with probability at least $\frac{1}{4D}$. Let $l^* = \max_{s' \neq s^*} l_{s'}$ and $\pi_{s'}$ be the policy that first stays at $s^*$ for $(l^* - l_{s'})$ steps and then follows $\pi_{s'}$ for $l_{s'}$ steps suffices.     □

Now we are able to present the switching procedure from Chandrasekaran and Tewari (2021), as in Algorithm 5.

**Theorem 31** (Correctness of Algorithm 5, Chandrasekaran and Tewari (2021, Lemma 5.6)). *Let the random variable denoting the time that Algorithm 5 terminates be $t_{switch}$. Then for any state $s \in \mathcal{S}$*

$$\Pr\{s_t = s \mid t_{switch} = t\} = \mu_\pi^t(s), \quad \forall t \in [T].$$

*Proof.* The key idea is to write

$$\Pr\{s_t = s \mid t_{\texttt{switch}} = t\} = \frac{\Pr\{s_t = s, g = s, t_{\texttt{switch}} = t\}}{\Pr\{t_{\texttt{switch}} = t\}}$$

and then bound the numerator and denominator separately. For the denominator,

$$
\begin{aligned}
&\Pr\{t_{\texttt{switch}} = t\} \\
&= \sum_{s \in \mathcal{S}} \Pr\{s_t = s, g = s, s_{t-l^*} = s^*\} \times \Pr\{t_{\texttt{switch}} = t \mid s_t = s, g = s, s_{t-l^*} = s^*\} \\
&= \sum_{s \in \mathcal{S}} \Pr\{s_t = s \mid g = s, s_{t-l^*} = s^*\} \times \Pr\{g = s, s_{t-l^*} = s^*\} \times \\
&\qquad \Pr\{t_{\texttt{switch}} = t \mid s_t = s, g = s, s_{t-l^*} = s^*\} \\
&= \sum_{s \in \mathcal{S}} p_s \times \Pr\{g = s, s_{t-l^*} = s^*\} \times \frac{p^*}{p_s} = p^* \times \Pr\{s_{t-l^*} = s^*\},
\end{aligned}
$$

where the last step used definition of $p_s$ and $I$. For the numerator,

$$
\begin{aligned}
&\Pr\{g = s, s_t = s, t_{\text{switch}} = t\} \\
&= \Pr\{g = s, s_t = s, s_{t-l^*} = s^*, t_{\text{switch}} = t\} \\
&= \Pr\{t_{\text{switch}} = t, s_t = s \mid g = s, s_{t-l^*} = s^*\} \times \Pr\{g = s, s_{t-l^*} = s^*\} \\
&= \Pr\{s_t = s \mid g = s, s_{t-l^*} = s^*\} \times \Pr\{t_{\text{switch}} = t \mid s_t = s, g = s, s_{t-l^*} = s^*\} \times \\
&\qquad \Pr\{s_{t-l^*} = s^*\} \times \Pr\{g = s \mid s_{t-l^*} = s^*\} \\
&= p_s \times \frac{p^*}{p_s} \times \Pr\{s_{t-l^*} = s^*\} \times \mu_{\pi'}^t(s).
\end{aligned}
$$

Plugging them back gives our desired result.     □

**Theorem 32** (Efficiency of Algorithm 5, Chandrasekaran and Tewari (2021, Lemma 5.7)). *The expected time spent on Algorithm 5 is bounded by $12D^2$ for each execution.*

**Algorithm 6** FTPL for Infinite-horizon AMDPs with Bandit Feedback and Known Transition

**Require:** Laplace distribution parameter $\eta$. Geometric Re-sampling parameter $L$.
1: Sample perturbations $\{z^t \in \mathbb{R}^{\mathcal{S} \times \mathcal{A}}\}_{t \in [T]}$ where $z^t(s, a) \sim \text{Laplace}(\eta)$.
2: **for** $j = 1, 2, \ldots, J$ **do**
3:      Calculate the policy $\pi_j$ for this epoch as

$$\pi_j = \operatorname*{argmin}_{\pi \in \Pi} \left( \sum_{j'=1}^{j-1} \sum_{t \in \mathcal{T}_{j'}} \langle \mu_\pi^t, \widehat{\ell}^t \rangle + \sum_{t=1}^{T} \langle \mu_\pi^t, z^t \rangle \right). \tag{21}$$

4:      Execute Algorithm 5 with parameters $s^t, \pi_j, t$ (note that Algorithm 5 will update $t$ internally).
5:      **for** All remaining time slots in $\mathcal{T}_j$, i.e., $\mathcal{T}_j \cap [t, T]$ **do**
6:          Play $a^t = \pi_j(s^t)$, observe the loss $\ell^t(s^t, a^t)$ and the next state $s^{t+1} \in \mathcal{S}$.
7:          **for** $M^t = 1, 2, \ldots, L$ **do**
8:              Resample a fresh perturbation and get new policy $\pi_j'$ from Eq. (21).
9:              Draw a sample from $\text{Ber}(\mu_{\pi_j}^t(s^t, a^t; \mathbb{P}))$. If it is 1 or $M^t = L$, terminate and set

$$\widehat{\ell}^t(s, a) = \mathbb{1}[(s, a) = (s^t, a^t)] \ell^t(s^t, a^t) M^t, \quad \forall (s, a) \in \mathcal{S} \times \mathcal{A}.$$

*Proof.* Every time we try to catch the policy from $s^*$, we succeed with probability $p^* \geq \frac{1}{4D}$. Thus, the expected number of times we try is $4D$ and each attempt takes $l^* \leq 2D$ steps. Between each of these attempts, we move at most D steps in expectation to reach $s^*$ again. Thus, in total, we have

$$\mathbb{E}[t_{\text{switch}} - t_0] \leq 4D(2D + D) \leq 12D^2,$$

as claimed. $\qquad\qquad\qquad\qquad\qquad\qquad\qquad\qquad\qquad\qquad\qquad\qquad\qquad\qquad\qquad\qquad\qquad\qquad\qquad$ $\square$

### D.1.2    The Algorithm

With the help of Algorithm 5, we now present our algorithm, Algorithm 6. As mentioned in the main text, another important difference due to the "non-resetting" nature of an infinite-horizon AMDP is that, we have to generate $T$ perturbations $z^1, z^2, \ldots, z^T$, whereas only $H$ perturbations is needed in the episodic settings. For each FTPL update, we will include all of them in the argmin operation, as in Eq. (21).

This difference can be explained from the contextual bandits' point of view (c.f. Appendix B.1.5). In infinite-horizon AMDPs, the possible number of "contexts" is now $T$, as for each policy $\pi$, it will have $T$ distinct features $\mu_\pi^1, \mu_\pi^2, \ldots, \mu_\pi^T$. In contrast, for episodic AMDPs, there are only $H$ different contexts as only $\{\mu_\pi^h\}_{h=1}^H$ can appear. Therefore, as noticed by Syrgkanis et al. (2016), we have to add perturbations to each of the contexts, which are in total $T$ of them.

### D.1.3    Proof of Main Theorem

*Proof of Theorem 7.* To calculate the regret guarantee of Algorithm 6, we consider the following quantity $\widetilde{\mathcal{R}_T}$ defined as if there is no cost for a policy switching. By Theorem 32, there can be at most $JD^2$ time slots spent on executing Algorithm 5. Henceforth, the difference between $\mathcal{R}_T$ and $\widetilde{\mathcal{R}_T}$ is at most $JD^2$.

$$\widetilde{\mathcal{R}_T} \triangleq \mathbb{E} \left[ \sum_{t=1}^{T} \langle \mu_{\pi_{j(t)}}^t, \ell^t \rangle - \langle \mu_{\pi^*}^t, \ell^t \rangle \right] = \mathbb{E} \left[ \sum_{j=1}^{J} \sum_{\pi \in \Pi} p_j(\pi) \sum_{t \in \mathcal{T}_j} \langle \mu_\pi^t, \ell^t \rangle - \sum_{t=1}^{T} \langle \mu_{\pi^*}^t, \ell^t \rangle \right], \quad (22)$$

where $p_j(\pi)$ is the probability of picking $\pi$ w.r.t. $z$, conditioning on $\mathcal{F}_{(j-1)H}$ and $j(t)$ is the epoch that $t$ belongs to, namely $j(t) = \lceil \frac{j}{H} \rceil$. Then, we can decompose $\widetilde{\mathcal{R}_T}$ into three terms exactly the same as what we did in Appendix B.1:

$$\widetilde{\mathcal{R}_T} = \mathbb{E} \underbrace{\left[ \sum_{t=1}^{T} \langle \mu_{\pi_{j(t)}}^t, \ell^t - \widehat{\ell}^t \rangle + \sum_{t=1}^{T} \langle \mu_{\pi^*}^t, \widehat{\ell}^t - \ell^t \rangle \right]}_{\text{GR error term}} +$$

$$\underbrace{\mathbb{E}\left[\sum_{j=1}^{J}\sum_{\pi\in\Pi}p_{j+1}(\pi)\sum_{t\in\mathcal{T}_j}\langle\mu_\pi^t-\mu_{\pi^*}^t,\widehat{\ell}^t\rangle\right]}_{\text{Error term}}+$$

$$\underbrace{\mathbb{E}\left[\sum_{j=1}^{J}\sum_{\pi\in\Pi}(p_j(\pi)-p_{j+1}(\pi))\sum_{t\in\mathcal{T}_j}\langle\mu_\pi^t,\widehat{\ell}^t\rangle\right]}_{\text{Stability term}}.$$

The GR error term is quite similar to Appendix B.1:

**Lemma 33.** *The GR error term is bounded by*

$$\mathbb{E}\left[\sum_{t=1}^{T}\langle\mu_{\pi_{j(t)}}^t,\ell^t-\widehat{\ell}^t\rangle+\sum_{t=1}^{T}\langle\mu_{\pi^*}^t,\widehat{\ell}^t-\ell^t\rangle\right]\leq\frac{SAT}{eL}.$$

For the error term, we still use the similar "be-the-leader" analysis as Lemma 10, except for we are now facing a slightly different $V$-function (which is defined for infinite-horizon). Moreover, as mentioned in the main text, we are using a different bound when facing $T$ different perturbations. As a result, we will have worse dependency on $S$ and $A$, but with better dependency on the number of contexts, which is $T$ here (and is $H$ in episodic settings). The result is stated as follows:

**Lemma 34.** *The error term is bounded by*

$$\mathbb{E}\left[\sum_{j=1}^{J}\sum_{\pi\in\Pi}p_{j+1}(\pi)\sum_{t\in\mathcal{T}_j}\langle\mu_\pi^t-\mu_{\pi^*}^t,\widehat{\ell}^t\rangle\right]\leq\frac{10}{\eta}S\sqrt{AT\ln A}.$$

For the stability term, again much similar to Appendix B.1, we have

**Lemma 35.** *The stability term is bounded by*

$$\mathbb{E}\left[\sum_{j=1}^{J}\sum_{\pi\in\Pi}(p_{j+1}(\pi)-p_j(\pi))\sum_{t\in\mathcal{T}_j}\langle\mu_\pi^t,\widehat{\ell}^t\rangle\right]\leq2\eta H^2SAJ.$$

Therefore, our regret is bounded by

$$\mathcal{R}_T\leq\widetilde{\mathcal{R}_T}+JD^2\leq\frac{SAT}{eL}+\frac{10}{\eta}S\sqrt{AT\ln A}+2\eta H^2SAJ+JD^2.$$

Picking $\eta=S^{1/3}D^{-2/3}T^{-1/3}$, $J=S^{2/3}A^{1/2}D^{-4/3}T^{5/6}$ and $L=S^{1/3}A^{1/2}D^{-2/3}T^{1/6}$ gives $\mathcal{R}_T=\widetilde{\mathcal{O}}\left(S^{2/3}A^{1/2}D^{2/3}T^{5/6}\right)$. $\square$

*Proof of Lemma 33.* We follow the proof of Lemma 9 by replacing $K$ with $J$ and the GR estimator $\widehat{\ell}_k^h$ with $\widehat{\ell}_t$. First notice that, from Lemma 38, $\mathbb{E}[\widehat{\ell}_t(s,a)\mid\mathcal{F}_{(j-1)H}]\leq\ell^t(s,a)$ for all $t\in\mathcal{T}_j$. Moreover, as $\pi^*$ is deterministic (i.e., it does not depend on the randomness from the algorithm), the term related to $\mu_{\pi^*}^t$ is bounded by

$$\mathbb{E}\left[\sum_{j=1}^{J}\sum_{t\in\mathcal{T}_j}\langle\mu_{\pi^*}^t,\widehat{\ell}^t-\ell^t\rangle\right]=\mathbb{E}\left[\sum_{j=1}^{J}\sum_{t\in\mathcal{T}_j}\langle\mu_{\pi^*}^h,\mathbb{E}[\widehat{\ell}^t\mid\mathcal{F}_{(j-1)H}]-\ell^t\rangle\right]\leq0.$$

For the first term, again by Lemma 38, we have

$$\mathbb{E}\left[\sum_{j=1}^{J}\sum_{t\in\mathcal{T}_j}\langle\mu_{\pi_j}^t,\ell^t-\widehat{\ell}^t\rangle\right]=\sum_{j=1}^{J}\sum_{t\in\mathcal{T}_j}\sum_{(s,a)\in\mathcal{S}\times\mathcal{A}}\mathbb{E}\left[\mu_{\pi_j}^t(s,a)\cdot(1-q^t(s,a))^L\ell^t(s,a)\right],$$

where $q^t(s,a)$ is the probability of visiting $(s,a)$ in a single execution of the Geometric Re-sampling process, which is just

$$q^t(s,a) = \mathbb{E}[\mu_{\pi_j}^t(s,a)] = \sum_{\pi \in \Pi} p_j(\pi) \mu_\pi^t(s,a)$$

in our case. By noticing that $q(1-q)^L \leq q e^{-Lq} \leq \frac{1}{eL}$ (Neu and Bartók, 2013), we have

$$\mathbb{E}\left[\sum_{j=1}^J \sum_{t \in \mathcal{T}_j} \langle \mu_{\pi_j}^t, \ell^t - \widehat{\ell}^t \rangle\right] \leq HJSA\frac{1}{eL} = \frac{SAT}{eL},$$

as claimed. □

*Proof of Lemma 34.* The proof still uses the standard "be-the-leader" technique, but in a slightly different manner as we are adding perturbations to all time indices. Instead, we follow the idea of Syrgkanis et al. (2016, Lemma 7) and prove by induction that the following inequality holds for all $J$ and any policy $\pi \in \Pi$:

$$\sum_{t=1}^T \langle \mu_{\pi_1}^t, z^t \rangle + \sum_{j=1}^J \sum_{t \in \mathcal{T}_j} \langle \mu_{\pi_{j+1}}^t, \ell^t \rangle \leq \sum_{t=1}^T \langle \mu_\pi^t, z^t \rangle + \sum_{j=1}^J \sum_{t \in \mathcal{T}_j} \langle \mu_\pi^t, \ell^t \rangle.$$

Obviously, for $J = 0$, this inequality holds. Suppose that this inequality holds for $J$, then we consider $J + 1$. Let $\pi = \pi_{J+2}$. Adding $\sum_{t \in \mathcal{T}_{J+1}} \langle \mu_{\pi_{J+2}}^t, \widehat{\ell}^t \rangle$ to both sides gives

$$\sum_{t=1}^T \langle \mu_{\pi_1}^t, z^t \rangle + \sum_{j=1}^{J+1} \sum_{t \in \mathcal{T}_j} \langle \mu_{\pi_{j+1}}^t, \ell^t \rangle \leq \sum_{t=1}^T \langle \mu_{\pi_{J+2}}^t, z^t \rangle + \sum_{j=1}^{J+1} \sum_{t \in \mathcal{T}_j} \langle \mu_{\pi_{J+2}}^t, \ell^t \rangle.$$

However, by definition of $\pi_{J+2}$ (which is the argmin of the right-handed-side for all policies), it is further bounded by

$$\sum_{t=1}^T \langle \mu_{\pi_1}^t, z^t \rangle + \sum_{j=1}^{J+1} \sum_{t \in \mathcal{T}_j} \langle \mu_{\pi_{j+1}}^t, \ell^t \rangle \leq \sum_{t=1}^T \langle \mu_\pi^t, z^t \rangle + \sum_{j=1}^{J+1} \sum_{t \in \mathcal{T}_j} \langle \mu_\pi^t, \ell^t \rangle$$

for any policy $\pi \in \Pi$, which means that the induction hypothesis for $J + 1$. Therefore, by picking $\pi = \pi^*$ for the real $J$, we can conclude that

$$\sum_{j=1}^J \sum_{t \in \mathcal{T}_j} \langle \mu_{\pi_{j+1}}^t, \ell^t \rangle - \sum_{j=1}^J \sum_{t \in \mathcal{T}_j} \langle \mu_{\pi^*}^t, \ell^t \rangle \leq \sum_{t=1}^T \langle \mu_{\pi^*}^t, z^t \rangle - \sum_{t=1}^T \langle \mu_{\pi_1}^t, z^t \rangle.$$

Then taking expectation on both sides gives the error term is bounded by

$$\mathbb{E}_z\left[\max_{\pi \in \Pi} \sum_{t=1}^T \langle \mu_\pi^t, z^t \rangle - \min_{\pi \in \Pi} \sum_{t=1}^T \langle \mu_\pi^t, z^t \rangle\right],$$

which is bounded by $\frac{10}{\eta}\sqrt{TSA \cdot \ln|\Pi|} = \frac{10}{\eta}S\sqrt{AT \ln A}$ by Lemma 45 (note that as $\ln|\Pi| = S \ln A < SAT$, the condition of applying Lemma 45 indeed holds). □

*Proof of Lemma 35.* This follows directly from Lemma 12 with some slight modifications as well. For clarity, we rewrite the full proof here.

We first give the single-step stability lemma for infinite-horizon AMDPs, whose proof will be presented later:

**Lemma 36.** *For all $j \in [J]$ and $(s,a) \in \mathcal{S} \times \mathcal{A}$,*

$$p_{j+1}(\pi) \geq p_j(\pi) \exp\left(-\eta \sum_{t \in \mathcal{T}_j} \|\widehat{\ell}^t\|_1\right), \quad \forall \pi \in \Pi.$$

By summing up Lemma 36 for all $\pi \in \Pi$ and using the fact that $1 - \exp(-x) \le x$, we have

$$\sum_{\pi \in \Pi} (p_j(\pi) - p_{j+1}(\pi)) \sum_{t \in \mathcal{T}_j} \langle \mu_\pi^t, \widehat{\ell}^t \rangle \le \eta \sum_{t' \in \mathcal{T}_j} \|\widehat{\ell}^{t'}\|_1 \cdot \sum_{\pi \in \Pi} p_j(\pi) \sum_{t \in \mathcal{T}_j} \langle \mu_\pi^t, \widehat{\ell}^t \rangle, \quad \forall j \in [J]. \quad (23)$$

Again noticing that $M^t = \min\{\mathrm{Geo}(q^t(s^t, a^t)), L\}$ where $q^t(s, a) = \sum_{\pi \in \Pi} p_j(\pi) \mu_\pi^t(s, a)$ if $t \in \mathcal{T}_j$. Then calculate the expectation of $\widehat{\ell}^t(s, a)$ only with respect to $M^t$, we will have

$$\mathbb{E}\left[\widehat{\ell}^t(s, a) \middle| (s^t, a^t) = (s, a)\right] \le \frac{\ell^t(s, a)}{q^t(s, a)}.$$

Let $\mathbb{1}^t(s, a)$ be the shorthand notation of $\mathbb{1}[(s^t, a^t) = (s, a)]$. Then for those $t' \ne t$ in Eq. (23),

$$\eta \mathbb{E}\left[\sum_{t \in \mathcal{T}_j} \sum_{s,a} \sum_{\pi \in \Pi} p_j(\pi) \mu^t(s, a) \widehat{\ell}^t(s, a) \sum_{t' \ne t} \|\widehat{\ell}^{t'}\|_1 \middle| \mathcal{F}_{(j-1)H}\right]$$

$$\overset{(a)}{\le} \eta \mathbb{E}\left[\sum_{h=1}^H \sum_{s,a} \mathbb{1}^t(s, a) \ell^t(s, a) \frac{\sum_{\pi \in \Pi} p_j(\pi) \mu_\pi^t(s, a)}{q^t(s, a)} \sum_{t' \ne t} \|\widehat{\ell}^{t'}\|_1 \middle| \mathcal{F}_{(j-1)H}\right]$$

$$\overset{(b)}{\le} \eta H \mathbb{E}\left[\sum_{t' \in \mathcal{T}_j} \sum_{s,a} \|\widehat{\ell}^{t'}\|_1 \middle| \mathcal{F}_{(j-1)H}\right] \overset{(c)}{\le} \eta H^2 SA.$$

where (a) is taking expectation w.r.t. $M^t$, (b) used the definition of $q^t$ together with the fact that $\sum_{(s,a)} \mathbb{1}^t(s, a) = 1$, and (c) used the fact that $\mathbb{E}[\widehat{\ell}^{t'}(s, a) \mid \mathcal{F}_{(j-1)H}] \le \ell^{t'}(s, a) \le 1$ (Lemma 38).

For those terms with $t' = t$ in Eq. (23), by direct calculation and the fact that $\widehat{\ell}^t$ is a one-hot vector, we can bound them as

$$\eta \mathbb{E}\left[\sum_{t \in \mathcal{T}_j} \sum_{s,a} \sum_{\pi \in \Pi} p_j(\pi) \mu_\pi^t(s, a) \left(\widehat{\ell}^t(s, a)\right)^2 \middle| \mathcal{F}_{(j-1)H}\right] \le 2\eta \mathbb{E}\left[\sum_{h,s,a} \frac{q^t(s, a)}{q^t(s, a)} \middle| \mathcal{F}_{(j-1)H}\right] \le 2\eta HSA$$

by noticing $\mathbb{E}[(\widehat{\ell}^t(s, a))^2 \mid \mathcal{F}_{(j-1)H}] \le 2(q^t(s, a))^{-1}$ (Lemma 39). Combining the terms with $t' \ne t$ and the ones with $t' = t$ gives our conclusion. □

*Proof of Lemma 36.* The proof will be similar to, but different from Lemma 11, as we are now adding different perturbations. We now use a slightly different definition of the best-function. Let $\pi = \mathrm{best}(\ell, z)$ where $\ell = \{\ell^1, \ell^2, \dots, \ell^m\}$ and $z = \{z^1, z^2, \dots, z^T\}$ to denote

$$\pi = \underset{\pi \in \Pi}{\mathrm{argmin}} \left(\sum_{t=1}^m \langle \mu_\pi^t, \ell^t \rangle + \sum_{t=1}^T \langle \mu_\pi^t, z^t \rangle\right).$$

Then we have

$$p_j(\pi) = \int_z \mathbb{1}\left[\pi = \mathrm{best}\left(\{\widehat{\ell}^1, \dots, \widehat{\ell}^{(j-1)H}\}, \{z^1, z^2, \dots, z^T\}\right)\right] f(z) \, \mathrm{d}z$$

$$= \int_z \mathbb{1}\left[\pi = \mathrm{best}\left(\{\widehat{\ell}^1, \dots, \widehat{\ell}^{(j-1)H}\}, \{z^1, \dots, z^{(j-1)H}, z^{(j-1)H+1} + \widehat{\ell}^{(j-1)H+1}, \dots, z^{jH} + \widehat{\ell}^{jH}, z^{jH+1}, \dots, z^T\}\right)\right]$$

$$f\left(z + \{0, \dots, 0, \widehat{\ell}^{(j-1)H+1}, \dots, \widehat{\ell}^{jH}, 0, 0, \dots, 0\}\right) \, \mathrm{d}z$$

$$= \int_z \mathbb{1}\left[\pi = \mathrm{best}\left(\{\widehat{\ell}^1, \dots, \widehat{\ell}^{(j-1)H}, \widehat{\ell}^{(j-1)H+1}, \dots \widehat{\ell}^{jH}\}, z\right)\right]$$

$$f\left(z + \{0, \dots, 0, \widehat{\ell}^{(j-1)H+1}, \dots, \widehat{\ell}^{jH}, 0, 0, \dots, 0\}\right) \, \mathrm{d}z,$$

---

**Algorithm 7** Hedge for Infinite-horizon AMDPs with Bandit Feedback and Known Transition

---

**Require:** Learning rate $\eta$. Number of epochs $J$.

1: **for** $j = 1, 2, \ldots, J$ **do**
2:     Calculate the distribution of policies for the $j$-th epoch as

$$p_j(\pi) \propto \exp\left(-\eta \sum_{t=1}^{(j-1)H} \langle \mu_\pi^t, \widehat{\ell}^t \rangle \right). \tag{24}$$

3:     Sample the policy $\pi_j \sim p_j$ for this epoch.
4:     Execute Algorithm 5 with parameters $s^t, \pi_j, t$ (note that Algorithm 5 will update $t$ internally).
5:     **for** All remaining $\mathcal{T}_j \cap [t, T]$ **do**
6:         Play $a^t = \pi_j(s^t)$, observe loss $\ell^t(s^t, a^t)$ and the next state $s^{t+1}$. Set

$$\widehat{\ell}^t(s, a) = \mathbb{1}[(s, a) = (s^t, a^t)] \frac{\ell^t(s^t, a^t)}{\sum_{\pi \in \Pi} p_j(\pi) \mu_\pi^t(s, a)}, \quad \forall (s, a) \in \mathcal{S} \times \mathcal{A}.$$

---

where $f(z)$ is the probability density function of $z$ and the second step makes use of the fact that $z + \widehat{\ell}^j$ is still linear in $z$. Moreover,

$$p_{j+1}(\pi) = \int_z \mathbb{1}\left[\pi = \text{best}\left(\{\widehat{\ell}^1, \ldots, \widehat{\ell}^{jH}\}, z\right)\right] f(z) \, \mathrm{d}z.$$

For simplicity, denote $\widetilde{\ell}^j = \{0, \ldots, 0, \widehat{\ell}^{(j-1)H+1}, \ldots, \widehat{\ell}^{jH}, 0, 0, \ldots, 0\} = \{\mathbb{1}[t \in \mathcal{T}_j]\widehat{\ell}^t\}_{t=1}^T$. Again using the fact that $f(z) = \prod_{h=1}^H \exp(-\eta\|z^h\|_1)$, we have

$$f\left(z + \widetilde{\ell}^j\right) = \prod_{t \in \mathcal{T}_j} \exp\left(-\eta\left(\|z^t + \widehat{\ell}^t\|_1 - \|z^t\|\right)\right) f(z),$$

which gives

$$\frac{f\left(z + \widetilde{\ell}^j\right)}{f(z)} \in \left[\exp\left(-\eta \sum_{t \in \mathcal{T}_j} \|\widehat{\ell}^t\|_1\right), \exp\left(\eta \sum_{t \in \mathcal{T}_j} \|\widehat{\ell}^t\|_1\right)\right]$$

by triangle inequality. Therefore, $p_{j+1}(\pi)/p_j(\pi)$ lies in this interval as well, which is just our claim. $\qquad\square$

## D.2   Hedge-Based Inefficient Algorithm (Theorem 8)

In this section, we present our Hedge-based inefficient algorithm for infinite-horizon AMDPs with bandit feedback and known transitions. We still use the same epoching mechanism as Algorithm 6.

For Hedge, which is different from FTPL, we will *explicitly* maintain a distribution $p_j \in \triangle(\Pi)$ over all policies for each epoch, and randomly draw one $\pi_j \sim p_j$ for the $j$-th epoch. As the distribution $p_j$ can be directly calculated (we do not care about computational efficiency now), we can use importance weighting estimator to estimate the losses. The algorithm is presented in Algorithm 7.

*Proof of Theorem 8.* As Appendix D.1, we still define $\widetilde{\mathcal{R}_T}$ as Eq. (22). We can still conclude that $\mathcal{R}_T \leq \widetilde{\mathcal{R}_T} + JD^2$. We first show that the importance weighting estimator is indeed unbiased. Notice that the probability of visiting $(s, a)$ at some slot $t \in \mathcal{T}_j$ is exactly $\sum_{\pi \in \Pi} p_j(\pi)\mu_\pi^t(s, a)$, which means, by Lemma 42, we have

$$\mathbb{E}\left[\widehat{\ell}^t(s, a)\Big|\mathcal{F}_{(j-1)H}\right] = \ell^t(s, a), \quad \forall (s, a) \in \mathcal{S} \times \mathcal{A}, t \in \mathcal{T}_j, j \in [J].$$

Let $\widetilde{\ell}_j(\pi)$ be the random variable denoting the total loss of policy $\pi$ for epoch $j$:

$$\widetilde{\ell}_j(\pi) = \sum_{t \in \mathcal{T}_j} \langle \mu_\pi^t, \widehat{\ell}^t \rangle.$$

So Eq. (24) is just $p_j(\pi) \propto \exp(-\eta \sum_{j'=1}^{j-1} \widetilde{\ell}_{j'}(\pi))$. Therefore, by standard properties of Hedge (Lemma 37), for any realization of $\{\widetilde{\ell}_j\}_{j \in [J]}$ (and also $\{\widehat{\ell}^t\}_{t \in [T]}$), we will have

$$\sum_{j=1}^J \langle p_j, \widetilde{\ell}_j \rangle - \sum_{j=1}^J \widetilde{\ell}_j(\pi^*) \leq \frac{\ln |\Pi|}{\eta} + \eta \sum_{j=1}^J \sum_{\pi \in \Pi} p_j(\pi) \widetilde{\ell}_j^2(\pi). \tag{25}$$

Consider the second term of the right-handed-side. For a fixed $j \in [J]$, it becomes

$$\sum_{\pi \in \Pi} p_j(\pi) \widetilde{\ell}_j^2(\pi) = \sum_{\pi \in \Pi} p_j(\pi) \left( \sum_{t \in \mathcal{T}_j} \sum_{(s,a) \in \mathcal{S} \times \mathcal{A}} \mu_\pi^t(s,a) \widehat{\ell}^t(s,a) \right)^2$$

$$\leq H \sum_{\pi \in \Pi} p_j(\pi) \sum_{t \in \mathcal{T}_j} \left( \sum_{(s,a) \in \mathcal{S} \times \mathcal{A}} \mu_\pi^t(s,a) \widehat{\ell}^t(s,a) \right)^2$$

$$= H \sum_{\pi \in \Pi} p_j(\pi) \sum_{t \in \mathcal{T}_j} \sum_{(s,a) \in \mathcal{S} \times \mathcal{A}} \left( \mu_\pi^t(s,a) \widehat{\ell}^t(s,a) \right)^2,$$

where the first inequality made use of Cauchy-Schwartz inequality while the second equality used the fact that $\widehat{\ell}_t$ is one-hot. Plugging back into Eq. (25) and taking expectation on both sides,

$$\mathbb{E}\left[ \sum_{j=1}^J \langle p_j, \widetilde{\ell}_j \rangle - \sum_{j=1}^J \widetilde{\ell}_j(\pi^*) \right]$$

$$\overset{(a)}{\leq} \frac{S \ln A}{\eta} + H \sum_{j=1}^J \mathbb{E}\left[ \sum_{\pi \in \Pi} p_j(\pi) \sum_{t \in \mathcal{T}_j} \sum_{(s,a) \in \mathcal{S} \times \mathcal{A}} \mu_\pi^t(s,a) \left( \widehat{\ell}^t(s,a) \right)^2 \middle| \mathcal{F}_{(j-1)H} \right]$$

$$\overset{(b)}{\leq} \frac{S \ln A}{\eta} + \eta H \sum_{j=1}^J \mathbb{E}\left[ \sum_{\pi \in \Pi} p_j(\pi) \sum_{t \in \mathcal{T}_j} \sum_{(s,a) \in \mathcal{S} \times \mathcal{A}} \mu_\pi^t(s,a) \cdot \frac{1}{\sum_{\pi \in \Pi} p_j(\pi) \mu_\pi^t(s,a)} \middle| \mathcal{F}_{(j-1)H} \right]$$

$$= \frac{S \ln A}{\eta} + \eta H^2 JSA = \frac{S \ln A}{\eta} + \eta HSAT,$$

where (a) used $\mu_\pi^t(s,a) \leq 1$ and (b) used Lemma 43. Moreover, for the left-hand side, we have

$$\mathbb{E}\left[ \sum_{j=1}^J \langle p_j, \widetilde{\ell}_j \rangle - \sum_{j=1}^J \widetilde{\ell}_j(\pi^*) \right] = \sum_{j=1}^J \mathbb{E}\left[ \sum_{\pi \in \Pi} p_j(\pi) \sum_{t \in \mathcal{T}_j} \langle \mu_\pi^t - \mu_{\pi^*}^t, \widehat{\ell}^t \rangle \middle| \mathcal{F}_{(j-1)H} \right].$$

By using Lemma 42, this is exactly

$$\sum_{j=1}^J \mathbb{E}\left[ \sum_{\pi \in \Pi} p_j(\pi) \sum_{t \in \mathcal{T}_j} \langle \mu_\pi^t - \mu_{\pi^*}^t, \widehat{\ell}^t \rangle \middle| \mathcal{F}_{(j-1)H} \right] = \mathbb{E}\left[ \sum_{j=1}^J \sum_{t \in \mathcal{T}_j} \langle \mu_{\pi_j}^t - \mu_{\pi^*}^t, \ell^t \rangle \right] = \widetilde{\mathcal{R}_T}.$$

Therefore, we will have

$$\mathcal{R}_T \leq \widetilde{\mathcal{R}_T} + JD^2 \leq \frac{S \ln A}{\eta} + \eta HSAT + JD^2,$$

which gives $\mathcal{R}_T = \widetilde{\mathcal{O}}\left( S^{2/3} A^{1/3} D^{2/3} T^{2/3} \right)$ when picking $J = S^{2/3} A^{1/3} D^{-4/3} T^{2/3}$ and $\eta = S^{1/3} A^{-1/3} D^{-2/3} T^{-2/3}$. $\qquad \square$

**Lemma 37** (Property of Hedge). *Suppose that we are using Hedge for $T$-round online learning problem that has $K$ actions, i.e., at time slot $t \in [T]$, picking $i_t$ according to the probability distribution $p_t \in \triangle([K])$ which is defined as:*

$$p_t(i) \propto \exp\left( -\eta \sum_{\tau=1}^{t-1} \ell_\tau(i) \right), \quad \forall i \in [K], t \in [T],$$

where $\ell_t(i) \geq 0$ is the non-negative loss associated with action $i$ at time slot $t$. Then, for all $i^* \in [K]$, we have

$$\sum_{t=1}^{T}(\langle p_t, \ell_t \rangle - \ell_t(i^*)) \leq \frac{\ln K}{\eta} + \eta \sum_{t=1}^{T}\sum_{i=1}^{K} p_t(i)\ell_t^2(i).$$

*Note that here we are considering non-randomized loss functions here.*

*Proof.* For simplicity, define $L_t(i)$ as $\sum_{\tau=1}^{t} \ell_\tau(i)$. Let

$$\Phi_t = \frac{1}{\eta} \ln\left(\sum_{i=1}^{K} \exp\left(-\eta L_t(i)\right)\right),$$

then

$$
\begin{aligned}
\Phi_t - \Phi_{t-1} &= \frac{1}{\eta} \ln\left(\frac{\sum_{i=1}^{K} \exp(-\eta L_t(i))}{\sum_{i=1}^{K} \exp(-\eta L_{t-1}(i))}\right) \\
&= \frac{1}{\eta} \ln\left(\sum_{i=1}^{K} p_t(i) \exp(-\eta \ell_t(i))\right) \\
&\overset{(a)}{\leq} \frac{1}{\eta} \ln\left(\sum_{i=1}^{K} p_t(i)(1 - \eta\ell_t(i) + \eta^2\ell_t^2(i))\right) \\
&= \frac{1}{\eta} \ln\left(1 - \eta\langle p_t, \ell_t\rangle + \eta^2 \sum_{i=1}^{K} p_t(i)\ell_t^2(i)\right) \\
&\overset{(b)}{\leq} -\langle p_t, \ell_t\rangle + \eta \sum_{i=1}^{K} p_t(i)\ell_t^2(i),
\end{aligned}
$$

where (a) used $\exp(-x) \leq 1 - x + x^2$ for all $x \geq 0$ and (b) used $\ln(1+x) \leq x$. Therefore, summing over $t$ gives

$$
\begin{aligned}
\sum_{t=1}^{T}\langle p_t, \ell_t\rangle &\leq \Phi_0 - \Phi_T + \eta \sum_{t=1}^{T}\sum_{i=1}^{N} p_t(i)\ell_t^2(i) \\
&\leq \frac{\ln N}{\eta} - \frac{1}{\eta} \ln\left(\exp(-\eta L_T(i^*))\right) + \eta \sum_{t=1}^{T}\sum_{i=1}^{N} p_t(i)\ell_t^2(i) \\
&\leq \frac{\ln N}{\eta} + L_T(i^*) + \eta \sum_{t=1}^{T}\sum_{i=1}^{N} p_t(i)\ell_t^2(i).
\end{aligned}
$$

Moving $L_T(i^*)$ to the left-handed-side then gives our conclusion. $\square$

# E  Auxiliary Lemmas

## E.1  Geometric Re-sampling Properties

In this section, we list two properties of the Geometric Re-sampling estimator (Neu and Bartók, 2013) that we used in the analysis. For the sake of completeness, we also include their proofs here.

**Lemma 38** (Neu and Bartók (2013, Lemma 1)). *Consider the Geometric Re-sampling estimator*

$$\widehat{\ell}_k^h(s,a) = \mathbb{1}[(s_k^h, a_k^h) = (s,a)] M_k^h(s,a) \ell_k^h(s,a). \tag{26}$$

*Let* $\Pr\{(s_k^h, a_k^h) = (s,a) \mid \mathcal{F}_{k-1}\} = q_k^h(s,a)$. *Suppose that the probability of visiting* $(s,a)$ *in the re-sampling process is also* $q_k^h(s,a)$, *then we have*

$$\mathbb{E}\left[\widehat{\ell}_k^h(s,a)\Big|\mathcal{F}_{k-1}\right] = \left(1 - (1 - q_k^h(s,a))^L\right)\ell_k^h(s,a).$$

*Proof.* By direct calculation, we have

$$\mathbb{E}\left[M_k^h(s,a)\big|\mathcal{F}_{k-1},(s_k^h,a_k^h)=(s,a)\right] = \sum_{n=1}^{\infty} n(1-q)^{n-1}q - \sum_{n=L}^{\infty} (n-L)(1-q)^{n-1}q$$

$$= \left(1-(1-q)^L\right)\sum_{n=1}^{\infty} n(1-q)^{n-1}q = \frac{1-(1-q)^L}{q}.$$

So we have

$$\mathbb{E}\left[\widehat{\ell}_k^h(s,a)\Big|\mathcal{F}_{k-1}\right] = \Pr\{(s_k^h,a_k^h)=(s,a)\mid\mathcal{F}_{k-1}\}\ell_k^h(s,a)\,\mathbb{E}\left[M_k^h(s,a)\big|\mathcal{F}_{k-1},(s_k^h,a_k^h)=(s,a)\right]$$
$$= \left(1-(1-q_k^h(s,a))^L\right)\ell_k^h(s,a),$$

as desired. $\square$

**Lemma 39.** *For the Geometric Re-sampling estimator as defined in the previous lemma, we have*

$$\mathbb{E}\left[(\widehat{\ell}_k^h(s,a))^2\Big|\mathcal{F}_{k-1}\right] \le \frac{2(\ell_k^h(s,a))^2}{q_k^h(s,a)}.$$

*Proof.* By definition, write

$$\mathbb{E}\left[(\widehat{\ell}_k^h(s,a))^2\Big|\mathcal{F}_{k-1}\right] = \mathbb{E}[\mathbb{1}[(s_k^h,a_k^h)=(s,a)]^2(\ell_k^h(s,a))^2(M_k^h(s,a))^2]$$
$$= \Pr\{(s_k^h,a_k^h)=(s,a)\}(\ell_k^h(s,a))^2\,\mathbb{E}\left[(M_k^h(s,a))^2\big|\mathcal{F}_{k-1},(s_k^h,a_k^h)=(s,a)\right]. \quad (27)$$

Simply write $q_k^h(s,a)$ as $q$. Note that $M_k^h(s,a) = \min\{L,\mathrm{Geo}(q)\}$, it is stochastically dominated by the geometric distribution with parameter $q$, whose second moment is bounded by

$$\mathbb{E}\left[(M_k^h(s,a))^2\big|\mathcal{F}_{k-1},(s_k^h,a_k^h)=(s,a)\right]$$
$$\le \mathbb{E}[(\mathrm{Geo}(q))^2] = \mathrm{Var}(\mathrm{Geo}(q)) + (\mathbb{E}[\mathrm{Geo}(q)])^2 = \frac{1-q}{q^2} + \frac{1}{q} \le \frac{2}{q^2}, \quad (28)$$

which means

$$\mathbb{E}\left[(\widehat{\ell}_k^h(s,a))^2\Big|\mathcal{F}_{k-1}\right] \le q(\ell_k^h(s,a))^2\frac{2}{q^2} = \frac{2(\ell_k^h(s,a))^2}{q},$$

as claimed. $\square$

**Corollary 40.** *Still consider the GR estimator defined in Eq. (26). Suppose that $\Pr\{(s_k^h,a_k^h) = (s,a) \mid \mathcal{F}_{k-1}\} = \widehat{q}_k^h(s,a)$ and the probability of visiting $(s,a)$ in each re-sampling procedure is $q_k^h(s,a)$ (where $\widehat{q}_k^h(s,a) \ne q_k^h(s,a)$). We then have*

$$\mathbb{E}\left[\widehat{\ell}_k^h(s,a)\Big|\mathcal{F}_{k-1}\right] = \frac{\widehat{q}_k^h(s,a)}{q_k^h(s,a)}\left(1-(1-q_k^h(s,a))^L\right)\ell_k^h(s,a).$$

*Proof.* The calculation of $\mathbb{E}[M_k^h(s,a) \mid \mathcal{F}_{k-1},(s_k^h,a_k^h)=(s,a)]$ is the same as the one in Lemma 38. Therefore,

$$\mathbb{E}\left[\widehat{\ell}_k^h(s,a)\Big|\mathcal{F}_{k-1}\right] = \Pr\{(s_k^h,a_k^h)=(s,a)\mid\mathcal{F}_{k-1}\}\ell_k^h(s,a)\,\mathbb{E}\left[M_k^h(s,a)\big|\mathcal{F}_{k-1},(s_k^h,a_k^h)=(s,a)\right]$$
$$= \frac{\widehat{q}_k^h(s,a)}{q_k^h(s,a)}\left(1-(1-q_k^h(s,a))^L\right)\ell_k^h(s,a),$$

as claimed. $\square$

**Corollary 41.** *Suppose the same condition as the previous corollary, i.e., still considering the GR estimator defined in Eq. (26) where $\Pr\{(s_k^h,a_k^h) = (s,a) \mid \mathcal{F}_{k-1}\} = \widehat{q}_k^h(s,a)$ and the probability of visiting $(s,a)$ in each re-sampling procedure is $q_k^h(s,a)$. We have*

$$\mathbb{E}\left[(\widehat{\ell}_k^h(s,a))^2\Big|\mathcal{F}_{k-1}\right] \le \frac{2(\ell_k^h(s,a))^2}{q_k^h(s,a)}\frac{\widehat{q}_k^h(s,a)}{q_k^h(s,a)}.$$

*Proof.* Still decompose the variance as Eq. (27). Still write $\widehat{q}_k^h(s,a)$ as $\widehat{q}$ and $q_k^h(s,a)$ as $q$. Then we still have $M_k^h(s,a) = \min\{L, \text{Geo}(q)\}$, which gives $\mathbb{E}[(M_k^h(s,a))^2 \mid \mathcal{F}_{k-1}, (s_k^h, a_k^h) = (s,a)] \leq \frac{2}{q^2}$ by Eq. (28). Therefore,

$$\mathbb{E}\left[(\widehat{\ell}_k^h(s,a))^2 \Big| \mathcal{F}_{k-1}\right] \leq \widehat{q}(\ell_k^h(s,a))^2 \frac{2}{q^2} = \frac{2(\ell_k^h(s,a))^2}{q_k^h(s,a)} \frac{\widehat{q}_k^h(s,a)}{q_k^h(s,a)},$$

as claimed. $\qquad\square$

### E.2  Importance Weighting Properties

**Lemma 42.** *For the Importance Weighting estimator*

$$\widehat{\ell}^t(s,a) = \mathbb{1}[(s^t, a^t) = (s,a)]\frac{\ell^t(s^t, a^t)}{\Pr\{(s^t, a^t) = (s,a) \mid \mathcal{F}\}}, \quad \forall (s,a) \in \mathcal{S} \times \mathcal{A},$$

*where $\mathcal{F}$ is a filtration, we will have*

$$\mathbb{E}[\widehat{\ell}^t(s,a) \mid \mathcal{F}] = \ell^t(s,a), \quad \forall (s,a) \in \mathcal{S} \times \mathcal{A}.$$

*Proof.* For simplicity, denote $q^t(s,a) = \Pr\{(s^t, a^t) = (s,a) \mid \mathcal{F}\}$. Then

$$\mathbb{E}[\widehat{\ell}^t(s,a) \mid \mathcal{F}] = q^t(s,a) \cdot \frac{\ell^t(s,a)}{q^t(s,a)} = \ell^t(s,a)$$

for all $(s,a) \in \mathcal{S} \times \mathcal{A}$. $\qquad\square$

**Lemma 43.** *For the same Importance Weighting Estimator, we will have*

$$\mathbb{E}\left[(\widehat{\ell}^t(s,a))^2 \Big| \mathcal{F}\right] = \frac{(\ell^t(s,a))^2}{q^t(s,a)}, \quad \forall (s,a) \in \mathcal{S} \times \mathcal{A},$$

*where $q^t(s,a) \triangleq \Pr\{(s^t, a^t) = (s,a) \mid \mathcal{F}\}$.*

*Proof.* Direct calculation gives $\mathbb{E}\left[(\widehat{\ell}^t(s,a))^2 \Big| \mathcal{F}\right] = q^t(s,a) \cdot \left(\frac{\ell^t(s,a)}{q^t(s,a)}\right)^2 = \frac{(\ell^t(s,a))^2}{q^t(s,a)}, \forall(s,a).$ $\quad\square$

### E.3  Auxiliary Lemmas for Error Terms

In this section, we present two lemmas that will play an important role when bounding the error terms (as used in Lemmas 10, 18 and 34).

**Lemma 44** (Wang and Dong (2020, Fact 2))**.** *Let $X_1, X_2, \ldots, X_n$ be i.i.d. random variables drawn from $Exp(\eta)$ which is the exponential distribution, then*

$$\mathbb{E}\left[\max_{1 \leq i \leq n} X_i\right] \leq \frac{1 + \ln n}{\eta}.$$

**Lemma 45** (Generalization of Syrgkanis et al. (2016, Lemma 8))**.** *Let $\{z^t \in \mathbb{R}^d\}_{t=1}^T$ be a sequence of $d$-dimensional random variable such that $z_i^t \sim Laplace(\eta)$ for all $i \in [m]$ and $t \in [T]$. Let $X$ be a set of sequences of the form $\{x^t \in [0,1]^d\}_{t=1}^T$. As long as $\ln|X| < dT$, we have*

$$\mathbb{E}_z\left[\max_{x \in X} \sum_{t=1}^T \langle x^t, z^t \rangle\right] - \mathbb{E}_z\left[\min_{x \in X} \sum_{t=1}^T \langle x^t, z^t \rangle\right] \leq \frac{10}{\eta}\sqrt{dT \ln|X|}.$$

*Proof.* Note that the key difference between this theorem and Syrgkanis et al. (2016, Lemma 8) is that, their theorem assumed a binary decision set, i.e., $x_i^t \in \{0, 1\}$ instead of $[0, 1]$. However, their proof still holds with only a little modification. The first step is still noticing that the distribution of Laplace

random variables is symmetric around 0, so we only need to bound $2\,\mathbb{E}_z\left[\max_{x\in X}\sum_{t=1}^T\langle x^t, z^t\rangle\right]$, which is bounded by, for any $\lambda \geq 0$,

$$
\begin{aligned}
\mathbb{E}_z\left[\max_{x\in X}\sum_{t=1}^T\langle x^t, z^t\rangle\right] &= \frac{1}{\lambda}\ln\left(\exp\left(\mathbb{E}_z\left[\max_{x\in X}\lambda\sum_{t=1}^T\langle x^t, z^t\rangle\right]\right)\right) \\
&\leq \frac{1}{\lambda}\ln\left(\mathbb{E}_z\left[\max_{x\in X}\exp\left(\lambda\sum_{t=1}^T\langle x^t, z^t\rangle\right)\right]\right) \\
&\leq \frac{1}{\lambda}\ln\left(\sum_{x\in X}\mathbb{E}_z\left[\exp\left(\lambda\sum_{t=1}^T\langle x^t, z^t\rangle\right)\right]\right) \\
&\leq \frac{1}{\lambda}\ln\left(\sum_{x\in X}\prod_{t=1}^T\mathbb{E}_z\left[\exp\left(\lambda\langle x^t, z^t\rangle\right)\right]\right) \\
&= \frac{1}{\lambda}\ln\left(\sum_{x\in X}\prod_{t=1}^T\mathbb{E}_z\left[\exp\left(\lambda\sum_{i=1}^d x_i^t z_i^t\right)\right]\right) \\
&\leq \frac{1}{\lambda}\ln\left(\sum_{x\in X}\prod_{t=1}^T\prod_{i=1}^d\left(\mathbb{E}_z\left[\exp\left(\lambda z_i^t\right)\right]\right)^{x_i^t}\right),
\end{aligned}
$$

where the last step used the fact that $x_i^t \leq 1$ (and thus $y^{x_i^t}$ is a concave function in $y$). Furthermore, by using the fact that $\mathbb{E}_z\left[\exp\left(\lambda z_i^t\right)\right]$ is just the moment generating function of Laplace random variables evaluated at $\lambda$, it is just $(1-\frac{\lambda^2}{\eta^2})^{-1}$ as long as $\lambda < \eta$. As it is always larger than 1, we can directly bound

$$
\begin{aligned}
\mathbb{E}_z\left[\max_{x\in X}\sum_{t=1}^T\langle x^t, z^t\rangle\right] &\leq \frac{1}{\lambda}\ln\left(\sum_{x\in X}\prod_{t=1}^T\prod_{i=1}^d\left(\mathbb{E}_z\left[\exp\left(\lambda z_i^t\right)\right]\right)^{x_i^t}\right) \\
&\leq \frac{1}{\lambda}\ln\left(\sum_{x\in X}\prod_{t=1}^T\prod_{i=1}^d\left(\frac{1}{1-\frac{\lambda^2}{\eta^2}}\right)^{x_i^t}\right) \leq \frac{1}{\lambda}\ln\left(\sum_{x\in X}\prod_{t=1}^T\prod_{i=1}^d\frac{1}{1-\frac{\lambda^2}{\eta^2}}\right) \\
&= \frac{1}{\lambda}\ln\left(|X|\left(\frac{1}{1-\frac{\lambda^2}{\eta^2}}\right)^{dT}\right) = \frac{1}{\lambda}\ln|X| + \frac{dT}{\lambda}\ln\left(\frac{1}{1-\frac{\lambda^2}{\eta^2}}\right).
\end{aligned}
$$

By using the fact that $\frac{1}{1-x} \leq \exp(2x)$ for all $x \leq \frac{1}{4}$, as long as $\lambda \leq \frac{\eta}{2}$, we will have

$$
\mathbb{E}_z\left[\max_{x\in X}\sum_{t=1}^T\langle x^t, z^t\rangle\right] \leq \frac{1}{\lambda}\ln\left(|X|\left(\frac{1}{1-\frac{\lambda^2}{\eta^2}}\right)^{dT}\right) = \frac{1}{\lambda}\ln|X| + \frac{2dT}{\lambda}\frac{\lambda^2}{\eta^2}.
$$

By picking $\lambda = \frac{\eta\sqrt{\ln|X|}}{2\sqrt{dT}} < \frac{\eta}{2}$ (according to the assumption that $\ln|X| < dT$) gives the bound $\frac{5}{\eta}\sqrt{dT\ln|X|}$, which is what we want. $\qquad\square$