# OpenReview forum: "Follow-the-Perturbed-Leader for Adversarial Markov Decision Processes with Bandit Feedback"
_NeurIPS.cc/2022/Conference — NeurIPS 2022 Accept_

### Official Review · Reviewer_Lks2 · 2022-07-09

**Rating:** 7
**Confidence:** 4
**Soundness:** 4 excellent
**Presentation:** 4 excellent
**Contribution:** 3 good

**Summary:**

This paper analyzes the Follow-the-perturbed-leader algorithm for the episodic, adversarial MDP problem where the learner only receives bandit feedback.

Several technical challenges are identified and solved; regret bounds for FTPL in this setting are derived and shown to be competitive with other methods (usually within a $\sqrt{H}$).

Generalized to the delayed setting and infinite-horizon setting (the later being a first), under the slightly less general assumption of known transition dynamics but with a mild mixing requirement.


**Questions:**

Regarding novelty, it seems that, aside from the different stability assumption, the novelty of the paper is in using many tricks already present in the literature instead of a new analysis technique. I don't think that this assessment really detracts from my opinion of the paper, but am I missing something?

**Limitations:**

The main technical limitation, the looseness of the stability lemma, is presented with adequate discussion.


**Strengths And Weaknesses:**

The presentation and results of this paper are both very strong. On the presentation side, I found the discussion and exposition throughout very clear and very well written for an online learning audience familiar with the standard proof techniques in the literature. Focusing on the difference between the  techniques used in this paper and the techniques used in the rest of the literature works out very well and makes all the results very easy to understand. The discussion throughout really has the right balance of technical details and intuition. The organization is logical, and the paper is a pleasure to read.

The results are well positioned in the literature: they tackle a topical problem (adversarial MDP learning) and fill an obvious algorithmic hole (the use of FTPL). Connections to other algorithms, and a comparison of theoretical results via the regret bound rates, is thorough. Even though the rates aren't necessarily state-of-the-art, the extension to FTPL opens up many future directions and is significant.

---

> ### Author Response · Authors · 2022-08-02
> **Response to Reviewer Lks2**
>
> Thank you for the positive comments! Here are our responses:
>
> 1. **Novelty:** Indeed, our key novelty is the discovery related to the stability term: Standard Eqs. (2) and (3) do not hold provably in our case, preventing previous efforts from applying FTPL to MDP with bandit feedback, but a new alternative that is only $H$ times larger is proposed. Building on top of this important discovery, we do combine many existing techniques to achieve our results in different settings. Still, we believe such combinations are also non-trivial and have their own merits, especially considering that it leads to the first result for the infinite-horizon setting (with bandit feedback and stochastic transition).
>
> We thank you once again for such a valuable review of our paper. We are more than happy to answer any further questions.

---

> > ### Comment · Reviewer_Lks2 · 2022-08-09
> > **Paper looks good**
> >
> > Thanks for the response. I will continue to think this paper deserves a 7.

---

### Official Review · Reviewer_Pgbo · 2022-07-11

**Rating:** 6
**Confidence:** 4
**Soundness:** 3 good
**Presentation:** 3 good
**Contribution:** 3 good

**Summary:**

Motivated by the computation advantage of follow-the-perturbed-leader (FTPL) methods against the online-mirror-descent (OMD) methods, the authors consider studying the regret minimization in the adversarial Markov decision process (AMDP) with bandit feedback using FTPL. They prove that FTPL can also achieve the near-optimal regret bounds in the finite-horizon AMDP. Further, using standard loss estimators, they also prove that their algorithm can achieve the order-optimal regret in the finite-horizon Markov decision process (MDP) with delayed bandit feedback under a more simplified analysis than the OMD-based methods. Taking inspirations from previous work studying infinite-horizon communicating AMDP with full-information feedback, the authors also establish the first algorithm with a sub-linear regret guarantee in communicating infinite-horizon AMDP with bandit feedback and known transition using FTPL.

**Questions:**

1.	On page 8 line 296 – line 297, why the regret in infinite-horizon defined in Eq. (1) is “counterfactual”? I do not seem to find it “counterfactual” since the losses in the first term and the second term in Eq. (1) are evaluated on the states generated by following policy \pi_t and some optimal policy \pi^\ast respectively, which are analogous to them in the regret definition of the finite-horizon MDP.

2.	As previous works have done [6, 7], it is natural to divide the whole horizon into different epochs to learn in the infinite-horizon (discounted) MDP. However, the additional policy switching procedure in Algorithm 5 is used to adjust the state distribution and seems only to be an artifact in the analysis based on the occupancy measure. I wonder whether it is possible to learn in infinite-horizon AMDP using non-occupancy-measure-based analysis (say policy optimization [8]) to remove the policy switching procedure.

3.	What is the concrete hardness to equip FTPL with a delay-adapted loss estimator similar to the UOB-REPS algorithm in [1]?

[6] Zhou, D., He, J., & Gu, Q. (2021, July). Provably efficient reinforcement learning for discounted mdps with feature mapping. In International Conference on Machine Learning (pp. 12793-12802). PMLR.

[7] Wei, C. Y., Jahromi, M. J., Luo, H., & Jain, R. (2021, March). Learning infinite-horizon average-reward mdps with linear function approximation. In International Conference on Artificial Intelligence and Statistics (pp. 3007-3015). PMLR.

[8] He, J., Zhou, D., & Gu, Q. (2022, May). Near-optimal Policy Optimization Algorithms for Learning Adversarial Linear Mixture MDPs. In International Conference on Artificial Intelligence and Statistics (pp. 4259-4280). PMLR.


**Limitations:**

Not applicable.

**Strengths And Weaknesses:**

Strengths

Originality: Several original contributions in this work have been made. Different from most works studying regret minimization in AMDP using OMD (or a similar algorithm called follow-the-regularized-leader (FTRL)), this work considers studying the same task but using FTPL to avoid solving the convex optimization problems over the complicated occupancy measure space. Specifically, the authors point out why directly extending the analysis in existing works studying (contextual) semi-bandit fails and give their modification to overcome this problem. Furthermore, the authors also demonstrate that their algorithm can also achieve the order-optimal regret bound in the finite-horizon AMDP with delayed bandit feedback using the standard loss estimator. Besides, the regret bound of their algorithm has better dependences on the state space size S and the action space size A with a more simplified analysis compared with [1]. Using FTPL, the authors also establish the first algorithm with a sub-linear regret guarantee in communicating infinite-horizon AMDP with known transition based on the policy switching procedure in [2].

Quality: The proposed algorithms and analysis are technically sound.

Clarity: Overall, this paper is well-organized and well-written. The authors also give sufficient comparisons of the algorithms and analysis with previous works. However, I have a question about the presentation. Please see the first question below.

Significance: The regret bound of FTPL in this paper matches the state-of-the-art bound obtained by OMD in the finite-horizon AMDP with unknown transition, is easier to analyze in the AMDP with delayed bandit feedback, and is the first sub-linear regret bound in communicating infinite-horizon AMDP with the known transition. These theoretical results in this paper are meaningful to the community in the sense that they can enable us to better understand the performance of learning AMDP using FTPL.

Weaknesses

Novelty: Though some contributions have been made, considering some key ingredients of the algorithms and analysis are similar to previous works (e.g., introducing the upper occupancy measure to handle the unknown transition as in [3], the policy switching procedure to adjust the state distribution after switching the policy in infinite-horizon AMDP as in [2]), it is currently not very clear whether there are additional technical challenges met and solved by the authors. I would suggest the authors give more explanations about the technical difficulties they met and how they address them. In particular, it would be better if the authors could give more discussions about the difficulty to bound the single-step stability in AMDP (Lemma 3) given the single-step stability in (contextual) semi-bandit [4], the difficulty to bound the EstReg term in AMDP with delayed feedback given the similar “cheating regret” decomposition in [5], and the difficulty to achieve the sub-linear bound of the error-term after adding T perturbations (Lemma 34).

[1] Jin, T., Lancewicki, T., Luo, H., Mansour, Y., & Rosenberg, A. (2022). Near-optimal regret for adversarial mdp with delayed bandit feedback. arXiv preprint arXiv:2201.13172.

[2] Chandrasekaran, G., & Tewari, A. (2021). Online Learning in Adversarial MDPs: Is the Communicating Case Harder than Ergodic?. arXiv preprint arXiv:2111.02024.

[3] Jin, C., Jin, T., Luo, H., Sra, S., & Yu, T. (2020, November). Learning adversarial markov decision processes with bandit feedback and unknown transition. In International Conference on Machine Learning (pp. 4860-4869). PMLR.

[4] Syrgkanis, V., Krishnamurthy, A., & Schapire, R. (2016, June). Efficient algorithms for adversarial contextual learning. In International Conference on Machine Learning (pp. 2159-2168). PMLR.

[5] Van Der Hoeven, D., & Cesa-Bianchi, N. (2022, May). Nonstochastic bandits and experts with arm-dependent delays. In International Conference on Artificial Intelligence and Statistics. PMLR.

---

> ### Author Response · Authors · 2022-08-02
> **Response to Reviewer Pgbo**
>
> Thank you for your insightful questions and concerns! Here are our responses:
>
> 1. **Novelty:** As discussed in Section 3.1, the critical technical challenge met and solved in this work is how to handle the stability term of FTPL: Standard Eqs. (2) and (3) do not hold provably in our case, preventing previous efforts from applying FTPL to MDP with bandit feedback, and we propose a new alternative that is only $H$ times larger.
> Admittedly, in hindsight, the proofs for Lemma 3 and the related Lemma 12 are by no means difficult, but we stress again that this discovery itself is novel and valuable. Similarly, while our "cheating regret" decomposition is similar to [5] and the proof of Lemma 34 follows the idea of [4], note that the point of these results (for both delayed feedback and infinite-horizon settings) is to demonstrate how fruitful our discovery on the stability is, especially considering that before our work there were no no-regret algorithms for the adversarial infinite-horizon setting with bandit feedback and stochastic transitions.
> 2. **Usage of the word "counterfactual":** We use the word "counterfactual" to refer to the fact that in the benchmark term of the regret, the entire state sequence is generated by the optimal policy, with no resets (unlike the finite-horizon case). This might not be the best wording, as we now realize, and we will further clarify it in the revision.
> 3. **Possibility to remove policy switching procedure:** The fundamental difficulty we see is that we are considering the infinite-horizon setting under the average-reward metric and only with the communicating assumption, which makes the switching procedure critical (if not necessary). We want to point out that policy optimization performs well in [6] and [7], as the reviewer mentioned, only because additional assumptions are invoked: [6] considers the discounted-reward metric, which only has an 'effective horizon' of $\frac{1}{1-\gamma}$. On the other hand, [7] makes stronger exploratory assumptions such as uniform mixing (similar to ergodicity) and uniformly excited features. (While Algorithms 1 and 2 in [7] do not need such ergodicity assumptions, they use optimism-based algorithms that are incompatible with adversarial environments.)
> 4. **Equipping FTPL with delay-adapted loss estimators:** Technically speaking, this is because the delay-adapted loss estimator of Jin et al. (2022) heavily relies on the exponential weight scheme (i.e., OMD with the negative entropy regularizer). In their Lemma D.5, they bounded the L1 distance between consecutive occupancy measures by the square root of the KL divergence (Pinsker's inequality). The KL divergence is then bounded by the well-bounded $$\sqrt{\eta^2 \sum_{s,a,h} \mu_{\pi_k}^h(s,a)\left (\sum_{k'+d_{k'}=k}\hat \ell_{k'}^h(s,a)\right )^2}$$ term according to their Lemma D.7. However, the second step (the bound on the KL term) heavily relies on the exponential weight scheme, and it is unclear to us whether FTPL enjoys a similar property. We would leave this for future investigation.
>
> We thank you once again for such a valuable review of our paper. We are more than happy to answer any further questions.

---

> > ### Comment · Reviewer_Pgbo · 2022-08-06
> > **Thanks for the reply**
> >
> > My concerns regarding the novelty, presentation and some technical details are addressed. I would like to keep my score.

---

### Official Review · Reviewer_xS2b · 2022-07-12

**Rating:** 8
**Confidence:** 4
**Soundness:** 4 excellent
**Presentation:** 4 excellent
**Contribution:** 3 good

**Summary:**

This paper studies the sample complexity of MDPs with adversarial reward functions and fixed transition kernel. This paper considers the settings of bandit reward feedback, known and unknown transition, and episodic and infinite horizon. Moreover, this paper also studies the delayed-reward setting. The algorithm studied in this paper is the well-known Follow the Perturbed Leader (FTPL). This paper proves that FTPL achieves sublinear regret in all these settings.

**Questions:**

1. Previous works have studied mirror descent for adversarial MDP and FTPL for online bandit learning. It would be nice to better explain the novelty given these two strands of literature.

2. Why the setting of adversarial MDP given in this paper makes sense -- why only assume the reward function is adversarial?

3. Is it possible to modify the proposed algorithm to the setting where transition kernel is also evolving? Is it possible to obtain a dynamic regret? Is It possible to devise an algorithm with low policy switches?


**Limitations:**

The main limitation is that the transition kernel is fixed but only the reward function can change adversarially. I understand that when the transition is also adversarial, there is some negative results. But it is still possible to consider the case where the transition kernel evolves rather slowly. It would be nice to have a discussion.



**Strengths And Weaknesses:**

Strengths:
FTPL algorithm is a popular algorithm in online learning. It is novel to show that FTPL is able to achieve sublinear regret in various RL settings. The theoretical results seem reliable. The presentation of this paper is quite clear.

Weaknesses:
It would be nice to motivate the readers better understand why FTPL is interesting, compared to OMD or FTRL. Does it offer better regret bounds or is it easier to implement? Moreover, previous works have studied mirror descent for adversarial MDP and FTPL for online bandit learning. It would be nice to better explain the novelty given these two strands of literature.

---

> ### Author Response · Authors · 2022-08-02
> **Response to Reviewer xS2b**
>
> Thank you for your insightful comments! Here are our responses:
>
> 1. **Advantages of FTPL compared to OMD/FTRL:** FTPL is easier to implement. It does not require solving complicated optimization problems, which are necessary for OMD/FTRL-based algorithms.
> 2. **Novelty:** Indeed, these two topics (mirror descent for adversarial MDP and FTPL for bandits) are well studied in the literature. However, rather than directly adapting the analysis of FTPL for (contextual) bandits to AMDPs, we found some unique difficulties, as we discussed at length in Sec 3.1 – Standard Eqs. (2) and (3) do not hold provably in our case, preventing previous efforts from applying FTPL to MDP with bandit feedback, and we propose a new alternative that is only $H$ times larger. We stress again that this discovery itself is novel and fruitful, as evidenced by our applications, especially the first no-regret algorithm for the infinite-horizon setting (with bandit feedback and stochastic transition).
> 3. **Only assuming rewards to be adversarial:** This is simply following a line of literature on this topic (see, e.g., Even-Dar et al. (2009), Neu et al. (2010; 2014), Zimin and Neu (2013), Rosenberg and Mansour (2019), Jin et al. (2020)). Our work should be considered as the first step in applying FTPL to MDP with bandit feedback in an environment with adversarial components. We do think further incorporating evolving transition is an important next step.
> 4. **Evolving transition kernels:** Thank you for referring to this! There is indeed an FTPL-based algorithm (Yu and Mannor, 2009) for evolving dynamics (modeled as a two-player game where the opponent's action gets revealed afterward). This algorithm builds upon the FTPL analysis by Even-Dar et al. (2004): It equips FTPL with an optimistic guess of transitions, but the FTPL analysis (Lemma III.3) directly follows from Even-Dar et al. (2004). As our work directly improves the performance guarantee of Even-Dar et al. (2004), it should be extendable to episodic MDPs with evolving transitions which are revealed at the end of each round.
> However, it is highly unclear at this point how to do the same for unrevealed transitions or the infinite-horizon setting with only the communicating assumption. We will include a brief discussion on evolving transition settings in the Conclusion part of the revision.
> *Additional references*:
> Jia Yuan Yu and Shie Mannor. "Arbitrarily modulated Markov decision processes." Proceedings of the 48h IEEE Conference on Decision and Control (CDC) held jointly with 2009 28th Chinese Control Conference. IEEE, 2009.
> Eyal Even-Dar, Sham M. Kakade, and Yishay Mansour. "Experts in a Markov decision process." Advances in neural information processing systems 17 (2004).
> (The second one is the conference version of Even-Dar et al. (2009) that we cited in our paper.)
> 5. **Dynamic regret:** Thank you for mentioning this! As it is standard in the online learning literature, periodically restarting our algorithm can achieve some dynamic regret bound. Doing so optimally, though, requires the knowledge of how non-stationary the environment will be (to decide how often the restarting should be).
> 6. **Low switching algorithms:** Our algorithm is already low-switching for infinite-horizon settings because of the epoching schedule. For episodic settings, unfortunately, ensuring $o(K)$ switches while preserving $\sqrt K$ regret is known to be impossible even for adversarial bandits; see Dekel et al. (2014). If we do epoching like the infinite-horizon case with $J=K^{2/3}$ epochs, each with length $K^{1/3}$, we can achieve $\mathcal O(K^{2/3})$ regret and $\mathcal O(K^{2/3})$ number of switches at the same time, which is again the best one can hope for according to Dekel et al. (2014).
>
> We thank you once again for such a valuable review of our paper. We are more than happy to answer any further questions.

---

> > ### Comment · Reviewer_xS2b · 2022-08-05
> > **Thanks for the response**
> >
> > The response addresses my questions. I would like to thank the authors for explanation.

---

### Official Review · Reviewer_Yxms · 2022-07-17

**Rating:** 7
**Confidence:** 4
**Soundness:** 4 excellent
**Presentation:** 3 good
**Contribution:** 3 good

**Summary:**

This paper studies the following variations of problems of sequential learning in tabular adversarial MDP with bandit feedback: learning in episodic MDP for the setting with known and unknown transitions of MDP, learning with delayed feedback and infinite-horizon setting. The main novelty of this work is that all proposed algorithms are based on the FTPL method, which often allows to develop more computationally efficient algorithms.

**Questions:**

The analysis is clear, so I have only minor comments:

- Line 9 in Algorithm 1: $\hat{l}_{1:k-1}$ is not defined
- Equation after line 215: $p_k(\pi)$ is not defined

**Limitations:**

The Algorithm 6 is only computationally efficient in the presence of the oracle and Algorithm 7 is computationally inefficient.

**Strengths And Weaknesses:**

- On my opinion,  the extension of  FTPL method to the AMDP setting (without assumption on the smallest visitation probability) is an important contribution, as the proposed algorithm is much more computationally efficient than running OMD to compute an occupancy measure as it was done in the previous works for the cases of both known and unknown transitions.
- On the technical side, Lemma 3 is a straightforward but important result as generalises the analysis of FTRL made for bandit problems to bound the regret in all considered settings. In the same time, once Lemma 3 is applied, there is no hope to get a regret bound that would be optimal in H, so in the future studies a refinement of this lemma would be needed.
- In the delayed feedback setting the result does not improve the current state of the art, but technically it is interesting to see that the FTPL can be applied.  Moreover, the obtained regret guarantees for the Algorithm 4 are only slightly worse in a non-leading term.
- I found the result of section 5 exiting, as this is the first result in the setting of infinite AMDP with bandit feedback, even though the obtained regret bound scales as $T^5/6$ (or $T^2/3$ for the computationally inefficient algorithm). While the authors adapt the analysis of Chandrasekaran and Tewari (2021),  they also show that their algorithm is computationally efficient under the assumption of access to the oracle.

---

> ### Author Response · Authors · 2022-08-02
> **Response to Reviewer Yxms**
>
> Thank you for your insightful comments! Here are our responses:
>
> 1. **Improving Lemma 3:** Removing the extra $H$ coming out from Lemma 3 is indeed an exciting and important topic because, as we discussed in the Conclusion section, this will lead to the first optimal $\mathfrak D$-related regret term (i.e., $H^{3/2}\sqrt{\mathfrak D}$) while preserving the computational efficiency. Improving Lemma 3 is left for future investigation.
> 2. **Minor comments:** $\hat \ell_{1:k-1}$ in Algorithm 1 is an analog of $\hat \ell_{0:k-1}$ (defined in Line 192) and denotes $\sum_{k’=1}^{k-1}\hat \ell_{k’}$. We will clarify the definition of $\hat \ell_{l:r}$. Besides, $p_k(\pi)$ is actually defined in Line 211. We will revise our paper according to your comments.
>
> We thank you once again for such a valuable review of our paper. We are more than happy to answer any further questions.

---

### Meta-Review · Area_Chair_37Sw · 2022-08-23

**Recommendation:** Accept
**Confidence:** Certain

**Metareview:**

This paper has received uniformly good reviews, and the reviewers were all happy with the author responses as well. Thus, the paper is clearly suitable for being published at NeurIPS 2022. I encourage the authors to execute all the small updates promised in the rebuttal period when preparing the final version of the paper.

**Award:**

No

---

### Decision · Program_Chairs · 2022-09-14

Accept